# Robust Persistence Diagrams using Reproducing Kernels

**Siddharth Vishwanath**
The Pennsylvania State University
suv87@psu.edu

**Kenji Fukumizu**[*]
The Institute of Statistical Mathematics
fukumizu@ism.ac.jp

**Satoshi Kuriki**[*]
The Institute of Statistical Mathematics
kuriki@ism.ac.jp

**Bharath Sriperumbudur**[*]
The Pennsylvania State University
bks18@psu.edu

## Abstract

Persistent homology has become an important tool for extracting geometric and topological features from data, whose multi-scale features are summarized in a persistence diagram. From a statistical perspective, however, persistence diagrams are very sensitive to perturbations in the input space. In this work, we develop a framework for constructing robust persistence diagrams from superlevel filtrations of robust density estimators constructed using reproducing kernels. Using an analogue of the influence function on the space of persistence diagrams, we establish the proposed framework to be less sensitive to outliers. The robust persistence diagrams are shown to be consistent estimators in bottleneck distance, with the convergence rate controlled by the smoothness of the kernel—this in turn allows us to construct uniform confidence bands in the space of persistence diagrams. Finally, we demonstrate the superiority of the proposed approach on benchmark datasets.

## 1 Introduction

Given a set of points $\mathbb{X}_n = \{\boldsymbol{X}_1, \boldsymbol{X}_2, \ldots, \boldsymbol{X}_n\}$ observed from a probability distribution $\mathbb{P}$ on an input space $\mathcal{X} \subseteq \mathbb{R}^d$, understanding the shape of $\mathbb{X}_n$ sheds important insights on low-dimensional geometric and topological features which underlie $\mathbb{P}$, and this question has received increasing attention in the past few decades. To this end, Topological Data Analysis (TDA), with a special emphasis on persistent homology [20, 44], has become a mainstay for extracting the shape information from data. In statistics and machine-learning, persistent homology has facilitated the development of novel methodology (e.g., [8, 11, 14]), which has been widely used in a variety of applications dealing with massive, unconventional forms of data (e.g., [5, 22, 43]).

Informally speaking, persistent homology detects the presence of topological features across a range of resolutions by examining a nested sequence of spaces, typically referred to as a *filtration*. The filtration encodes the birth and death of topological features as the resolution varies, and is presented in the form of a concise representation—a persistence diagram or barcode. In the context of data-analysis, there are two different methods for obtaining filtrations. The first is computed from the pairwise Euclidean distances of $\mathbb{X}_n$, such as the Vietoris-Rips, Čech, and Alpha filtrations [20]. The second approach is based on choosing a function on $\mathcal{X}$ that reflects the density of $\mathbb{P}$ (or its approximation based on $\mathbb{X}_n$), and, then, constructing a filtration. While the two approaches explore the topological features governing $\mathbb{P}$ in different ways, in essence, they generate similar insights.

---

[*]Authors arranged alphabetically

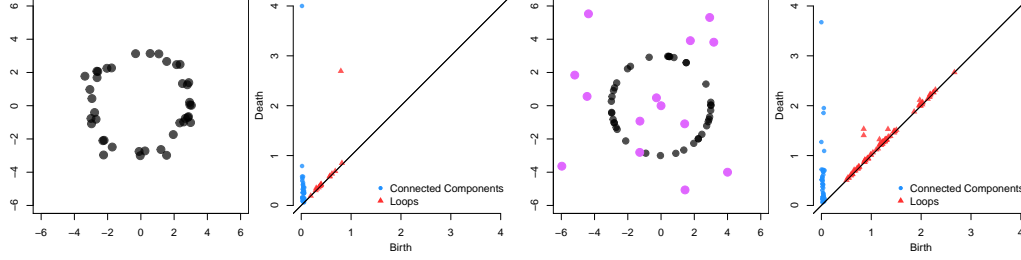

Figure 1: (Left) $\mathbb{X}_n$ is sampled from a circle with small perturbations to each point. The persistence diagram detects the presence of the loop, as guaranteed by the stability of persistence diagrams [12, 16]. (Right) $\mathbb{X}_n$ is sampled from a circle but with just a few outliers. The resulting persistence diagram changes dramatically — the persistence of the main loop plummets, and other spurious loops appear, as elaborated in Section 2.

Despite obvious advantages, the adoption of persistent homology in mainstream statistical methodology is still limited. An important limitation among others, in the statistical context, is that the resulting persistent homology is highly sensitive to outliers. While the stability results of [12, 16] guarantee that small perturbations on all of $\mathbb{X}_n$ induce only small changes in the resulting persistence diagrams, a more pathological issue arises when a small fraction of $\mathbb{X}_n$ is subject to very large perturbations. Figure 1 illustrates how inference from persistence diagrams can change dramatically when $\mathbb{X}_n$ is contaminated with only a few outliers. Another challenge is the mathematical difficulty in performing sensitivity analysis in a formal statistical context. Since the space of persistence diagrams has an unusual mathematical structure, it falls victim to issues such as non-uniqueness of Fréchet means and unbounded curvature of geodesics [18, 29, 36]. With this background, the *central objective* of this paper is to develop outlier robust persistence diagrams, develop a framework for examining the sensitivity of the resulting persistence diagrams to noise, and establish statistical convergence guarantees. To the best of our knowledge, not much work has been carried out in this direction. Bendich et al. [4] construct persistence diagrams from Rips filtrations on $\mathbb{X}_n$ by replacing the Euclidean distance with diffusion distance, Brécheteau and Levrard [7] use a coreset of $\mathbb{X}_n$ for computing persistence diagrams from the distance-to-measure, and Anai et al. [2] use weighted-Rips filtrations on $\mathbb{X}_n$ to construct more stable persistent diagrams. However, no sensitivity analysis of the resultant diagrams are carried out in [2, 4, 7] to demonstrate their robustness.

**Contributions.** The main contributions of this work are threefold. 1) We propose robust persistence diagrams constructed from filtrations induced by an RKHS-based robust KDE (kernel density estimator) [27] of the underlying density function of $\mathbb{P}$ (Section 3). While this idea of inducing filtrations by an appropriate function—[13, 21, 32] use KDE, distance-to-measure (DTM) and kernel distance (KDist), respectively—has already been explored, we show the corresponding persistence diagrams to be less robust compared to our proposal. 2) In Section 4.1, we generalize the notions of *influence function* and *gross error sensitivity*—which are usually defined for normed spaces—to the space of persistence diagrams, which lack the vector space structure. Using these generalized notions, we investigate the sensitivity of persistence diagrams constructed from filtrations induced by different functions (e.g., KDE, robust KDE, DTM) and demonstrate the robustness of the proposed method, both mathematically (Remark 4.3) and numerically (Section 5). 3) We establish the statistical consistency of the proposed robust persistence diagrams and provide uniform confidence bands by deriving exponential concentration bounds for the uniform deviation of the robust KDE (Section 4.2).

**Definitions and Notations.** For a metric space $\mathcal{X}$, the ball of radius $r$ centered at $\boldsymbol{x} \in \mathcal{X}$ is denoted by $B_{\mathcal{X}}(\boldsymbol{x}, r)$. $\mathcal{P}(\mathbb{R}^d)$ is the set of all Borel probability measures on $\mathbb{R}^d$, and $\mathcal{M}(\mathbb{R}^d)$ denotes the set of probability measures on $\mathbb{R}^d$ with compact support and *tame* density function (See Section 2). $\delta_{\boldsymbol{x}}$ denotes a Dirac measure at $\boldsymbol{x}$. For bandwidth $\sigma > 0$, $\mathcal{H}_\sigma$ denotes a reproducing kernel Hilbert space (RKHS) with $K_\sigma : \mathbb{R}^d \times \mathbb{R}^d \to \mathbb{R}$ as its reproducing kernel. We denote by $\Phi_\sigma(\boldsymbol{x}) = K_\sigma(\cdot, \boldsymbol{x}) \in \mathcal{H}_\sigma$, the feature map associated with $K_\sigma$, which embeds $\boldsymbol{x} \in \mathbb{R}^d$ into $\Phi_\sigma(\boldsymbol{x}) \in \mathcal{H}_\sigma$. Throughout this paper, we assume that $K_\sigma$ is radial, i.e., $K_\sigma(\boldsymbol{x}, \boldsymbol{y}) = \sigma^{-d}\psi(\|\boldsymbol{x} - \boldsymbol{y}\|_2/\sigma)$ with $\psi(\|\cdot\|_2)$ being a pdf on $\mathbb{R}^d$, where $\|\boldsymbol{x}\|_2^2 = \sum_{i=1}^d x_i^2$ for $\boldsymbol{x} = (x_1, \ldots, x_d) \in \mathbb{R}^d$. Some common examples include the Gaussian, Matérn and inverse multiquadric kernels. We denote $\|K_\sigma\|_\infty \doteq \sup_{\boldsymbol{x}, \boldsymbol{y} \in \mathbb{R}^d} K_\sigma(\boldsymbol{x}, \boldsymbol{y}) = \sigma^{-d}\psi(0)$. Without loss of generality, we assume $\psi(0) = 1$. For $\mathbb{P} \in \mathcal{P}(\mathbb{R}^d)$, $\mu_{\mathbb{P}} \doteq \int K_\sigma(\cdot, \boldsymbol{y})d\mathbb{P}(\boldsymbol{y}) \in \mathcal{H}_\sigma$ is called the mean embedding of $\mathbb{P}$, and $\mathcal{D}_\sigma \doteq \{\mu_{\mathbb{P}} : \mathbb{P} \in \mathcal{P}(\mathbb{R}^d)\}$ is the space of mean embeddings [30].

## 2   Persistent Homology: Preliminaries

We present the necessary background on persistent homology for completeness. See [9, 42] for a comprehensive introduction.

**Persistent Homology.** Let $\phi : \mathfrak{X} \to \mathbb{R}_{\geq 0}$ be a function on the metric space $(\mathfrak{X}, d)$. At level $r > 0$, the *sublevel* set $\mathfrak{X}_r = \phi^{-1}([0, r]) = \{\boldsymbol{x} \in \mathfrak{X} : \phi(\boldsymbol{x}) \leq r\}$ encodes the topological information in $\mathfrak{X}$. For $r < s$, the sublevel sets are nested, i.e., $\mathfrak{X}_r \subseteq \mathfrak{X}_s$. Thus $\{\mathfrak{X}_r\}_{0 \leq r < \infty}$ is a nested sequence of topological spaces, called a *filtration*, denoted by $\mathrm{Sub}(\phi)$, and $\phi$ is called the *filter function*. As the level $r$ varies, the evolution of the topology is captured in the filtration. Roughly speaking, new cycles (i.e., connected components, loops, voids and higher order analogues) can appear or existing cycles can merge. A new $k$-dimensional feature is said to be born at $b \in \mathbb{R}$ when a nontrivial $k$-cycle appears in $\mathfrak{X}_b$. The same $k$-cycle dies at level $d > b$ when it disappears in all $\mathfrak{X}_{d+\epsilon}$ for $\epsilon > 0$. Persistent homology is an algebraic module which tracks the *persistence pairs* $(b, d)$ of births $b$ and deaths $d$ with multiplicity $\mu$ across the entire filtration $\mathrm{Sub}(\phi)$. Mutatis mutandis, a similar notion holds for superlevel sets $\mathfrak{X}^r = \phi^{-1}([r, \infty))$, inducing the filtration $\mathrm{Sup}(\phi)$. For $r < s$, the inclusion $\mathfrak{X}^r \supseteq \mathfrak{X}^s$ is reversed and a cycle born at $b$ dies at a level $d < b$, resulting in the persistence pair $(d, b)$ instead. Figure 2 shows 3 connected components in the superlevel set for $r = 8$. The components were born as $r$ swept through the blue points, and die when $r$ approaches the red points. In practice, the filtrations are computed on a grid representation of the underlying space using cubical homology. We refer the reader to Appendix E for more details.

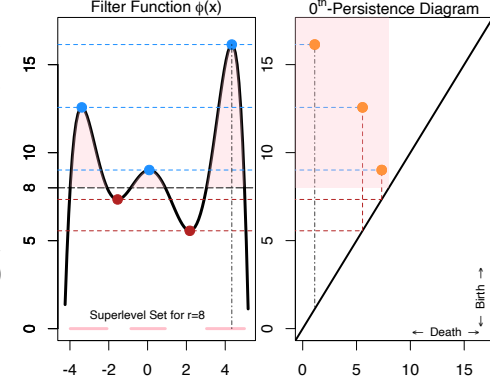

Figure 2: $\mathrm{Dgm}\,(\mathrm{Sup}(\phi))$ for $\phi : \mathbb{R} \to \mathbb{R}$.

**Persistence Diagrams.** By collecting all persistence pairs, the persistent homology features are concisely represented as a persistence diagram $\mathrm{Dgm}\,(\mathrm{Sub}(\phi)) \doteq \{(b, d) \in \mathbb{R}^2 : 0 \leq b < d \leq \infty\}$. A similar definition carries over to $\mathrm{Dgm}\,(\mathrm{Sup}(\phi))$, using $(d, b)$ instead. See Figure 2 for an illustration. When the context is clear, we drop the reference to the filtration and simply write $\mathrm{Dgm}(\phi)$. The $k^{th}$ persistence diagram is the subset of $\mathrm{Dgm}(\phi)$ corresponding to the $k$-dimensional features. The space of persistence diagrams is the locally-finite multiset of points on $\Omega = \{(x, y) : 0 \leq x < y \leq \infty\}$, endowed with the family of $p$-Wasserstein metrics $W_p$, for $1 \leq p \leq \infty$. We refer the reader to [18, 19] for a thorough introduction. $W_\infty$ is commonly referred to as the *bottleneck distance*.

**Definition 2.1.** *Given two persistence diagrams $D_1$ and $D_2$, the bottleneck distance is given by*

$$W_\infty\,(D_1, D_2) = \inf_{\gamma \in \Gamma} \sup_{p \in D_1 \cup \Delta} \|p - \gamma(p)\|_\infty,$$

*where $\Gamma = \{\gamma : D_1 \cup \Delta \to D_2 \cup \Delta\}$ is the set of all bijections from $D_1$ to $D_2$, including the diagonal $\Delta = \{(x, y) \in \mathbb{R}^2 : 0 \leq x = y \leq \infty\}$ with infinite multiplicity.*

An assumption we make at the outset is that the filter function $f$ is *tame*. Tameness is a metric regularity condition which ensures that the number of points on the persistence diagrams are finite, and, in addition, the number of nontrivial cycles which share identical persistence pairings are also finite. Tame functions satisfy the celebrated stability property w.r.t. the bottleneck distance.

**Proposition 2.2** (Stability of Persistence Diagrams [12, 16]). *Given two tame functions $f, g : \mathfrak{X} \to \mathbb{R}$,*

$$W_\infty\,(\mathrm{Dgm}(f), \mathrm{Dgm}(g)) \leq \|f - g\|_\infty.$$

The space of persistence diagrams is, in general, challenging to work with. However, the stability property provides a handle on the persistence space through the function space of filter functions.

## 3   Robust Persistence Diagrams

Given $\mathbb{X}_n = \{\boldsymbol{X}_1, \boldsymbol{X}_2, \ldots, \boldsymbol{X}_n\} \subseteq \mathbb{R}^d$ drawn iid from a probability distribution $\mathbb{P} \in \mathcal{M}(\mathbb{R}^d)$ with density $f$, the corresponding persistence diagram can be obtained by considering a filter function $\phi_n : \mathbb{R}^d \to \mathbb{R}$, constructed from $\mathbb{X}_n$ as an approximation to its population analogue, $\phi_{\mathbb{P}} : \mathbb{R}^d \to \mathbb{R}$, that carries the topological information of $\mathbb{P}$.

Commonly used $\phi_{\mathbb{P}}$ include the (i) kernelized density, $f_\sigma$, (ii) Kernel Distance (KDist), $d_{\mathbb{P}}^{K_\sigma}$, and (iii) distance-to-measure (DTM), $d_{\mathbb{P},m}$, which are defined as:

$$f_\sigma(\boldsymbol{x}) \doteq \int_{\mathcal{X}} K_\sigma(\boldsymbol{x}, \boldsymbol{y}) d\mathbb{P}(\boldsymbol{y}) \; ; \quad d_{\mathbb{P}}^{K_\sigma} \doteq \|\mu_{\delta_{\boldsymbol{x}}} - \mu_{\mathbb{P}}\|_{\mathcal{H}_\sigma} \; ; \quad d_{\mathbb{P},m}(\boldsymbol{x}) \doteq \sqrt{\frac{1}{m} \int_0^m F_{\boldsymbol{x}}^{-1}(u) du},$$

where $F_{\boldsymbol{x}}(t) = \mathbb{P}(\|\boldsymbol{X} - \boldsymbol{x}\|_2 \leq t)$ and $\sigma, m > 0$. For these $\phi_{\mathbb{P}}$, the corresponding empirical analogues, $\phi_n$, are constructed by replacing $\mathbb{P}$ with the empirical measure, $\mathbb{P}_n \doteq \frac{1}{n} \sum_{i=1}^n \delta_{\boldsymbol{X}_i}$. For example, the empirical analogue of $f_\sigma$ is the familiar kernel density estimator (KDE), $f_\sigma^n = \frac{1}{n} \sum_{i=1}^n K_\sigma(\cdot, \boldsymbol{X}_i)$. While KDE and KDist encode the shape and distribution of mass for $\mathrm{supp}(\mathbb{P})$ by approximating the density $f$ (sublevel sets of KDist are rescaled versions of superlevel sets of KDE [13, 32]), DTM, on the other hand, approximates the distance function to $\mathrm{supp}(\mathbb{P})$.

Since $\phi_n$ is based on $\mathbb{P}_n$, it is sensitive to outliers in $\mathbb{X}_n$, which, in turn affect the persistence diagrams (as illustrated in Figure 1). To this end, in this paper, we propose *robust persistence diagrams* constructed using superlevel filtrations of a robust density estimator of $f$, i.e., the filter function, $\phi_n$ is chosen to be a robust density estimator of $f$. Specifically, we use the robust KDE, $f_{\rho,\sigma}^n$, introduced by [27] as the filter function, which is defined as a solution to the following M-estimation problem:

$$f_{\rho,\sigma}^n \doteq \arg\inf_{g \in \mathcal{G}} \int_{\mathcal{X}} \rho\left(\|\Phi_\sigma(\boldsymbol{y}) - g\|_{\mathcal{H}_\sigma}\right) d\mathbb{P}_n(\boldsymbol{y}), \tag{1}$$

where $\rho : \mathbb{R}_{\geq 0} \to \mathbb{R}_{\geq 0}$ is a robust loss function, and $\mathcal{G} = \mathcal{H}_\sigma \cap \mathcal{D}_\sigma = \mathcal{D}_\sigma$ is the hypothesis class. Observe that when $\rho(z) = \frac{1}{2} z^2$, the unique solution to Eq. (1) is given by the KDE, $f_\sigma^n$. Therefore, a robust KDE is obtained by replacing the square loss with a *robust loss*, which satisfies the following assumptions. These assumptions, which are similar to those of [27, 39] guarantee the existence and uniqueness (if $\rho$ is convex) of $f_{\rho,\sigma}^n$ [27], and are satisfied by most robust loss functions, including the Huber loss, $\rho(z) = \frac{1}{2} z^2 \mathbb{1}\{z \leq 1\} + \left(z - \frac{1}{2}\right) \mathbb{1}\{z > 1\}$ and the Charbonnier loss, $\rho(z) = \sqrt{1 + z^2} - 1$.

($\mathcal{A}1$) $\rho$ is strictly-increasing and $M$-Lipschitz, with $\rho(0) = 0$.

($\mathcal{A}2$) $\rho'(x)$ is continuous and bounded with $\rho'(0) = 0$ .

($\mathcal{A}3$) $\varphi(x) = \rho'(x)/x$ is bounded, $L$-Lipschitz and continuous, with $\varphi(0) < \infty$.

($\mathcal{A}4$) $\rho''$ exists, with $\rho''$ and $\varphi$ nonincreasing.

Unlike for squared loss, the solution $f_{\rho,\sigma}^n$ cannot be obtained in a closed form. However, it can be shown to be the fixed point of an iterative procedure, referred to as KIRWLS algorithm [27]. The KIRWLS algorithm starts with initial weights $\{w_i^{(0)}\}_{i=1}^n$ such that $\sum_{i=1}^n w_i^{(0)} = 1$, and generates the iterative sequence of estimators $\{f_{\rho,\sigma}^{(k)}\}_{k \in \mathbb{N}}$ as

$$f_{\rho,\sigma}^{(k)} = \sum_{i=1}^n w_i^{(k-1)} K_\sigma(\cdot, \boldsymbol{X}_i) \; ; \qquad w_i^{(k)} = \frac{\varphi(\|\Phi_\sigma(\boldsymbol{X}_i) - f_{\rho,\sigma}^{(k)}\|_{\mathcal{H}_\sigma})}{\sum_{j=1}^n \varphi(\|\Phi_\sigma(\boldsymbol{X}_j) - f_{\rho,\sigma}^{(k)}\|_{\mathcal{H}_\sigma})}.$$

Intuitively, note that if $\boldsymbol{X}_i$ is an outlier, then the corresponding weight $w_i$ is small (since $\varphi$ is nonincreasing) and therefore less weight is given to the contribution of $\boldsymbol{X}_i$ in the density estimator. Hence, the weights serve as a measure of *inlyingness*—smaller (*resp.* larger) the weights, lesser (*resp.* more) inlying are the points. When $\mathbb{P}_n$ is replaced by $\mathbb{P}$, the solution of Eq. (1) is its population analogue, $f_{\rho,\sigma}$. Although $f_{\rho,\sigma}$ does not admit a closed form solution, it can be shown [27] that there exists a non-negative real-valued function $w_\sigma$ satisfying $\int_{\mathbb{R}^d} w_\sigma(\boldsymbol{x}) d\mathbb{P}(\boldsymbol{x}) = 1$ such that

$$f_{\rho,\sigma} = \int_{\mathbb{R}^d} K_\sigma(\cdot, \boldsymbol{x}) w_\sigma(\boldsymbol{x}) d\mathbb{P}(\boldsymbol{x}) = \int_{\mathbb{R}^d} \frac{\varphi(\|\Phi_\sigma(\boldsymbol{x}) - f_{\rho,\sigma}\|_{\mathcal{H}_\sigma})}{\int_{\mathbb{R}^d} \varphi(\|\Phi_\sigma(\boldsymbol{y}) - f_{\rho,\sigma}\|_{\mathcal{H}_\sigma}) d\mathbb{P}(\boldsymbol{y})} K_\sigma(\cdot, \boldsymbol{x}) \, d\mathbb{P}(\boldsymbol{x}), \tag{2}$$

where $w_\sigma$ acts as a population analogue of the weights in KIRWLS algorithm.

To summarize our proposal, the fixed point of the KIRWLS algorithm, which yields the robust density estimator $f_{\rho,\sigma}^n$, is used as the filter function to obtain a robust persistence diagram of $\mathbb{X}_n$. On the computational front, note that $f_{\rho,\sigma}^n$ is computationally more complex than the KDE, $f_\sigma^n$, requiring $O(n\ell)$ computations compared to $O(n)$ of the latter, with $\ell$ being the number of iterations required to reach the fixed point of KIRWLS. However, once these filter functions are computed, the corresponding persistence diagrams have similar computational complexity as both require computing superlevel sets, which, in turn, require function evaluations that scale as $O(n)$ for both $f_{\rho,\sigma}^n$ and $f_\sigma^n$.

# 4  Theoretical Analysis of Robust Persistence Diagrams

In this section, we investigate the theoretical properties of the proposed robust persistence diagrams. First, in Section 4.1, we examine the sensitivity of persistence diagrams to outlying perturbations through the notion of *metric derivative* and compare the effect of different filter functions. Next, in Section 4.2, we establish consistency and convergence rates for the robust persistence diagram to its population analogue. These results allow to construct uniform confidence bands for the robust persistence diagram. The proofs of the results are provided in Appendix A.

## 4.1  A measure of sensitivity of persistence diagrams to outliers

The influence function and gross error sensitivity are arguably the most popular tools in robust statistics for diagnosing the sensitivity of an estimator to a single adversarial contamination [23, 26]. Given a statistical functional $T : \mathcal{P}(\mathfrak{X}) \to (V, \|\cdot\|_V)$, which takes an input probability measure $\mathbb{P} \in \mathcal{P}(\mathfrak{X})$ on the input space $\mathfrak{X}$ and produces a statistic $\mathbb{P} \mapsto T(\mathbb{P})$ in some normed space $(V, \|\cdot\|_V)$, the *influence function* of $\boldsymbol{x} \in \mathfrak{X}$ at $\mathbb{P}$ is given by the Gâteaux derivative of $T$ at $\mathbb{P}$ restricted to the space of signed Borel measures with zero expectation:

$$\mathsf{IF}(T; \mathbb{P}, \boldsymbol{x}) \doteq \frac{\partial}{\partial \epsilon} T\Big((1-\epsilon)\mathbb{P} + \epsilon\delta_{\boldsymbol{x}}\Big)\Big|_{\epsilon=0} = \lim_{\epsilon \to 0} \frac{T\left((1-\epsilon)\mathbb{P} + \epsilon\delta_{\boldsymbol{x}}\right) - T(\mathbb{P})}{\epsilon},$$

and the *gross error sensitivity* at $\mathbb{P}$ is given by $\Gamma(T; \mathbb{P}) \doteq \sup_{\boldsymbol{x} \in \mathfrak{X}} \|\mathsf{IF}(T; \mathbb{P}, \boldsymbol{x})\|_V$. However, a persistence diagram (which is a statistical functional) does not take values in a normed space and therefore the notion of influence functions has to be generalized to metric spaces through the concept of a metric derivative: Given a complete metric space $(X, d_X)$ and a curve $s : [0, 1] \to X$, the *metric derivative* at $\epsilon = 0$ is given by $|s'|(0) \doteq \lim_{\epsilon \to 0} \frac{1}{\epsilon} d_X(s(0), s(\epsilon))$. Using this generalization, we have the following definition, which allows to examine the influence an outlier has on the persistence diagram obtained from a filtration.

**Definition 4.1.** *Given a probability measure $\mathbb{P} \in \mathcal{P}(\mathbb{R}^d)$ and a filter function $\phi_{\mathbb{P}}$ depending on $\mathbb{P}$, the* persistence influence *of a perturbation $\boldsymbol{x} \in \mathbb{R}^d$ on $\mathsf{Dgm}(\phi_{\mathbb{P}})$ is defined as*

$$\Psi(\phi_{\mathbb{P}}; \boldsymbol{x}) = \lim_{\epsilon \to 0} \frac{1}{\epsilon} W_\infty\left(\mathsf{Dgm}\left(\phi_{\mathbb{P}_{\boldsymbol{x}}^\epsilon}\right), \mathsf{Dgm}\left(\phi_{\mathbb{P}}\right)\right),$$

*where $\mathbb{P}_{\boldsymbol{x}}^\epsilon \doteq (1-\epsilon)\mathbb{P} + \epsilon\delta_{\boldsymbol{x}}$, and the* gross-influence *is defined as $\Gamma(\phi_{\mathbb{P}}) = \sup_{\boldsymbol{x} \in \mathbb{R}^d} \Psi(\phi_{\mathbb{P}}; \boldsymbol{x})$.*

For $\epsilon > 0$, let $f_{\rho,\sigma}^{\epsilon;\boldsymbol{x}}$ be the robust KDE associated with the probability measure $\mathbb{P}_{\boldsymbol{x}}^\epsilon$. The following result (proved in Appendix A.1) bounds the persistence influence for the persistence diagram induced by the filter function $f_{\rho,\sigma}$, which is the population analogue of robust KDE.

**Theorem 4.2.** *For a loss $\rho$ satisfying $(\mathcal{A}1)$–$(\mathcal{A}3)$, and $\sigma > 0$, if $\lim_{\epsilon \to 0} \frac{1}{\epsilon}\left(f_{\rho,\sigma}^{\epsilon;\boldsymbol{x}} - f_{\rho,\sigma}\right)$ exists, then the persistence influence of $\boldsymbol{x} \in \mathbb{R}^d$ on $\mathsf{Dgm}(f_{\rho,\sigma})$ satisfies*

$$\Psi(f_{\rho,\sigma}; \boldsymbol{x}) \leq \|K_\sigma\|_\infty^{\frac{1}{2}} \rho'\left(\|\Phi_\sigma(\boldsymbol{x}) - f_{\rho,\sigma}\|_{\mathcal{H}_\sigma}\right)\left(\int_{\mathbb{R}^d} \zeta\left(\|\Phi_\sigma(\boldsymbol{y}) - f_{\rho,\sigma}\|_{\mathcal{H}_\sigma}\right)d\mathbb{P}(\boldsymbol{y})\right)^{-1}, \quad (3)$$

*where $\zeta(z) = \varphi(z) - z\varphi'(z)$.*

**Remark 4.3.** We make the following observations from Theorem 4.2.

  **(i)** Choosing $\rho(z) = \frac{1}{2}z^2$ and noting that $\varphi(z) = \rho''(z) = 1$, a similar analysis, as in the proof of Theorem 4.2, yields a bound for the persistence influence of the KDE as

$$\Psi(f_\sigma; \boldsymbol{x}) \leq \sigma^{-d/2} \|\Phi_\sigma(\boldsymbol{x}) - f_\sigma\|_{\mathcal{H}_\sigma}.$$

On the other hand, for robust loss functions, the term in Eq. (3) involving $\rho'$ is bounded because of $(\mathcal{A}2)$, making them less sensitive to very large perturbations. In fact, for nonincreasing $\varphi$, it can be shown (see Appendix C) that

$$\Psi(f_{\rho,\sigma}; \boldsymbol{x}) \leq \sigma^{-d/2} w_\sigma(\boldsymbol{x}) \|\Phi_\sigma(\boldsymbol{x}) - f_{\rho,\sigma}\|_{\mathcal{H}_\sigma},$$

where, in contrast to KDE, the measure of inlyingness, $w_\sigma$, weighs down extreme outliers.

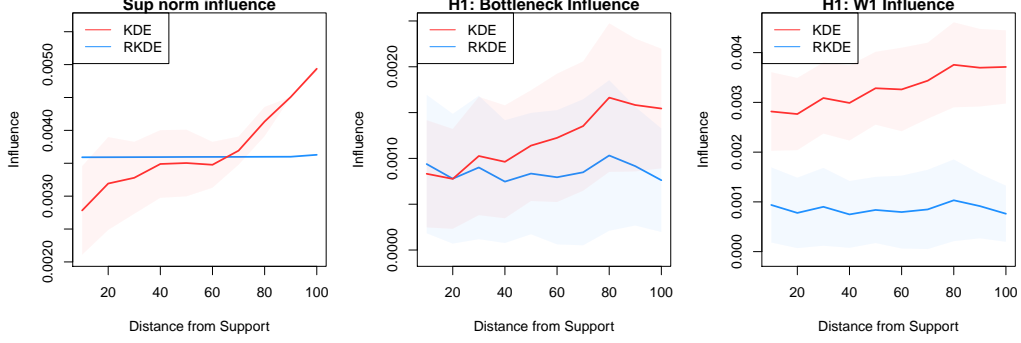

Figure 3: Points $\mathbb{X}_n$ are sampled from $\mathbb{P}$ with nontrivial $1^{st}$-order homological features and outliers $\mathbb{Y}_m$ are added at a distance $r$ from the support of $\mathbb{P}$. (Left) The average $L_\infty$ distance between the density estimators computed using $\mathbb{X}_n$ and $\mathbb{X}_n \cup \mathbb{Y}_m$ as $r$ increases. (Center) The average $W_\infty$ distance between the corresponding persistence diagrams for the $1^{st}$-order homological features. (Right) The $W_1$ distance (defined in Eq. E.1 in Appendix E) between the same persistence diagrams. The results show that the outliers $\mathbb{Y}_m$ have little influence on the persistence diagrams from the robust KDEs. In contrast, as the outliers become more extreme (i.e., $r$ increases) their influence on the persistence diagrams from the KDE becomes more prominent.

**(ii)** For the generalized Charbonnier loss (a robust loss function), given by $\rho(z) = \left(1 + z^2\right)^{\alpha/2} - 1$ for $1 \le \alpha < 2$, the persistence influence satisfies

$$\Psi\left(f_{\rho,\sigma}; \boldsymbol{x}\right) \le \sigma^{-d/2} \left(1 + \|\Phi_\sigma(\boldsymbol{x}) - f_{\rho,\sigma}\|_{\mathcal{H}_\sigma}^2\right)^{\frac{\alpha-1}{2}} \left(1 + \int_{\mathbb{R}^d} \|\Phi_\sigma(\boldsymbol{y}) - f_{\rho,\sigma}\|_{\mathcal{H}_\sigma}^2 \, d\mathbb{P}(\boldsymbol{y})\right)^{\frac{1-\alpha}{2}}.$$

Note that for $\alpha = 1$, the bound on the persistence influence $\Psi\left(f_{\rho,\sigma}; \boldsymbol{x}\right)$ does not depend on how extreme the outlier $\boldsymbol{x}$ is. Similarly, for the Cauchy loss, given by $\rho(z) = \log(1 + z^2)$, we have

$$\Psi\left(f_{\rho,\sigma}; \boldsymbol{x}\right) \le \sigma^{-d/2} \left(1 + \int_{\mathbb{R}^d} \|\Phi_\sigma(\boldsymbol{y}) - f_{\rho,\sigma}\|_{\mathcal{H}_\sigma}^2 \, d\mathbb{P}(\boldsymbol{y})\right).$$

This shows that for large perturbations, the gross error sensitivity for the Cauchy and Charbonnier losses are far more stable than that of KDE. This behavior is also empirically illustrated in Figure 3. The experiment is detailed in Appendix C.

**(iii)** For the DTM function, it can be shown that

$$\Psi\left(d_{\mathbb{P},m}; \boldsymbol{x}\right) \le \frac{2}{\sqrt{m}} \sup\left\{\left|f(\boldsymbol{x}) - \int_{\mathbb{R}^d} f(\boldsymbol{y}) d\mathbb{P}(\boldsymbol{y})\right| : \|\nabla f\|_{L_2(\mathbb{P})} \le 1\right\}. \tag{4}$$

While $d_{\mathbb{P},m}$ cannot be compared to both $f_\sigma$ and $f_{\rho,\sigma}$, as it captures topological information at a different scale, determined by $m$, we point out that when $supp(\mathbb{P})$ is compact, $\Psi\left(d_{\mathbb{P},m}; \boldsymbol{x}\right)$ is not guaranteed to be bounded, unlike in $\Psi\left(f_{\rho,\sigma}; \boldsymbol{x}\right)$. We refer the reader to Appendix C for more details.

It follows from Remark 4.3 that as $\sigma \to 0$, the persistence influence of both the KDE and robust KDE behave as $O(\sigma^{-d})$, showing that the robustness of robust persistence diagrams manifests only in cases where $\sigma > 0$. However, robustness alone has no bearing if the robust persistence diagram and the persistence diagram from the KDE are fundamentally different, i.e., they estimate different quantities as $\sigma \to 0$. The following result (proved in Appendix A.2) shows that as $\sigma \to 0$, $\mathsf{Dgm}\left(f_{\rho,\sigma}\right)$ recovers the same information as that in $\mathsf{Dgm}\left(f_\sigma\right)$, which is same as $\mathsf{Dgm}\left(f\right)$, where $f$ is the density of $\mathbb{P}$.

**Theorem 4.4.** *For a strictly-convex loss $\rho$ satisfying $(\mathcal{A}1)$–$(\mathcal{A}4)$, and $\sigma > 0$, suppose $\mathbb{P} \in \mathcal{M}(\mathbb{R}^d)$ with density $f$, and $f_{\rho,\sigma}$ is the robust KDE. Then $W_\infty\left(\mathsf{Dgm}\left(f_{\rho,\sigma}\right), \mathsf{Dgm}\left(f\right)\right) \to 0$ as $\sigma \to 0$.*

Suppose $\mathbb{P} = (1 - \pi)\mathbb{P}_0 + \pi\mathbb{Q}$, where $\mathbb{P}_0$ corresponds to the true signal which we are interested in studying, and $\mathbb{Q}$ manifests as some ambient noise with $0 < \pi < \frac{1}{2}$. In light of Theorem 4.4, by letting $\sigma \to 0$, along with the topological features of $\mathbb{P}_0$, we are also capturing the topological features of $\mathbb{Q}$, which may obfuscate any statistical inference made using the persistence diagrams. In a manner, choosing $\sigma > 0$ suppresses the noise in the resulting persistence diagrams, thereby making them more stable. On a similar note, the authors in [21] state that for a suitable bandwidth $\sigma > 0$, the level sets of $f_\sigma$ carry the same topological information as $supp(\mathbb{P})$, despite the fact that some subtle details in $f$ may be omitted. In what follows, we consider the setting where robust persistence diagrams are constructed for a fixed $\sigma > 0$.

## 4.2 Statistical properties of robust persistence diagrams from samples

Suppose $\mathsf{Dgm}\left(f_{\rho,\sigma}^n\right)$ is the robust persistence diagram obtained from the robust KDE on a sample $\mathbb{X}_n$ and $\mathsf{Dgm}\left(f_{\rho,\sigma}\right)$ is its population analogue obtained from $f_{\rho,\sigma}$. The following result (proved in Appendix A.3) establishes the consistency of $\mathsf{Dgm}\left(f_{\rho,\sigma}^n\right)$ in the $W_\infty$ metric.

**Theorem 4.5.** *For convex loss $\rho$ satisfying $(\mathcal{A}1)$–$(\mathcal{A}4)$, and fixed $\sigma > 0$, suppose $\mathbb{X}_n$ is observed iid from a distribution $\mathbb{P} \in \mathcal{M}(\mathbb{R}^d)$ with density $f$. Then*

$$W_\infty\left(\mathsf{Dgm}\left(f_{\rho,\sigma}^n\right), \mathsf{Dgm}\left(f_{\rho,\sigma}\right)\right) \xrightarrow{p} 0 \quad \text{as } n \to \infty.$$

We present the convergence rate of the above convergence in Theorem 4.7, which depends on the smoothness of $\mathcal{H}_\sigma$. In a similar spirit to [21], this result paves the way for constructing uniform confidence bands. Before we present the result, we first introduce the notion of *entropy numbers* associated with an RKHS.

**Definition 4.6** (Entropy Number). *Given a metric space $(T, d)$ the $n^{th}$ entropy number is defined as*

$$e_n(T, d) \doteq \inf\left\{\epsilon > 0 : \exists \ \{t_1, t_2, \ldots, t_{2^{n-1}}\} \subset T \ \text{ such that } T \subset \bigcup_{i=1}^{2^{n-1}} B_d(t_i, \epsilon)\right\}.$$

*Further, if $(V, \|\cdot\|_V)$ and $(W, \|\cdot\|_W)$ are two normed spaces and $L : V \to W$ is a bounded, linear operator, then $e_n(L) = e_n(L : V \to W) \doteq e_n\left(L(B_V), \|\cdot\|_W\right)$, where $B_V$ is a unit ball in $V$.*

Loosely speaking, entropy numbers are related to the eigenvalues of the integral operator associated with the kernel $K_\sigma$, and measure the capacity of the RKHS in approximating functions in $L_2(\mathbb{R}^d)$. In our context, the entropy numbers will provide useful bounds on the covering numbers of sets in the hypothesis class $\mathcal{G}$. We refer the reader to [35] for more details. With this background, the following theorem (proved in Appendix A.4) provides a method for constructing uniform confidence bands for the persistence diagram constructed using the robust KDE on $\mathbb{X}_n$.

**Theorem 4.7.** *For convex loss $\rho$ satisfying $(\mathcal{A}1)$–$(\mathcal{A}4)$, and fixed $\sigma > 0$, suppose the kernel $K_\sigma$ satisfies $e_n\left(\text{id} : \mathcal{H}_\sigma \to L_\infty(\mathcal{X})\right) \leq a_\sigma n^{-\frac{1}{2p}}$, where $a_\sigma > 1$, $0 < p < 1$ and $\mathcal{X} \subset \mathbb{R}^d$. Then, for a fixed confidence level $0 < \alpha < 1$,*

$$\sup_{\mathbb{P} \in \mathcal{M}(\mathcal{X})} \mathbb{P}^{\otimes n}\left\{W_\infty\left(\mathsf{Dgm}\left(f_{\rho,\sigma}^n\right), \mathsf{Dgm}\left(f_{\rho,\sigma}\right)\right) > \frac{2M\|K_\sigma\|_\infty^{\frac{1}{2}}}{\mu}\left(\xi(n, p) + \delta\sqrt{\frac{2\log\left(1/\alpha\right)}{n}}\right)\right\} \leq \alpha,$$

*where $\xi(n, p)$ is given by*

$$\xi(n, p) = \begin{cases} \gamma\dfrac{a_\sigma^p}{(1-2p)} \cdot \dfrac{1}{\sqrt{n}} & \text{if } 0 < p < 1/2, \\[2ex] \gamma C\sqrt{a_\sigma} \cdot \dfrac{\log(n)}{\sqrt{n}} & \text{if } p = 1/2, \\[2ex] \gamma\dfrac{p\sqrt{a_\sigma}}{2p-1} \cdot \dfrac{1}{n^{1/4p}} & \text{if } 1/2 < p < 1, \end{cases}$$

*for fixed constants $\gamma > \frac{12}{\sqrt{\log 2}}$, $C > 3 - \log(9a_\sigma)$ and $\mu = 2\min\left\{\varphi(2\|K_\sigma\|_\infty^{\frac{1}{2}}), \rho''(2\|K_\sigma\|_\infty^{\frac{1}{2}})\right\}$.*

**Remark 4.8.** We highlight some salient observations from Theorem 4.7.

(i) If $diam(\mathcal{X}) = r$, and the kernel $K_\sigma$ is $m$-times differentiable, then from [35, Theorem 6.26], the entropy numbers associated with $K_\sigma$ satisfy $e_n\left(\text{id} : \mathcal{H}_\sigma \to L_\infty(\mathcal{X})\right) \leq cr^m n^{-\frac{m}{d}}$. In light of Theorem 4.7, for $p = \frac{d}{2m}$, we can make two important observations. First, as the dimension of the input space $\mathcal{X}$ increases, we have that the rate of convergence decreases; which is a direct consequence from the curse of dimensionality. Second, for a fixed dimension of the input space, the parameter $p$ in Theorem 4.7 can be understood to be inversely proportional to the smoothness of the kernel. Specifically, as the smoothness of the kernel increases, the rate of convergence is faster, and we obtain sharper confidence bands. This makes a case for employing smoother kernels.

(ii) A similar result is obtained in [21, Lemma 8] for persistence diagrams from the KDE, with a convergence rate $O_p(n^{-1/2})$, where the proof relies on a simple application of Hoeffding's inequality, unlike the sophisticated tools the proof of Theorem 4.7 warrants for the robust KDE.

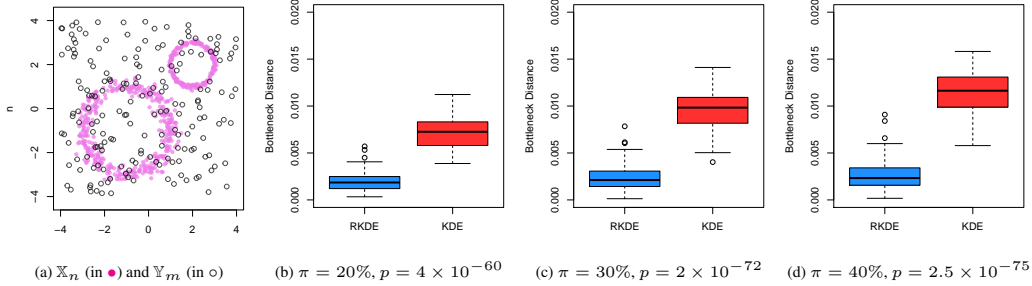

(a) $\mathbb{X}_n$ (in •) and $\mathbb{Y}_m$ (in ○)    (b) $\pi = 20\%, p = 4 \times 10^{-60}$    (c) $\pi = 30\%, p = 2 \times 10^{-72}$    (d) $\pi = 40\%, p = 2.5 \times 10^{-75}$

Figure 4: (a) A realization of $\mathbb{X}_n \cup \mathbb{Y}_m$. (b, c, d) As the noise level $\pi$ increases, boxplots for $W_\infty \left( \mathbf{D}_{\rho,\sigma}, \mathcal{D}_\sigma^\# \right)$ in blue and $W_\infty \left( \mathbf{D}_\sigma, \mathcal{D}_\sigma^\# \right)$ in red show that the robust persistence diagram recovers the underlying signal better.

## 5    Experiments

We illustrate the performance of robust persistence diagrams in machine learning applications through synthetic and real-world experiments.[1] In all the experiments, the kernel bandwidth $\sigma$ is chosen as the median distance of each $\boldsymbol{x}_i \in \mathbb{X}_n$ to its $k^{th}$–nearest neighbour using the Gaussian kernel with the Hampel loss (similar setting as in [27])—we denote this bandwidth as $\sigma(k)$. Since DTM is closely related to the $k$-NN density estimator [6], we choose the DTM smoothing parameter as $m(k) = k/n$. Additionally, the KIRWLS algorithm is run until the relative change of empirical risk $< 10^{-6}$.

**Runtime Analysis.** For $n = 1000$, $\mathbb{X}_n$ is sampled from a torus inside $[0, 2]^3$. For each grid resolution $\alpha \in \{0.04, 0.06, 0.08, 0.10\}$, the robust persistence diagram Dgm $\left( f_{\rho,\sigma}^n \right)$ and the KDE persistence diagram Dgm $\left( f_\sigma^n \right)$ are constructed from the superlevel filtration of cubical homology. The *total* time taken to compute the persistence diagrams is reported in Table 1. The results demonstrate that the computational bottleneck is the persistent homology pipeline, and not the KIRWLS for $f_{\rho,\sigma}^n$.

Table 1: Runtime (in Seconds) for computing Dgm $\left( f_{\rho,\sigma}^n \right)$ and Dgm $\left( f_\sigma^n \right)$ at each grid resolution.

| Grid Resolution | 0.04 | 0.06 | 0.08 | 0.10 |
|---|---|---|---|---|
| Average runtime for Dgm $\left( f_{\rho,\sigma}^n \right)$ | 76.7s | 17.1s | 6.7s | 3.5s |
| Average runtime for Dgm $\left( f_\sigma^n \right)$ | 75.5s | 15.3s | 4.7s | 1.8s |

**Bottleneck Simulation.** The objective of this experiment is to assess how the robust KDE persistence diagram compares to the KDE persistence diagram in recovering the topological features of the underlying signal. $\mathbb{X}_n$ is observed uniformly from two circles and $\mathbb{Y}_m$ is sampled uniformly from the enclosing square such that $m = 200$ and $m/n = \pi \in \{20\%, 30\%, 40\%\}$—shown in Figure 4 (a). For each noise level $\pi$, and for each of $N = 100$ realizations of $\mathbb{X}_n$ and $\mathbb{Y}_m$, the robust persistence diagram $\mathbf{D}_{\rho,\sigma}$ and the KDE persistence diagram $\mathbf{D}_\sigma$ are constructed from the noisy samples $\mathbb{X}_n \cup \mathbb{Y}_m$. In addition, we compute the KDE persistence diagram $\mathcal{D}_\sigma^\#$ on $\mathbb{X}_n$ alone as a proxy for the target persistence diagram one would obtain in the absence of any contamination. The bandwidth $\sigma(k) > 0$ is chosen for $k = 5$. For each realization $i$, bottleneck distances $U_i = W_\infty \left( \mathbf{D}_{\rho,\sigma}, \mathcal{D}_\sigma^\# \right)$ and $V_i = W_\infty \left( \mathbf{D}_\sigma, \mathcal{D}_\sigma^\# \right)$ are computed for $1^{st}$-order homological features. The boxplots and $p$-values for the one-sided hypothesis test $H_0 : U - V = 0$ vs. $H_1 : U - V < 0$ are reported in Figures 4 (b, c, d). The results demonstrate that the robust persistence diagram is noticeably better in recovering the true homological features, and in fact demonstrates superior performance when the noise levels are higher.

**Spectral Clustering using Persistent Homology.** We perform a variant of the six-class benchmark experiment from [1, Section 6.1]. The data comprises of six different 3D "objects": `cube`, `circle`, `sphere`, `3clusters`, `3clustersIn3clusters`, and `torus`. 25 point clouds are sampled from each object with additive Gaussian noise (SD= 0.1), *and* ambient Matérn cluster noise. For each point cloud, $\mathbb{X}_n$, the robust persistence diagram Dgm $\left( f_{\rho,\sigma}^n \right)$ and the persistence diagram Dgm $(d_{\mathbb{X}_n})$, from the distance function, are constructed. Additionally, Dgm $(d_{\mathbb{X}_n})$ is transformed to the persistence image Img $(d_{\mathbb{X}_n}, h)$ for $h = 0.1$. Note that Dgm $\left( f_{\rho,\sigma}^n \right)$ is a robust diagram while Img $(d_{\mathbb{X}_n}, h)$ is a stable vectorization of a non-robust diagram [1]. For each homological order $\{H_0, H_1, H_2\}$, distance

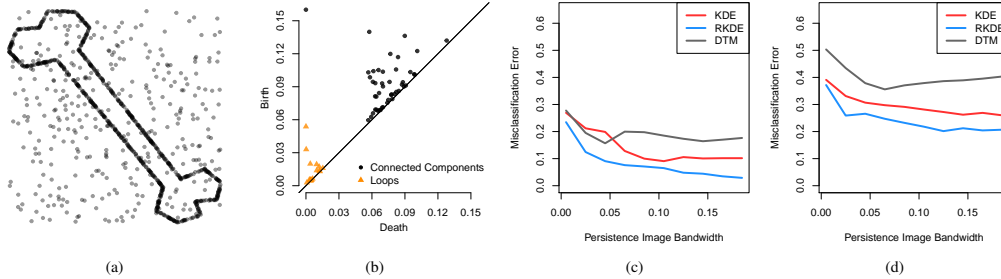

Figure 5: (a) $\mathbb{X}_n$ is sampled from the image boundary of a *bone*, and uniform noise is added. (b) The resulting persistence diagram from the robust KDE. The persistence diagram picks up the $1^{st}$–order features near the joints of the cartoon bone. The misclassification error for the KDE, robust KDE and DTM as the persistence image bandwidth increases, (c) for the three-class classification and, (d) for the five-class classification.

matrices $\{\Delta_0, \Delta_1, \Delta_2\}$ are computed: $W_p$ metric for $\mathsf{Dgm}\,(f_{\rho,\sigma})$, and $L_p$ metric for $\mathsf{Img}\,(d_{\mathbb{X}_n}, h)$ with $p \in \{1, 2, \infty\}$, and spectral clustering is performed on the resulting distance-matrices. The quality of the clustering is assessed using the rand-index. The results, reported in Table 2, evidence the superiority of employing inherently robust persistence diagrams in contrast to a robust vectorization of an inherently noisy persistence diagram.

Table 2: Rand-index for spectral clustering using distance matrices for $\mathsf{Dgm}\,(f_{\rho,\sigma})$ and $\mathsf{Img}\,(d_{\mathbb{X}_n}, h)$.

|  | $\mathsf{Dgm}\,(f_{\rho,\sigma})$ | | | $\mathsf{Img}\,(d_{\mathbb{X}_n}, h)$ | | |
|---|---|---|---|---|---|---|
| Distance Metric | $W_1$ | $W_2$ | $W_\infty$ | $L_1$ | $L_2$ | $L_\infty$ |
| $\Delta_0$ (from $H_0$) | 95.30% | 93.65% | 94.44% | 78.53% | 81.77% | 80.05% |
| $\Delta_1$ (from $H_1$) | 91.43% | 88.56% | 84.53% | 81.89% | 81.14% | 77.75% |
| $\Delta_2$ (from $H_2$) | 86.33% | 73.91% | 73.62% | 80.09% | 77.12% | 77.35% |
| $\Delta_{\max} = \max\{\Delta_0, \Delta_1, \Delta_2\}$ | 95.72% | 93.65% | 94.44% | 82.43% | 78.80% | 79.78% |

**MPEG7.** In this experiment, we examine the performance of persistence diagrams in a classification task on [28]. For simplicity, we only consider five classes: *beetle*, *bone*, *spring*, *deer* and *horse*. We first extract the boundary of the images using a Laplace convolution, and sample $\mathbb{X}_n$ uniformly from the boundary of each image, adding uniform noise ($\pi = 15\%$) in the enclosing region. Persistence diagrams $\mathsf{Dgm}\,(f_\sigma^n)$ and $\mathsf{Dgm}\,(f_{\rho,\sigma}^n)$ from the KDE and robust KDE are constructed. In addition, owing to their ability to capture nuanced multi-scale features, we also construct $\mathsf{Dgm}\,(d_{n,m})$ from the DTM filtration. The smoothing parameters $\sigma(k)$ and $m(k)$ are chosen as earlier for $k = 5$. The persistence diagrams are normalized to have a max persistence $\max\{|d - b| = 1 : (b, d) \in \mathsf{Dgm}(\phi)\}$, and then vectorized as persistence images, $\mathsf{Img}\,(f_\sigma^n, h)$, $\mathsf{Img}\,(f_{\rho,\sigma}^n, h)$, and $\mathsf{Img}\,(d_{n,m}, h)$ for various bandwidths $h$. A linear SVM classifier is then trained on the resulting persistence images. In the first experiment we only consider the first three classes, and in the second experiment we consider all five classes. The results for the classification error, shown in Figure 5, demonstrate the superiority of the proposed method. We refer the reader to Appendix D for additional experiments.

# 6   Conclusion & Discussion

In this paper, we proposed a statistically consistent robust persistent diagram using RKHS-based robust KDE as the filter function. By generalizing the notion of influence function to the space of persistence diagrams, we mathematically and empirically demonstrated the robustness of the proposed method to that of persistence diagrams induced by other filter functions such as KDE. Through numerical experiments, we demonstrated the advantage of using robust persistence diagrams in machine learning applications. We would like to highlight that most of the theoretical results of this paper crucially hinge on the loss function being convex. As a future direction, we would like to generalize the current results to non-convex loss functions, and explore robust persistence diagrams induced other types of robust density estimators, which could potentially yield more robust persistence diagrams. Another important direction we intend to explore is to enhance the computational efficiency of the proposed approach using coresets, as in [7], and/or using weighted Rips filtrations, as in [2]. We provide a brief discussion in Appendix E.

## Broader Impact

Over the last decade, Topological Data Analysis has become an important tool for extracting geometric and topological information from data, and its applications have been far reaching. For example, it has been used successfully in the study the fragile X-syndrome, to discover traumatic brain injuries, and has also become an important tool in the study of protein structure. In astrophysics, it has aided the study of cosmic microwave background, and the discovery of cosmic voids and filamental structures in cosmological data. With a continual increase in its adoption in data analysis, it has become important to understand the limitations of using persistent homology in machine learning applications. As real-world data is often flustered with measurement errors and other forms of noise, in this work, we examine the sensitivity of persistence diagrams to such noise, and provide methods to mitigate the effect of this noise, so as to make reliable topological inference.

## Acknowledgments and Disclosure of Funding

The authors would like to thank the anonymous reviewers for their helpful comments and constructive feedback. Siddharth Vishwanath and Bharath Sriperumbudur are supported in part by NSF DMS CAREER Award 1945396. Kenji Fukumizu is supported in part by JST CREST Grant Number JPMJCR15D3, Japan. Satoshi Kuriki is partially supported by JSPS KAKENHI Grant Number JP16H02792, Japan.

## Footnotes

[1] https://github.com/sidv23/robust-PDs

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
