[Supplementary Material]

# Supplementary Material:
## Robust Persistence Diagrams using Reproducing Kernels

## A  Proofs for Section 4

In what follows, given a metric space $(\mathcal{X}, \varrho)$, $L_p(\mathcal{X}, \mu)$ is the Banach space of functions of $p^{th}$-power $\mu$-integrable functions with norm $\|\cdot\|_p$, where $\mu$ is a Borel measure defined on $\mathcal{X}$. For a fixed loss $\rho$, we will use the notation $\ell_g(\cdot) = \ell(\cdot, g) = \rho\left(\|\Phi(\cdot) - g\|_{\mathcal{H}_\sigma}\right)$ in order to emphasize the dependency of the loss on the choice of $g \in \mathcal{G}$. Borrowing some notation from empirical process theory, we define the empirical risk-functional in Eq. (1) as

$$\mathcal{J}_n(g) \doteq \mathbb{P}_n \ell_g = \sum_{i=1}^n \rho\left(\|\Phi_\sigma(\boldsymbol{X}_i) - g\|_{\mathcal{H}_\sigma}\right),$$

and, similarly, the population risk functional $\mathcal{J}(g)$ is given by

$$\mathcal{J}(g) \doteq \mathbb{P}\ell_g = \int_{\mathbb{R}^d} \rho\left(\|\Phi_\sigma(\boldsymbol{x}) - g\|_{\mathcal{H}_\sigma}\right) d\mathbb{P}(\boldsymbol{x}).$$

### A.1  Proof of Theorem 4.2

For $\epsilon > 0$, define the risk functional associated with $\mathbb{P}_{\boldsymbol{x}}^\epsilon$ to be

$$\mathcal{J}_{\epsilon, \boldsymbol{x}}(g) = \mathbb{P}_{\boldsymbol{x}}^\epsilon \ell_g = (1 - \epsilon)\mathcal{J}(g) + \epsilon\rho\left(\|\Phi_\sigma(\boldsymbol{x}) - g\|_{\mathcal{H}_\sigma}\right),$$

and let $f_{\rho,\sigma}^{\epsilon;\boldsymbol{x}} = \arg\inf_{g\in\mathcal{G}} \mathcal{J}_{\epsilon,\boldsymbol{x}}(g)$ be its minimizer. From the stability result of Proposition 2.2 we have that

$$\Psi\left(f_{\rho,\sigma}; \boldsymbol{x}\right) = \lim_{\epsilon\to 0}\frac{1}{\epsilon}W_\infty\left(\mathsf{Dgm}\left(f_{\rho,\sigma}^{\epsilon;\boldsymbol{x}}\right), \mathsf{Dgm}\left(f_{\rho,\sigma}\right)\right) \leq \lim_{\epsilon\to 0}\frac{1}{\epsilon}\left\|f_{\rho,\sigma}^{\epsilon;\boldsymbol{x}} - f_{\rho,\sigma}\right\|_\infty.$$

Using Propositions B.3 and B.5, we know that the sequence $\{\mathcal{J}_{\epsilon,\boldsymbol{x}}\}$ is equi-coercive, and $\mathcal{J}_{\epsilon,\boldsymbol{x}}$ $\Gamma$–converges to $\mathcal{J}$ as $\epsilon \to 0$. From the fundamental theorem of $\Gamma$–convergence [17, Theorem 7.8] we have that $\left\|f_{\rho,\sigma}^{\epsilon,\boldsymbol{x}} - f_{\rho,\sigma}\right\|_{\mathcal{H}_\sigma} \to 0$, and, consequently, from Lemma B.6, $\left\|f_{\rho,\sigma}^{\epsilon,\boldsymbol{x}} - f_{\rho,\sigma}\right\|_\infty \to 0$ as $\epsilon \to 0$. Thus,

$$\lim_{\epsilon\to 0}\frac{1}{\epsilon}\left\|f_{\rho,\sigma}^{\epsilon,\boldsymbol{x}} - f_{\rho,\sigma}\right\|_\infty = \left\|\lim_{\epsilon\to 0}\frac{f_{\rho,\sigma}^{\epsilon,\boldsymbol{x}} - f_{\rho,\sigma}}{\epsilon}\right\|_\infty. \tag{A.1}$$

Let the limit in the right hand side of Eq. (A.1) be denoted by $\dot{f}_{\rho,\sigma}$. Although $\dot{f}_{\rho,\sigma}$ does not admit a closed-form solution, from [27, Theorem 8] we have that $\dot{f}_{\rho,\sigma}$ satisfies $V = a\dot{f}_{\rho,\sigma} + B$, where

$$V = \varphi\left(\|\Phi_\sigma(\boldsymbol{x}) - f_{\rho,\sigma}\|_{\mathcal{H}_\sigma}\right) \cdot (\Phi_\sigma(\boldsymbol{x}) - f_{\rho,\sigma}),$$

$$a = \int_{\mathbb{R}^d} \varphi\left(\|\Phi_\sigma(\boldsymbol{y}) - f_{\rho,\sigma}\|_{\mathcal{H}_\sigma}\right) d\mathbb{P}(\boldsymbol{y}), \quad \text{and}$$

$$B = \int_{\mathbb{R}^d}\left(\frac{\varphi'\left(\|\Phi_\sigma(\boldsymbol{y}) - f_{\rho,\sigma}\|_{\mathcal{H}_\sigma}\right)}{\|\Phi_\sigma(\boldsymbol{y}) - f_{\rho,\sigma}\|_{\mathcal{H}_\sigma}}\left\langle\dot{f}_{\rho,\sigma}, \Phi_\sigma(\boldsymbol{y}) - f_{\rho,\sigma}\right\rangle_{\mathcal{H}_\sigma} \cdot (\Phi_\sigma(\boldsymbol{y}) - f_{\rho,\sigma})\right) d\mathbb{P}(\boldsymbol{y}).$$

For brevity, we adopt the notation $z(\boldsymbol{y}) = \|\Phi_\sigma(\boldsymbol{y}) - f_{\rho,\sigma}\|_{\mathcal{H}_\sigma}$ and $u(\cdot, \boldsymbol{y}) = \frac{\Phi_\sigma(\boldsymbol{y}) - f_{\rho,\sigma}}{\|\Phi_\sigma(\boldsymbol{y}) - f_{\rho,\sigma}\|_{\mathcal{H}_\sigma}} \in \mathcal{H}_\sigma$. Then note that $a \in \mathbb{R}$ and $B \in \mathcal{H}_\sigma$ are given by

$$a = \int_{\mathbb{R}^d} \varphi\left(z(\boldsymbol{y})\right) d\mathbb{P}(\boldsymbol{y}), \quad \text{and}$$

$$B = \int_{\mathbb{R}^d} z(\boldsymbol{y})\varphi'\left(z(\boldsymbol{y})\right)\left\langle\dot{f}_{\rho,\sigma}, u(\cdot, \boldsymbol{y})\right\rangle_{\mathcal{H}_\sigma} u(\cdot, \boldsymbol{y}) d\mathbb{P}(\boldsymbol{y}).$$

Using the reverse triangle inequality we have

$$\|V\|_{\mathcal{H}_\sigma} \geq a \left\|\dot{f}_{\rho,\sigma}\right\|_{\mathcal{H}_\sigma} - \|B\|_{\mathcal{H}_\sigma}. \tag{A.2}$$

We now look for an upper bound on $\|B\|_{\mathcal{H}_\sigma}$. By noting that

$$\left\langle \dot{f}_{\rho,\sigma}, u(\cdot,\boldsymbol{x})\right\rangle_{\mathcal{H}_\sigma} \left\langle \dot{f}_{\rho,\sigma}, u(\cdot,\boldsymbol{y})\right\rangle_{\mathcal{H}_\sigma} \left\langle u(\cdot,\boldsymbol{x}), u(\cdot,\boldsymbol{y})\right\rangle_{\mathcal{H}_\sigma} \leq \left\|\dot{f}_{\rho,\sigma}\right\|_{\mathcal{H}_\sigma}^2,$$

we have

$$\|B\|_{\mathcal{H}_\sigma}^2 = \left\langle B, B\right\rangle_{\mathcal{H}_\sigma} \leq \iint z(\boldsymbol{x})\varphi'(z(\boldsymbol{x}))z(\boldsymbol{y})\varphi'(z(\boldsymbol{y})) \left\|\dot{f}_{\rho,\sigma}\right\|_{\mathcal{H}_\sigma}^2 d\mathbb{P}(\boldsymbol{x})\, d\mathbb{P}(\boldsymbol{y})$$

$$= \left\|\dot{f}_{\rho,\sigma}\right\|_{\mathcal{H}_\sigma}^2 \left(\int_{\mathbb{R}^d} z(\boldsymbol{y})\varphi'(z(\boldsymbol{y}))\, d\mathbb{P}(\boldsymbol{y})\right)^2.$$

Plugging this back into Eq. (A.2) we get

$$\|V\|_{\mathcal{H}_\sigma} \geq \left\|\dot{f}_{\rho,\sigma}\right\|_{\mathcal{H}_\sigma} \int_{\mathbb{R}^d} \varphi(z(\boldsymbol{y})) - z(\boldsymbol{y})\varphi'(z(\boldsymbol{y}))\, d\mathbb{P}(\boldsymbol{y})$$

$$= \left\|\dot{f}_{\rho,\sigma}\right\|_{\mathcal{H}_\sigma} \int_{\mathbb{R}^d} \zeta(z(\boldsymbol{y}))\, d\mathbb{P}(\boldsymbol{y}), \tag{A.3}$$

where $\zeta(z) = \varphi(z) - z\varphi'(z)$. Similarly, by using the definition of $\varphi$, it follows that

$$\|V\|_{\mathcal{H}_\sigma} = \varphi\left(\|\Phi_\sigma(\boldsymbol{x}) - f_{\rho,\sigma}\|_{\mathcal{H}_\sigma}\right) \cdot \|\Phi_\sigma(\boldsymbol{x}) - f_{\rho,\sigma}\|_{\mathcal{H}_\sigma} = \rho'\left(\|\Phi_\sigma(\boldsymbol{x}) - f_{\rho,\sigma}\|_{\mathcal{H}_\sigma}\right).$$

Combining this with Eq. (A.3) we get

$$\left\|\dot{f}_{\rho,\sigma}\right\|_{\mathcal{H}_\sigma} \leq \frac{\rho'\left(\|\Phi_\sigma(\boldsymbol{x}) - f_{\rho,\sigma}\|_{\mathcal{H}_\sigma}\right)}{\int_{\mathbb{R}^d} \zeta\left(\|\Phi_\sigma(\boldsymbol{y}) - f_{\rho,\sigma}\|_{\mathcal{H}_\sigma}\right) d\mathbb{P}(\boldsymbol{y})}.$$

By noting that $\left\|\dot{f}_{\rho,\sigma}\right\|_\infty \leq \|K_\sigma\|_\infty^{\frac{1}{2}} \left\|\dot{f}_{\rho,\sigma}\right\|_{\mathcal{H}_\sigma}$ and $\Psi(f_{\rho,\sigma};\boldsymbol{x}) \leq \left\|\dot{f}_{\rho,\sigma}\right\|_\infty$, the result follows. ∎

## A.2  Proof for Theorem 4.4

Using the triangle inequality we can break our problem down as follows

$$\left\|f_{\rho,\sigma}^n - f\right\|_\infty \leq \underbrace{\|f_\sigma - f\|_\infty}_{\text{(a)}} + \underbrace{\|f_{\rho,\sigma} - f_\sigma\|_\infty}_{\text{(b)}},$$

where, $f_\sigma = \int_{\mathbb{R}^d} K_\sigma(\cdot,\boldsymbol{x}) d\mathbb{P}(\boldsymbol{x})$ is the population level KDE. For term (a), for $\mathbb{P} \in \mathcal{M}(\mathbb{R}^d)$, it is well known [15] that the approximation error for the KDE vanishes, i.e.,

$$\|f_\sigma - f\|_\infty \to 0,$$

as $\sigma \to 0$. So, it remains to verify that (b) vanishes, i.e., $\|f_{\rho,\sigma} - f_\sigma\|_\infty \to 0$. With this in mind, consider the map $T_\sigma : \mathcal{G} \to \mathcal{G}$ given by

$$T_\sigma(g) = \int_{\mathbb{R}^d} \frac{\varphi\left(\|\Phi_\sigma(\boldsymbol{x}) - g\|_{\mathcal{H}_\sigma}\right)}{\int_{\mathbb{R}^d} \varphi\left(\|\Phi_\sigma(\boldsymbol{x}) - g\|_{\mathcal{H}_\sigma}\right) d\mathbb{P}(\boldsymbol{x})} \Phi_\sigma(\boldsymbol{x}) d\mathbb{P}(\boldsymbol{x}).$$

Our approach to verifying that (b) vanishes is similar to Vandermeulen and Scott [39, Lemma 9], where we show that the map $T_\sigma$ is a contraction map when restricted to the subspace

$$\mathcal{Q}_\sigma \doteq B_{\mathcal{H}_\sigma}(\mathbf{0}, \delta\nu_\sigma) \cap \mathcal{D}_\sigma.$$

A key difference is that we work with $\|\cdot\|_\infty$–norm, requiring us to obtain a sharper bound for the Lipschitz constant associated with the contraction.

For brevity, we adopt the notation $m(\boldsymbol{x}, g) = \varphi\left(\|\Phi_\sigma(\boldsymbol{x}) - g\|_{\mathcal{H}_\sigma}\right)$. The authors in [27] show that $f_{\rho,\sigma}$ is a fixed point of the map $T_\sigma$, i.e., $T_\sigma(f_{\rho,\sigma}) = f_{\rho,\sigma}$, and that $f_\sigma$ is the image of $\mathbf{0}$ under $T_\sigma$, i.e., $T_\sigma(\mathbf{0}) = f_\sigma$. Additionally, from Lemma B.7, we know that $\|f_\sigma\|_{\mathcal{H}_\sigma} \leq \delta\nu_\sigma$, for some $0 < \delta < 1$. Thus, we can rewrite $f_{\rho,\sigma} - f_\sigma = T_\sigma(f_{\rho,\sigma}) - T_\sigma(\mathbf{0})$.

Let $g, h \in \mathcal{Q}_\sigma$. Then we have that

$$
\begin{aligned}
T_\sigma(g) - T_\sigma(h) &= \int_{\mathbb{R}^d} \frac{m(\boldsymbol{x}, g)}{\int_{\mathbb{R}^d} m(\boldsymbol{y}, g)d\mathbb{P}(\boldsymbol{y})}\Phi_\sigma(\boldsymbol{x})d\mathbb{P}(\boldsymbol{x}) - \int_{\mathbb{R}^d} \frac{m(\boldsymbol{u}, h)}{\int_{\mathbb{R}^d} m(\boldsymbol{v}, h)d\mathbb{P}(\boldsymbol{v})}\Phi_\sigma(\boldsymbol{u})d\mathbb{P}(\boldsymbol{u}) \\
&= \frac{1}{\alpha\beta} \cdot \left(\beta \int_{\mathbb{R}^d} m(\boldsymbol{x}, g)\Phi_\sigma(\boldsymbol{x})d\mathbb{P}(\boldsymbol{x}) - \alpha \int_{\mathbb{R}^d} m(\boldsymbol{u}, h)\Phi_\sigma(\boldsymbol{u})d\mathbb{P}(\boldsymbol{x})\right) \\
&= \frac{1}{\alpha\beta} \cdot \xi,
\end{aligned}
\tag{A.4}
$$

where $\alpha \doteq \int_{\mathbb{R}^d} m(\boldsymbol{y}, g)d\mathbb{P}(\boldsymbol{y}) \in \mathbb{R}$, $\beta \doteq \int_{\mathbb{R}^d} m(\boldsymbol{v}, h)d\mathbb{P}(\boldsymbol{v}) \in \mathbb{R}$ and the numerator $\xi \in \mathcal{H}_\sigma$.

By Tonelli's theorem

$$
\begin{aligned}
\xi &= \beta \int_{\mathbb{R}^d} m(\boldsymbol{x}, g)\Phi_\sigma(\boldsymbol{x})d\mathbb{P}(\boldsymbol{x}) - \alpha \int_{\mathbb{R}^d} m(\boldsymbol{u}, h)\Phi_\sigma(\boldsymbol{u})d\mathbb{P}(\boldsymbol{x}) \\
&= \int_{\mathbb{R}^d} m(\boldsymbol{x}, g)\Phi_\sigma(\boldsymbol{x}) \left(\int_{\mathbb{R}^d} m(\boldsymbol{v}, h)d\mathbb{P}(\boldsymbol{v})\right) d\mathbb{P}(\boldsymbol{x}) \\
&\qquad - \int_{\mathbb{R}^d} m(\boldsymbol{u}, h)\Phi_\sigma(\boldsymbol{u}) \left(\int_{\mathbb{R}^d} m(\boldsymbol{y}, g)d\mathbb{P}(\boldsymbol{y})\right) d\mathbb{P}(\boldsymbol{x}) \\
&= \iint_{\mathbb{R}^d \times \mathbb{R}^d} m(\boldsymbol{x}, g)m(\boldsymbol{v}, h)\Phi_\sigma(\boldsymbol{x})d\mathbb{P}(\boldsymbol{v})d\mathbb{P}(\boldsymbol{x}) - \iint_{\mathbb{R}^d \times \mathbb{R}^d} m(\boldsymbol{u}, h)m(\boldsymbol{y}, g)\Phi_\sigma(\boldsymbol{u})d\mathbb{P}(\boldsymbol{y})d\mathbb{P}(\boldsymbol{u}) \\
&= \iint_{\mathbb{R}^d \times \mathbb{R}^d} \Phi_\sigma(\boldsymbol{x}) \left[m(\boldsymbol{x}, g)m(\boldsymbol{y}, h) - m(\boldsymbol{x}, h)m(\boldsymbol{y}, g)\right] d\mathbb{P}(\boldsymbol{x})d\mathbb{P}(\boldsymbol{y}).
\end{aligned}
\tag{A.5}
$$

Then by adding and subtracting $m(\boldsymbol{x}, h)m(\boldsymbol{y}, h)$ to the term inside, we get

$$
\begin{aligned}
m(\boldsymbol{x}, g)m(\boldsymbol{y}, h) - m(\boldsymbol{x}, h)m(\boldsymbol{y}, g) &= m(\boldsymbol{y}, h)\left\{m(\boldsymbol{x}, g) - m(\boldsymbol{x}, h)\right\} \\
&\qquad + m(\boldsymbol{x}, h)\left\{m(\boldsymbol{y}, h) - m(\boldsymbol{y}, g)\right\}.
\end{aligned}
$$

Plugging this back into Eq. (A.5), we get $\xi = \xi_1 + \xi_2$ where

$$
\begin{aligned}
\xi_1 &= \iint_{\mathbb{R}^d \times \mathbb{R}^d} \Phi_\sigma(\boldsymbol{x})\left\{m(\boldsymbol{x}, g) - m(\boldsymbol{x}, h)\right\} m(\boldsymbol{y}, h)d\mathbb{P}(\boldsymbol{y})d\mathbb{P}(\boldsymbol{x}) \\
&= \int_{\mathbb{R}^d} m(\boldsymbol{y}, h)d\mathbb{P}(\boldsymbol{y}) \int_{\mathbb{R}^d} \Phi_\sigma(\boldsymbol{x})\left\{m(\boldsymbol{x}, g) - m(\boldsymbol{x}, h)\right\} d\mathbb{P}(\boldsymbol{x}) \\
&= \beta \int_{\mathbb{R}^d} K_\sigma(\cdot, \boldsymbol{x})\left\{m(\boldsymbol{x}, g) - m(\boldsymbol{x}, h)\right\} d\mathbb{P}(\boldsymbol{x}) \\
&\stackrel{(i)}{=} \beta \cdot \left[\psi_\sigma * \left((m(\cdot, g) - m(\cdot, h))\, f(\cdot)\right)\right],
\end{aligned}
$$

where (i) follows from the fact that the kernel $K_\sigma(\boldsymbol{x}, \boldsymbol{y}) = \psi_\sigma(\boldsymbol{x} - \boldsymbol{y}) \doteq \sigma^{-d}\psi(\|\boldsymbol{x} - \boldsymbol{y}\|_2/\sigma)$ is translation invariant and $f$ is the density associated with $\mathbb{P}$.

Similarly,

$$\xi_2 = \iint\limits_{\mathbb{R}^d \times \mathbb{R}^d} \Phi_\sigma(\boldsymbol{x}) m(\boldsymbol{x}, h) \left\{ m(\boldsymbol{y}, h) - m(\boldsymbol{y}, g) \right\} d\mathbb{P}(\boldsymbol{x}) d\mathbb{P}(\boldsymbol{y})$$

$$= \int\limits_{\mathbb{R}^d} \left[ m(\boldsymbol{y}, h) - m(\boldsymbol{y}, g) \right] d\mathbb{P}(\boldsymbol{y}) \int\limits_{\mathbb{R}^d} \Phi_\sigma(\boldsymbol{x}) m(\boldsymbol{x}, h) d\mathbb{P}(\boldsymbol{x})$$

$$\leq \left\| m(\cdot, h) - m(\cdot, g) \right\|_\infty \cdot \left[ \psi_\sigma * (m(\cdot, h) f(\cdot)) \right].$$

The upper bound for $\|\xi_1\|_\infty$ is as follows

$$\|\xi_1\|_\infty = \beta \left\| \psi_\sigma * ((m(\cdot, g) - m(\cdot, h)) f(\cdot)) \right\|_\infty$$

$$\overset{(i)}{\leq} \beta \left\| \psi_\sigma \right\|_1 \left\| (m(\cdot, g) - m(\cdot, h)) f(\cdot) \right\|_\infty$$

$$\overset{(ii)}{\leq} \beta \left\| m(\cdot, g) - m(\cdot, h) \right\|_\infty \left\| f \right\|_\infty , \tag{A.6}$$

where (i) follows from Young's inequality [25, Theorem 20.18] and (ii) follows from the fact that $\|fg\|_\infty \leq \|f\|_\infty \|g\|_\infty$. Similarly, for $\xi_2$ we have

$$\|\xi_2\|_\infty \leq \left\| m(\cdot, h) - m(\cdot, g) \right\|_\infty \left\| \psi_\sigma * (m(\cdot, h) f(\cdot)) \right\|_\infty$$

$$\overset{(i)}{\leq} \left\| m(\cdot, h) - m(\cdot, g) \right\|_\infty \left\| \psi_\sigma \right\|_1 \left\| m(\cdot, h) f(\cdot) \right\|_\infty$$

$$\overset{(ii)}{\leq} \left\| m(\cdot, h) - m(\cdot, g) \right\|_\infty \left\| m(\cdot, h) \right\|_\infty \left\| f \right\|_\infty . \tag{A.7}$$

From the proof of [39, Lemma 9, Page 20–22], for $g, h \in \mathcal{Q}_\sigma$ for fixed constants $c_1, c_2 > 0$ we have the following two bounds:

$$\alpha, \beta \geq \frac{1}{c_1 \nu_\sigma}, \tag{A.8}$$

and

$$\left\| m(\cdot, h) - m(\cdot, g) \right\|_\infty \leq \left\| g - h \right\|_{\mathcal{H}_\sigma} c_2 \nu_\sigma^{-2}, \tag{A.9}$$

where the last inequality follows from the Lipschitz property of $\varphi$ and fact that $\rho$ is strictly convex. Additionally, for $c_3 = \|\rho'\|_\infty < \infty$ we have

$$m(\boldsymbol{x}, g) = \varphi \left( \left\| \Phi_\sigma(\boldsymbol{x}) - g \right\|_{\mathcal{H}_\sigma} \right)$$

$$= \frac{\rho' \left( \left\| \Phi_\sigma(\boldsymbol{x}) - g \right\|_{\mathcal{H}_\sigma} \right)}{\left\| \Phi_\sigma(\boldsymbol{x}) - g \right\|_{\mathcal{H}_\sigma}}$$

$$\leq \frac{c_3}{\left\| \Phi_\sigma(\boldsymbol{x}) - g \right\|_{\mathcal{H}_\sigma}}$$

$$\overset{(iii)}{\leq} \frac{c_3}{\left| \left\| \Phi_\sigma(\boldsymbol{x}) \right\|_{\mathcal{H}_\sigma} - \left\| g \right\|_{\mathcal{H}_\sigma} \right|}$$

$$= \frac{c_3}{(1 - \delta)\nu_\sigma}, \tag{A.10}$$

where (iii) follows from reverse triangle inequality. Plugging the bounds in equations (A.8), (A.9) and (A.10) back into equations (A.6) and (A.7) we get,

$$\|\xi_1\|_\infty + \|\xi_2\|_\infty \leq \|f\|_\infty \left( \beta c_2 \nu_\sigma^{-2} \|g - h\|_{\mathcal{H}_\sigma} + \frac{c_2 c_3}{(1 - \delta)} \nu_\sigma^{-3} \|g - h\|_{\mathcal{H}_\sigma} \right).$$

Using this upper bound in Eq. (A.4) we get

$$
\begin{aligned}
\|T_\sigma(g) - T_\sigma(h)\|_\infty &= \left\| \frac{\xi}{\alpha\beta} \right\|_\infty \\
&\leq \frac{\|\xi_1\|_\infty + \|\xi_2\|_\infty}{\alpha\beta} \\
&\overset{(iv)}{\leq} \|f\|_\infty \left( \frac{c_1 c_2}{c_1} \nu_\sigma^{-1} \|g - h\|_{\mathcal{H}_\sigma} + \frac{c_2 c_3}{c_1^2(1-\delta)} \nu_\sigma^{-1} \|g - h\|_{\mathcal{H}_\sigma} \right) \\
&\overset{(v)}{=} C \nu_\sigma^{-1} \|g - h\|_{\mathcal{H}_\sigma} \\
&\overset{(vi)}{\leq} C \nu_\sigma^{-1} \|g - h\|_\infty^{\frac{1}{2}},
\end{aligned}
$$

where in (iv) we use Eq. (A.8), in (v) we use the fact that whenever $\mathbb{P} \in \mathcal{M}(\mathbb{R}^d)$, we have $\|f\|_\infty < \infty$ and $C > 0$ is a constant depending only on $c_1, c_2, c_3$ and $\|f\|_\infty$. Additionally, (vi) holds through an application of Lemma B.6 to $g - h \in \mathcal{Q}_\sigma \subset \mathcal{D}_\sigma$. This confirms that $T_\sigma$ is a contraction mapping. We use this to show that (b) vanishes as $\sigma \to 0$. Since $f_{\rho,\sigma}, \mathbf{0} \in \mathcal{Q}_\sigma$ and $f_{\rho,\sigma} - \mathbf{0} \in \mathcal{D}_\sigma$, we have that

$$
\begin{aligned}
\|f_{\rho,\sigma} - f_\sigma\|_\infty &= \|T_\sigma(f_{\rho,\sigma}) - T_\sigma(\mathbf{0})\|_\infty \\
&\leq C \nu_\sigma^{-1} \|f_{\rho,\sigma} - \mathbf{0}\|_\infty^{\frac{1}{2}} \\
&= C \nu_\sigma^{-1} \|f_{\rho,\sigma}\|_\infty^{\frac{1}{2}}.
\end{aligned}
$$

Using the triangle inequality $\|f_{\rho,\sigma}\|_\infty^{\frac{1}{2}} \leq \|f_{\rho,\sigma} - f_\sigma\|_\infty^{\frac{1}{2}} + \|f_\sigma\|_\infty^{\frac{1}{2}}$ we get

$$
\begin{aligned}
\|f_{\rho,\sigma} - f_\sigma\|_\infty &\leq C \nu_\sigma^{-1} \left( \|f_{\rho,\sigma} - f_\sigma\|_\infty^{\frac{1}{2}} + \|f_\sigma\|_\infty^{\frac{1}{2}} \right) \\
&= C \nu_\sigma^{-1} \left( \|T_\sigma(f_{\rho,\sigma}) - T_\sigma(\mathbf{0})\|_\infty^{\frac{1}{2}} + \|f_\sigma\|_\infty^{\frac{1}{2}} \right) \\
&\leq C \nu_\sigma^{-1} \left( \left( C \nu_\sigma^{-1} \|f_{\rho,\sigma} - \mathbf{0}\|_\infty^{\frac{1}{2}} \right)^{\frac{1}{2}} + \|f_\sigma\|_\infty^{\frac{1}{2}} \right) \\
&= C^{\frac{3}{2}} \nu_\sigma^{-\frac{3}{2}} \|f_{\rho,\sigma}\|_\infty^{\frac{1}{4}} + C \nu_\sigma^{-1} \|f_\sigma\|_\infty^{\frac{1}{2}},
\end{aligned}
\tag{A.11}
$$

by using the contraction mapping twice. Observe that

$$
\|f_{\rho,\sigma}\|_\infty \leq \nu_\sigma \|f_{\rho,\sigma}\|_{\mathcal{H}_\sigma} \leq \delta \nu_\sigma^2,
$$

where the first inequality follows from Lemma B.6 and the second inequality follows from the fact that $\|f_{\rho,\sigma}\|_{\mathcal{H}_\sigma} \leq \delta \nu_\sigma$ since $f_{\rho,\sigma} \in \mathcal{Q}_\sigma$. Furthermore, $\|f_\sigma\|_\infty = \|\psi_\sigma * f\|_\infty \leq \|\psi_\sigma\|_\infty \|f\|_1 \leq \nu_\sigma$ from Young's inequality. By noting that $\nu_\sigma = \psi_\sigma(0) = \sigma^{-d}\psi(0)$, collecting these bounds back into Eq. (A.11) we get

$$
\|f_{\rho,\sigma} - f_\sigma\|_\infty \leq C^{\frac{3}{2}} \delta^{\frac{1}{4}} \nu_\sigma^{-1} + C \nu_\sigma^{-\frac{1}{2}} \sqrt{\psi(0)}.
$$

yielding that $\|f_{\rho,\sigma} - f_\sigma\|_\infty \to 0$ as $\sigma \to 0$, thereby verifying that (b) vanishes as $\sigma \to 0$. ∎

### A.3 Proof of Theorem 4.5

The proof proceeds in two steps: We first establish the uniform consistency for the robust KDE and then use the bottleneck stability to show consistency of the robust persistence diagrams in $W_\infty$. From the stability theorem for persistence diagrams [12, 16], we have that $W_\infty \left( \mathsf{Dgm} \left( f_{\rho,\sigma}^n \right), \mathsf{Dgm} \left( f_{\rho,\sigma} \right) \right) \leq \left\| f_{\rho,\sigma}^n - f_{\rho,\sigma} \right\|_\infty$. Thus, it suffices to show that $\left\| f_{\rho,\sigma}^n - f_{\rho,\sigma} \right\|_\infty \overset{p}{\to} 0$ as $n \to \infty$. In order to prove the latter, we adapt the argmax consistency theorem [38, Theorem 5.7] for minimizers of a risk function.

**Lemma A.1** (Theorem 5.7, [38]). *Given a metric space $(\mathcal{G}, d)$, let $\mathcal{J}_n$ be random functions and $\mathcal{J}$ be a fixed function of $g \in \mathcal{G}$ such that for every $\epsilon > 0$,*

*(1)* $\displaystyle\inf_{g:d(g,g_0)\geq\epsilon} \mathcal{J}(g) > \mathcal{J}(g_0)$, *and*

*(2)* $\displaystyle\sup_{g\in\mathcal{G}} |\mathcal{J}_n(g) - \mathcal{J}(g)| \xrightarrow{p} 0.$

*Then any sequence $g_n$ satisfying $\mathcal{J}_n(g_n) < \mathcal{J}_n(g_0) + O_p(1)$ satisfies $d(g_n, g_0) \xrightarrow{p} 0$.*

For $\mathcal{G} = \mathcal{H}_\sigma \cap \mathcal{D}_\sigma$, and $d(f_{\rho,\sigma}^n, f_{\rho,\sigma}) = \left\| f_{\rho,\sigma}^n - f_{\rho,\sigma} \right\|_\infty$, in order to establish uniform consistency of the robust KDE, as per Lemma A.1, we need to verify that conditions (1) and (2) are satisfied.

Condition (1) follows from the strict convexity of $\mathcal{J}(g)$ in Proposition B.1. Specifically, [27] establish that assumptions $(\mathcal{A}1) - (\mathcal{A}3)$ guarantee the existence and uniqueness of $f_{\rho,\sigma} = \arg\inf_{g\in\mathcal{G}} \mathcal{J}(g)$. Then, for any $g \in \mathcal{G}$ such that $\|g - f_{\rho,\sigma}\|_{\mathcal{H}_\sigma} > \delta$, we have that $\mathcal{J}(g) > \mathcal{J}(f_{\rho,\sigma})$.

We now turn to verifying condition (2). Observe that $\sup_{g\in\mathcal{G}} |\mathcal{J}_n(g) - \mathcal{J}(g)|$ can be rewritten as the supremum of an empirical process, i.e.,

$$\sup_{g\in\mathcal{G}} |\mathcal{J}_n(g) - \mathcal{J}(g)| = \sup_{\ell_g\in\widetilde{\mathcal{F}}} |\mathbb{P}_n\ell_g - \mathbb{P}\ell_g| \doteq \|\mathbb{P}_n - \mathbb{P}\|_{\widetilde{\mathcal{F}}},$$

where $\widetilde{\mathcal{F}} = \{\ell_g : g \in \mathcal{G}\}$, and $\ell_g(\boldsymbol{x}) = \rho\left(\|\Phi_\sigma(\boldsymbol{x}) - g\|_{\mathcal{H}_\sigma}\right)$. Verifying condition (2) reduces to showing that $\widetilde{\mathcal{F}}$ is a Glivenko-Cantelli class. Define $\eta(\cdot) = \|\Phi_\sigma(\cdot) - g\|_{\mathcal{H}_\sigma}^2$ and let $\mathcal{F} = \{\eta_g : g \in \mathcal{G}\}$. For the continuous map $\xi : [0,\infty) \to [0,\infty)$ given by $\xi(t) = \rho(\sqrt{t})$, we have that

$$\xi \circ \mathcal{F} = \{\xi(f) : f \in \mathcal{F}\} = \{\xi \circ \eta_g(\cdot) : g \in \mathcal{G}\} = \left\{\rho(\|\Phi_\sigma(\cdot) - g\|_{\mathcal{H}_\sigma}) : g \in \mathcal{G}\right\} = \widetilde{\mathcal{F}}.$$

By the preservation theorem for Glivenko-Cantelli classes [37, Theorem 3], it holds that if $\mathcal{F}$ is a Glivenko-Cantelli class, then $\widetilde{\mathcal{F}}$ is also a Glivenko-Cantelli class. So verifying condition (2) reduces to verifying that $\mathcal{F}$ is a Glivenko-Cantelli class. To this end, we first show that $F(\boldsymbol{x}_{1:n}) = F(\boldsymbol{x}_1, \boldsymbol{x}_2, \ldots, \boldsymbol{x}_n) = \sup_{g\in\mathcal{G}} |\mathbb{P}_n\eta_g - \mathbb{P}\eta_g| = \|\mathbb{P}_n - \mathbb{P}\|_{\mathcal{F}}$ satisfies the self-bounded property for McDiarmid's inequality, i.e.,

$$\sup_{\boldsymbol{x}_i\neq\boldsymbol{x}_i'} |F(\boldsymbol{x}_{1:n}) - F(\boldsymbol{x}_{1:n}')| \leq \frac{1}{n} \sup_{\boldsymbol{x}_i,\boldsymbol{x}_i'} \sup_{g\in\mathcal{G}} \left(\|\Phi_\sigma(\boldsymbol{x}_i)\|_{\mathcal{H}_\sigma}^2 + \|\Phi_\sigma(\boldsymbol{x}_i')\|_{\mathcal{H}_\sigma}^2 + 2\,|g(\boldsymbol{x}_i)| + 2\,|g(\boldsymbol{x}_i')|\right).$$

Observe that $\|\Phi_\sigma(\boldsymbol{x})\|_{\mathcal{H}_\sigma}^2 = K_\sigma(\boldsymbol{x}, \boldsymbol{x}) \leq \|K_\sigma\|_\infty$ and $|g(\boldsymbol{x})| \leq \|g\|_\infty < \|K_\sigma\|_\infty$ by Lemma B.6. Thus, we have that

$$\sup_{\boldsymbol{x}_i\neq\boldsymbol{x}_i'} |F(\boldsymbol{x}_{1:n}) - F(\boldsymbol{x}_{1:n}')| \leq \frac{6\,\|K_\sigma\|_\infty}{n}.$$

From [3, Theorem 9], we have that with probability greater than $1 - e^{-\delta}$,

$$\|\mathbb{P}_n - \mathbb{P}\|_{\mathcal{F}} \leq 2\mathfrak{R}_n(\mathcal{F}) + \sqrt{\frac{3\delta\,\|K_\sigma\|_\infty}{n}}, \tag{A.12}$$

where $\mathfrak{R}_n(\mathcal{F})$ is the Rademacher complexity of $\mathcal{F}$ given by,

$$\begin{aligned}
\mathfrak{R}_n(\mathcal{F}) &= \mathbb{E}_\epsilon\left(\sup_{g\in\mathcal{G}} \left|\frac{1}{n}\sum_{i=1}^n \epsilon_i \|\Phi_\sigma(\boldsymbol{x}_i) - g\|_{\mathcal{H}_\sigma}^2\right|\right)\\
&\leq \mathbb{E}_\epsilon\left(\sup_{g\in\mathcal{G}} \left\{\left|\frac{1}{n}\sum_{i=1}^n \epsilon_i \|\Phi_\sigma(\boldsymbol{x}_i)\|_{\mathcal{H}_\sigma}^2\right| + \left|\frac{1}{n}\sum_{i=1}^n \epsilon_i \|g\|_{\mathcal{H}_\sigma}^2\right| + 2\left|\frac{1}{n}\sum_{i=1}^n \epsilon_i g(\boldsymbol{x}_i)\right|\right\}\right)\\
&= \text{①} + \text{②} + \text{③}.
\end{aligned}$$

Note that $\mathbb{E}_\epsilon\left(f(\epsilon_{1:n}, \boldsymbol{x}_{1:n})\right) \doteq \mathbb{E}\left(f(\epsilon_{1:n}, \boldsymbol{x}_{1:n})|\boldsymbol{x}_{1:n}\right)$ is the conditional expectation of the Rademacher random variables $\epsilon_1, \epsilon_2, \ldots, \epsilon_n$, keeping $\boldsymbol{x}_1, \boldsymbol{x}_2, \ldots, \boldsymbol{x}_n$ fixed. First, we have that,

$$
\begin{aligned}
\textcircled{1} = \mathbb{E}_\epsilon\left(\sup_{g\in\mathcal{G}}\left|\frac{1}{n}\sum_{i=1}^{n}\epsilon_i\left\|\Phi_\sigma(\boldsymbol{x}_i)\right\|_{\mathcal{H}_\sigma}^2\right|\right) &\overset{(i)}{=} \mathbb{E}_\epsilon\left|\frac{1}{n}\sum_{i=1}^{n}\epsilon_i K_\sigma(\boldsymbol{x}_i, \boldsymbol{x}_i)\right| \\
&\overset{(ii)}{\leq} \sqrt{\mathbb{E}_\epsilon\left|\frac{1}{n}\sum_{i=1}^{n}\epsilon_i K_\sigma(\boldsymbol{x}_i, \boldsymbol{x}_i)\right|^2} \\
&\leq \sqrt{\mathbb{E}_\epsilon\left(\frac{1}{n^2}\sum_{i,j}\epsilon_i\epsilon_j K_\sigma(\boldsymbol{x}_i, \boldsymbol{x}_i)K_\sigma(\boldsymbol{x}_j, \boldsymbol{x}_j)\right)} \\
&\overset{(iii)}{=} \frac{1}{\sqrt{n}}\left\|K_\sigma\right\|_\infty,
\end{aligned}
$$

where (i) follows from the absence of $g$ inside the expectation, (ii) follows from Jensen's inequality and (iii) follows from the fact that $\epsilon_i \perp\!\!\!\perp \epsilon_j$ for $i \neq j$. For the second term, we have

$$
\begin{aligned}
\textcircled{2} = \mathbb{E}_\epsilon\left(\sup_{g\in\mathcal{G}}\left|\frac{1}{n}\sum_{i=1}^{n}\epsilon_i\left\|g\right\|_{\mathcal{H}_\sigma}^2\right|\right) &= \mathbb{E}_\epsilon\left(\sup_{g\in\mathcal{G}}\left\|g\right\|_{\mathcal{H}_\sigma}^2\left|\frac{1}{n}\sum_{i=1}^{n}\epsilon_i\right|\right) \\
&\leq \sup_{g\in\mathcal{G}}\left\|g\right\|_{\mathcal{H}_\sigma}^2\sqrt{\mathbb{E}_\epsilon\left|\frac{1}{n}\sum_{i=1}^{n}\epsilon_i\right|^2}, \\
&\overset{(iv)}{\leq} \frac{1}{\sqrt{n}}\left\|K_\sigma\right\|_\infty,
\end{aligned}
$$

where (iv) follows from the fact that $\left\|g\right\|_{\mathcal{H}_\sigma}^2 \leq \left\|K_\sigma\right\|_\infty$. Lastly, we have

$$
\begin{aligned}
\textcircled{3} = 2\mathbb{E}_\epsilon\left(\sup_{g\in\mathcal{G}}\left|\frac{1}{n}\sum_{i=1}^{n}\epsilon_i g(\boldsymbol{x}_i)\right|\right) &\overset{(v)}{=} 2\mathbb{E}_\epsilon\left(\sup_{g\in\mathcal{G}}\left|\left\langle g, \frac{1}{n}\sum_{i=1}^{n}\epsilon_i K_\sigma(\cdot, \boldsymbol{x}_i)\right\rangle_{\mathcal{H}_\sigma}\right|\right) \\
&\overset{(vi)}{\leq} 2\mathbb{E}_\epsilon\left(\sup_{g\in\mathcal{G}}\left\|g\right\|_{\mathcal{H}_\sigma}\left\|\frac{1}{n}\sum_{i=1}^{n}\epsilon_i K_\sigma(\cdot, \boldsymbol{x}_i)\right\|_{\mathcal{H}_\sigma}\right) \\
&= 2\sup_{g\in\mathcal{G}}\left\|g\right\|_{\mathcal{H}_\sigma}\mathbb{E}_\epsilon\left(\sqrt{\frac{1}{n^2}\sum_{i,j}\epsilon_i\epsilon_j K_\sigma(\boldsymbol{x}_i, \boldsymbol{x}_j)}\right) \\
&\overset{(vii)}{\leq} 2\frac{\left\|K_\sigma\right\|_\infty^{\frac{1}{2}}}{n}\sqrt{\mathbb{E}_\epsilon\left(\sum_{i,j}\epsilon_i\epsilon_j K_\sigma(\boldsymbol{x}_i, \boldsymbol{x}_j)\right)} \\
&\overset{(viii)}{\leq} \frac{2}{\sqrt{n}}\left\|K_\sigma\right\|_\infty,
\end{aligned}
$$

where (v) follows from the reproducing property, (vi) is obtained from Cauchy-Schwarz inequality, (vii) follows from Jensen's inequality, and (viii) follows from the fact that $\epsilon_i \perp\!\!\!\perp \epsilon_j$ for $i \neq j$. Collecting these three inequalities, we have

$$
\mathfrak{R}_n(\mathcal{F}) = \textcircled{1} + \textcircled{2} + \textcircled{3} \leq \frac{4}{\sqrt{n}}\left\|K_\sigma\right\|_\infty.
$$

Plugging this into Eq. (A.12), we have with probability greater than $1 - e^{-\delta}$,

$$
\left\|\mathbb{P}_n - \mathbb{P}\right\|_\mathcal{F} \leq \frac{8\left\|K_\sigma\right\|_\infty}{\sqrt{n}} + \sqrt{\frac{3\delta\left\|K_\sigma\right\|_\infty}{n}},
$$

which implies that $\left\|\mathbb{P}_n - \mathbb{P}\right\|_\mathcal{F} \to 0$ as $n \to \infty$, implying that $\mathcal{F}$ is a Glivenko-Cantelli class. The result, therefore, follows from Lemma A.1. ∎

### A.4 Proof of Theorem 4.7

For $g \in \mathcal{G}$ define the random fluctuation w.r.t. $f_{\rho,\sigma}$ as

$$\Delta(\boldsymbol{X}, g) = \left(\ell_g(\boldsymbol{X}) - \ell_{f_{\rho,\sigma}}(\boldsymbol{X})\right) - \left(\mathcal{J}(g) - \mathcal{J}(f_{\rho,\sigma})\right).$$

The fluctuation process is an empirical process defined as

$$\Delta_n(g) = \mathbb{P}_n \Delta(\boldsymbol{X}, g) = \left(\mathcal{J}_n(g) - \mathcal{J}_n(f_{\rho,\sigma})\right) - \left(\mathcal{J}(g) - \mathcal{J}(f_{\rho,\sigma})\right),$$
$$= \mathbb{P}_n \left(\ell_g - \ell_{f_{\rho,\sigma}}\right) - \mathbb{P} \left(\ell_g - \ell_{f_{\rho,\sigma}}\right).$$

We first show that the behaviour of $\left\|f_{\rho,\sigma}^n - f_{\rho,\sigma}\right\|_{\mathcal{H}_\sigma}$ is controlled by the tail behaviour of the supremum of the fluctuation process. To this end, for $\delta > 0$, let

$$\mathcal{G}_\delta = \left\{g \in \mathcal{G} : \|g - f_{\rho,\sigma}\|_{\mathcal{H}_\sigma} \leq \delta\right\} = B_{\mathcal{H}_\sigma}\left(f_{\rho,\sigma}, \delta\right) \cap \mathcal{D}_\sigma.$$

Suppose $f_{\rho,\sigma}^n$ is such that $\left\|f_{\rho,\sigma}^n - f_{\rho,\sigma}\right\|_{\mathcal{H}_\sigma} > \delta$, then, for sufficiently small $\lambda \in (0, 1)$ such that $g = \lambda f_{\rho,\sigma}^n + (1 - \lambda)f_{\rho,\sigma} \in \mathcal{G}_\delta$, we have that

$$\mathcal{J}_n(g) - \mathcal{J}_n(f_{\rho,\sigma}) \overset{(i)}{<} \lambda \mathcal{J}_n(f_{\rho,\sigma}^n) + (1 - \lambda)\mathcal{J}_n(f_{\rho,\sigma}) - \mathcal{J}_n(f_{\rho,\sigma})$$
$$= \lambda \cdot \left(\mathcal{J}_n(f_{\rho,\sigma}^n) - \mathcal{J}_n(f_{\rho,\sigma})\right) \overset{(ii)}{\leq} 0, \tag{A.13}$$

where (i) follows from the strict convexity of $\mathcal{J}_n$ (Proposition B.1), and (ii) follows from the fact that $f_{\rho,\sigma}^n = \arg\inf_{g \in \mathcal{G}} \mathcal{J}_n(g)$. From Proposition B.1, we also know that $\mathcal{J}$ is strongly convex such that

$$\mathcal{J}(g) - \mathcal{J}(f_{\rho,\sigma}) \geq \frac{\mu}{2} \|g - f_{\rho,\sigma}\|_{\mathcal{H}_\sigma}^2. \tag{A.14}$$

Combining equations (A.13) and (A.14) we have

$$\frac{\mu}{2} \|g - f_{\rho,\sigma}\|_{\mathcal{H}_\sigma}^2 \leq \mathcal{J}(g) - \mathcal{J}(f_{\rho,\sigma}),$$
$$= -\left\{\left(\mathcal{J}_n(g) - \mathcal{J}_n(f_{\rho,\sigma})\right) - \left(\mathcal{J}(g) - \mathcal{J}(f_{\rho,\sigma})\right)\right\} + \left(\mathcal{J}_n(g) - \mathcal{J}_n(f_{\rho,\sigma})\right)$$
$$\leq -\Delta_n(g) \leq \sup_{g \in \mathcal{G}_\delta} |\Delta_n(g)|.$$

By taking the supremum of the left hand side in the above inequality over all $g \in \mathcal{G}_\delta$ we have

$$\sup_{g \in \mathcal{G}_\delta} |\Delta_n(g)| \geq \frac{\mu}{2} \delta^2 \tag{A.15}$$

This implies that whenever $\left\|f_{\rho,\sigma}^n - f_{\rho,\sigma}\right\|_{\mathcal{H}_\sigma} > \delta$ holds, then the condition in Eq. (A.15) holds. Therefore,

$$\mathbb{P}^{\otimes n}\left\{\boldsymbol{X}_{1:n} : \left\|f_{\rho,\sigma}^n - f_{\rho,\sigma}\right\|_{\mathcal{H}_\sigma} > \delta\right\} \leq \mathbb{P}^{\otimes n}\left\{\boldsymbol{X}_{1:n} : \sup_{g \in \mathcal{G}_\delta} |\Delta_n(g)| \geq \frac{\mu}{2} \delta^2\right\}. \tag{A.16}$$

We now study the behaviour of the r.h.s. in Eq. (A.16) using tools from empirical process theory. First, we show that $F(\boldsymbol{x}_{1:n}) = F(\boldsymbol{x}_1, \boldsymbol{x}_2, \ldots, \boldsymbol{x}_n) = \sup_{g \in \mathcal{G}_\delta} |\Delta_n(g)|$ satisfies the self-bounding property.

$$\sup_{\boldsymbol{x}_i \neq \boldsymbol{x}_i'} |F(\boldsymbol{x}_{1:n}) - F(\boldsymbol{x}_{1:n}')| = \sup_{\boldsymbol{x}_i \neq \boldsymbol{x}_i'} \left|\sup_{g \in \mathcal{G}_\delta} |\Delta_n(g)| - \sup_{g \in \mathcal{G}_\delta} |\Delta_n(g)|\right|,$$
$$\leq \sup_{\boldsymbol{x}_i \neq \boldsymbol{x}_i'} \sup_{g \in \mathcal{G}_\delta} \left|\Delta_n(g) - \Delta_n'(g)\right|,$$
$$= \frac{1}{n} \sup_{\boldsymbol{x}_i \neq \boldsymbol{x}_i'} \sup_{g \in \mathcal{G}_\delta} \left|\left(\ell_g(\boldsymbol{x}_i) - \ell_{f_{\rho,\sigma}}(\boldsymbol{x}_i)\right) - \left(\ell_g(\boldsymbol{x}_i') - \ell_{f_{\rho,\sigma}}(\boldsymbol{x}_i')\right)\right|,$$
$$\leq \frac{1}{n} \sup_{\boldsymbol{x}_i \neq \boldsymbol{x}_i'} \sup_{g \in \mathcal{G}_\delta} \left|\left(\ell_g(\boldsymbol{x}_i) - \ell_{f_{\rho,\sigma}}(\boldsymbol{x}_i)\right)\right| + \left|\left(\ell_g(\boldsymbol{x}_i') - \ell_{f_{\rho,\sigma}}(\boldsymbol{x}_i')\right)\right|,$$
$$\overset{(i)}{\leq} \frac{1}{n} \sup_{g \in \mathcal{G}_\delta} 2M \|g - f_{\rho,\sigma}\|_{\mathcal{H}_\sigma} = \frac{2M\delta}{n},$$

where (i) follows from Proposition B.1 that $\ell_g$ is $M$-Lipschitz w.r.t. $\|\cdot\|_{\mathcal{H}_\sigma}$. Therefore, from McDiarmid's inequality [40, Theorem 2.9.1] we have

$$\mathbb{P}^{\otimes n}\left\{\boldsymbol{X}_{1:n} : \sup_{g\in\mathcal{G}_\delta}|\Delta_n(g)| > \mathbb{E}\sup_{g\in\mathcal{G}_\delta}|\Delta_n(g)| + \epsilon\right\} \leq \exp\left(-\frac{n\epsilon^2}{2M^2\delta^2}\right). \tag{A.17}$$

Next, we find an upper bound for the expected supremum of the fluctuation process. In order to do so, we first show that $\Delta_n(g)$ has sub-Gaussian increments. For fixed $g, h \in \mathcal{G}$ we have that $\mathbb{E}\left(\Delta(\boldsymbol{X}, g) - \Delta(\boldsymbol{X}, h)\right) = 0$ and

$$\left|\Delta(\boldsymbol{X}, g) - \Delta(\boldsymbol{X}, h)\right| \leq \left|\ell_g(\boldsymbol{X}) - \ell_h(\boldsymbol{X})\right| - \left|\mathcal{J}(g) - \mathcal{J}(h)\right| \leq 2M\|g - h\|_{\mathcal{H}_\sigma}.$$

Since $\left|\Delta(\boldsymbol{X}, g) - \Delta(\boldsymbol{X}, h)\right|$ is bounded, it is, therefore, sub-Gaussian and from Vershynin [40, Example 2.5.8], we have that the sub-Gaussian norm $\|\Delta(\boldsymbol{X}, g) - \Delta(\boldsymbol{X}, h)\|_{\psi_2} \leq 2cM\|g - h\|_{\mathcal{H}_\sigma}$ for $c > 1/\sqrt{\log 2}$. Consequently, the fluctuation process has sub-Gaussian increments with respect to the metric $\|g - h\|_{\mathcal{H}_\sigma}$, i.e.,

$$\|\Delta_n(g) - \Delta_n(h)\|_{\psi_2} \leq \frac{1}{n}\sqrt{\sum_{i=1}^n \|\Delta(\boldsymbol{X}_i, g) - \Delta(\boldsymbol{X}_i, h)\|_{\psi_2}^2} \leq \frac{M}{\sqrt{n}}\|g - h\|_{\mathcal{H}_\sigma}.$$

From the generalized entropy integral [33, Lemma A.3], for a fixed constant $\gamma > 12/\sqrt{\log 2}$ we have

$$\mathbb{E}\sup_{g\in\mathcal{G}_\delta}|\Delta_n(g)| \leq \inf_{\alpha>0}\left\{2\alpha + \frac{\gamma M}{\sqrt{n}}\int_\alpha^\delta \sqrt{\log\mathcal{N}\left(\mathcal{G}_\delta, \|\cdot\|_{\mathcal{H}_\sigma}, \epsilon\right)}d\epsilon\right\}, \tag{A.18}$$

where $\mathcal{N}\left(\mathcal{G}_\delta, d, \epsilon\right)$ is the $\epsilon$-covering number of the class $\mathcal{G}_\delta$ with respect to metric $d$.

We now turn our attention to finding an upper bound for $\mathcal{N}\left(\mathcal{G}_\delta, d, \epsilon\right)$. Note that if $\mathcal{B}_{\mathcal{H}_\sigma}$ is a unit ball in the RKHS, then

$$\log\mathcal{N}\left(\mathcal{G}_\delta, \|\cdot\|_{\mathcal{H}_\sigma}, \epsilon\right) = \log\mathcal{N}\left(\mathcal{B}_{\mathcal{H}_\sigma} \cap \mathcal{D}_\sigma, \|\cdot\|_{\mathcal{H}_\sigma}, \frac{\epsilon}{\delta}\right)$$

$$\overset{(i)}{\leq} \log\mathcal{N}\left(\mathcal{B}_{\mathcal{H}_\sigma} \cap \mathcal{D}_\sigma, \|\cdot\|_\infty, \left(\frac{\epsilon}{\delta}\right)^2\right)$$

$$\leq \log\mathcal{N}\left(\mathcal{B}_{\mathcal{H}_\sigma}, \|\cdot\|_\infty, \left(\frac{\epsilon}{\delta}\right)^2\right),$$

where (i) follows from Lemma B.6 that $\|g - h\|_{\mathcal{H}_\sigma}^2 \leq \|g - h\|_\infty$. When the entropy numbers $e_n(\mathrm{id} : \mathcal{H}_\sigma \to L_\infty(\mathcal{X}))$ satisfy the assumption, from [35, Lemma 6.21] we have

$$\log\mathcal{N}\left(\mathcal{B}_{\mathcal{H}_\sigma}, \|\cdot\|_\infty, \left(\frac{\epsilon}{\delta}\right)^2\right) \leq \left(\frac{a_\sigma\delta^2}{\epsilon^2}\right)^{2p}.$$

Plugging this into Eq. (A.18), we have that

$$\mathbb{E}\sup_{g\in\mathcal{G}_\delta}|\Delta_n(g)| \leq \inf_{\alpha>0}\left\{2\alpha + \frac{\gamma M a_\sigma\delta^{2p}}{\sqrt{n}}\int_\alpha^\delta \epsilon^{-2p}d\epsilon\right\} = \inf_{\alpha>0}T(\alpha),$$

where $T(\alpha)$ is given by

$$T(\alpha) = \begin{cases} 2\alpha + \gamma M\delta\sqrt{\frac{a_\sigma}{n}}\log\left(\frac{\delta}{\alpha}\right) & \text{if } p = \frac{1}{2}, \\ 2\alpha + \frac{\gamma M}{(1-2p)\sqrt{n}}\left(\delta - \delta^{2p}\alpha^{1-2p}\right) & \text{if } 0 < p \neq \frac{1}{2} < 1. \end{cases}$$

At the value $\alpha_0$ where $T(\alpha_0) = \inf_{\alpha>0}T(\alpha)$, we have

$$T(\alpha_0) = \begin{cases} \gamma Ca_\sigma^{\frac{1}{2}} \cdot \frac{M\delta\log(n)}{\sqrt{n}} & \text{if } p = \frac{1}{2}, \\ \frac{\gamma a_\sigma^p}{(1-2p)} \cdot \frac{M\delta}{\sqrt{n}} - \frac{Kpa_\sigma^{\frac{1}{2}}}{(1-2p)} \cdot \frac{M\delta}{n^{1/4p}} & \text{if } 0 < p \neq \frac{1}{2} < 1, \end{cases} \tag{A.19}$$

for some fixed constant $C > 3 - \log(9a)$. Observe that when $0 < p < \frac{1}{2}$, the last term of Eq. (A.19) is negative, and similarly when $\frac{1}{2} < p < 1$, the first term is negative. From this, we have that $T(\alpha_0) \leq M\delta\xi(n,p)$ where

$$
\xi(n,p) = \begin{cases}
\frac{\gamma a_\sigma^p}{(1-2p)} \cdot \frac{1}{\sqrt{n}} & \text{if } 0 < p < \frac{1}{2}, \\[2ex]
\gamma C a_\sigma^{\frac{1}{2}} \cdot \frac{\log(n)}{\sqrt{n}} & \text{if } p = \frac{1}{2}, \\[2ex]
\frac{\gamma p a_\sigma^{\frac{1}{2}}}{2p-1} \cdot \frac{1}{n^{1/4p}} & \text{if } \frac{1}{2} < p < 1.
\end{cases}
$$

Plugging this into Eq. (A.17), we have that with probability greater than $1 - e^{-t}$,

$$
\sup_{g \in \mathcal{G}_\delta} |\Delta_n(g)| < M\delta\xi(n,p) + M\delta\sqrt{\frac{2t}{n}}. \tag{A.20}
$$

From Eq. (A.16), this implies that

$$
\mathbb{P}^{\otimes n}\left\{\boldsymbol{X}_{1:n} : \left\|f_{\rho,\sigma}^n - f_{\rho,\sigma}\right\|_{\mathcal{H}_\sigma} > \delta\right\} \leq \mathbb{P}^{\otimes n}\left\{\boldsymbol{X}_{1:n} : \sup_{g \in \mathcal{G}_\delta} \Delta_n(g) \geq \frac{\mu\delta^2}{2}\right\}.
$$

Thus, in Eq. (A.20), by letting

$$
\frac{\mu\delta^2}{2} = \left(M\delta\xi(n,p) + M\delta\sqrt{\frac{2t}{n}}\right),
$$

we have that with probability greater than $1 - e^{-t}$,

$$
\left\|f_{\rho,\sigma}^n - f_{\rho,\sigma}\right\|_{\mathcal{H}_\sigma} \leq \frac{2M}{\mu}\left(\xi(n,p) + \sqrt{\frac{2t}{n}}\right).
$$

Observe that $\left\|f_{\rho,\sigma}^n - f_{\rho,\sigma}\right\|_\infty \leq \|K_\sigma\|_\infty^{\frac{1}{2}} \left\|f_{\rho,\sigma}^n - f_{\rho,\sigma}\right\|_{\mathcal{H}_\sigma}$. For $0 < \alpha < 1$, by choosing $\delta_n$ as

$$
\delta_n = \frac{2M \|K_\sigma\|_\infty^{\frac{1}{2}}}{\mu}\left(\xi(n,p) + \sqrt{\frac{2\log(1/\alpha)}{n}}\right),
$$

we have that

$$
\mathbb{P}^{\otimes n}\left\{\boldsymbol{X}_{1:n} : \left\|f_{\rho,\sigma}^n - f_{\rho,\sigma}\right\|_\infty \leq \delta_n\right\} > 1 - \alpha.
$$

From the stability of persistence diagrams in Proposition 2.2, this implies that

$$
\mathbb{P}^{\otimes n}\left\{\boldsymbol{X}_{1:n} : W_\infty\Big(\mathsf{Dgm}\left(f_{\rho,\sigma}^n\right), \mathsf{Dgm}\left(f_{\rho,\sigma}\right)\Big) > \delta_n\right\} \leq \alpha,
$$

yielding the desired result. $\blacksquare$

# B  Supplementary Results

In this section, we establish some results which play a key role in the proofs presented in Section A.

## B.1  Properties of the Risk Functional $\mathcal{J}(g)$

We establish some important properties of the risk functional, given by

$$
\mathcal{J}(g) = \int_{\mathbb{R}^d} \ell_g(\boldsymbol{x}) \, d\mathbb{P}(\boldsymbol{x}) = \int_{\mathbb{R}^d} \rho\left(\|\Phi_\sigma(\boldsymbol{x}) - g\|_{\mathcal{H}_\sigma}\right) d\mathbb{P}(\boldsymbol{x}).
$$

The following result establishes that some important properties of the robust loss $\rho$ carry forward to $\mathcal{J}(g)$. (i) The Lipschitz property of $\rho$ is inherited by $\mathcal{J}(g)$, (ii) the convexity of $\rho$ is strengthened to guarantee that $\mathcal{J}(g)$ is strictly convex, and (iii) $\mathcal{J}(g)$ is strongly convex with respect to the $\|\cdot\|_{\mathcal{H}_\sigma}$–norm around its minimizer.

**Proposition B.1** (Convexity and Lipchitz properties of $\mathcal{J}$). *Under assumptions* $(\mathcal{A}1) - (\mathcal{A}3)$,

    (i) *The risk functionals* $\mathcal{J}(g)$ *and* $\mathcal{J}_n(g)$ *are* $M$-*Lipschitz w.r.t.* $\|\cdot\|_{\mathcal{H}_\sigma}$.

    (ii) *Furthermore, if* $\rho$ *is convex,* $\mathcal{J}(g)$ *and* $\mathcal{J}_n(g)$ *are strictly convex.*

    (iii) *Additionally, under assumption* $(\mathcal{A}4)$, *for* $f_{\rho,\sigma} = \arg\inf_{g \in \mathcal{G}} \mathcal{J}(g)$, *the risk functional satisfies the strong convexity condition*

$$\mathcal{J}(g) - \mathcal{J}(f_{\rho,\sigma}) \geq \frac{\mu}{2} \|f_{\rho,\sigma} - g\|_{\mathcal{H}_\sigma}^2,$$

*for* $\mu = 2\min\left\{ \varphi\left(2\|K_\sigma\|_\infty^{\frac{1}{2}}\right), \rho''\left(2\|K_\sigma\|_\infty^{\frac{1}{2}}\right)\right\}$.

*Proof.* **Lipschitz property.** Observe that,

$$
\begin{aligned}
|\ell_{g_1}(\boldsymbol{x}) - \ell_{g_2}(\boldsymbol{x})| &= \left|\rho\left(\|\Phi_\sigma(\boldsymbol{x}) - g_1\|_{\mathcal{H}_\sigma}\right) - \rho\left(\|\Phi_\sigma(\boldsymbol{x}) - g_2\|_{\mathcal{H}_\sigma}\right)\right| \\
&\leq M \left|\|\Phi_\sigma(\boldsymbol{x}) - g_1\|_{\mathcal{H}_\sigma} - \|\Phi_\sigma(\boldsymbol{x}) - g_2\|_{\mathcal{H}_\sigma}\right| \\
&\leq M \|g_1 - g_2\|_{\mathcal{H}_\sigma},
\end{aligned}
$$

where the first inequality follows from the fact that $\rho$ is $M$-Lipschitz and the last inequality follows from reverse triangle inequality. This shows that the loss functions $\ell_g(\cdot)$ are $M$-Lipschitz with respect to $g$. For the risk functionals, we have that,

$$
\begin{aligned}
|\mathcal{J}(g_1) - \mathcal{J}(g_2)| &= \left| \int_{\mathbb{R}^d} (\ell_{g_1}(\boldsymbol{x}) - \ell_{g_2}(\boldsymbol{x})) \, d\mathbb{P}(\boldsymbol{x}) \right| \\
&\leq \int_{\mathbb{R}^d} \left| \ell_{g_1}(\boldsymbol{x}) - \ell_{g_2}(\boldsymbol{x}) \right| d\mathbb{P}(\boldsymbol{x}) \\
&\leq M \|g_1 - g_2\|_{\mathcal{H}_\sigma},
\end{aligned}
$$

where the first inequality follows from Jensen's inequality. This verifies that $\mathcal{J}(g)$ is $M$-Lipchitz. The proof for $\mathcal{J}_n(g)$ is identical.

**Strict Convexity.** We begin by establishing that for translation invariant kernels $\|\Phi_\sigma(\boldsymbol{x}) - \cdot\|_{\mathcal{H}_\sigma}$ is strictly convex. Suppose $g_1, g_2 \in \mathcal{H}_\sigma \cap \mathcal{D}_\sigma$ and $\lambda \in (0, 1)$, and let $g = (1 - \lambda)g_1 + \lambda g_2$. Then

$$
\begin{aligned}
\|\Phi_\sigma(\boldsymbol{x}) - g\|_{\mathcal{H}_\sigma}^2 &= \|(1 - \lambda)(\Phi_\sigma(\boldsymbol{x}) - g_1) + \lambda(\Phi_\sigma(\boldsymbol{x}) - g_2)\|_{\mathcal{H}_\sigma}^2 \\
&= (1 - \lambda)^2 \|\Phi_\sigma(\boldsymbol{x}) - g_1\|_{\mathcal{H}_\sigma}^2 \\
&\quad + \lambda^2 \|\Phi_\sigma(\boldsymbol{x}) - g_2\|_{\mathcal{H}_\sigma}^2 + 2\lambda(1 - \lambda)\Big\langle \Phi_\sigma(\boldsymbol{x}) - g_1, \Phi_\sigma(\boldsymbol{x}) - g_2\Big\rangle_{\mathcal{H}_\sigma}. \quad \text{(B.1)}
\end{aligned}
$$

From Cauchy-Schwarz inequality, we know that

$$\Big\langle \Phi_\sigma(\boldsymbol{x}) - g_1, \Phi_\sigma(\boldsymbol{x}) - g_2\Big\rangle_{\mathcal{H}_\sigma} \leq \|\Phi_\sigma(\boldsymbol{x}) - g_1\|_{\mathcal{H}_\sigma} \|\Phi_\sigma(\boldsymbol{x}) - g_2\|_{\mathcal{H}_\sigma}.$$

In the following, we argue that for translation invariant kernels,

$$\Big\langle \Phi_\sigma(\boldsymbol{x}) - g_1, \Phi_\sigma(\boldsymbol{x}) - g_2\Big\rangle_{\mathcal{H}_\sigma} < \|\Phi_\sigma(\boldsymbol{x}) - g_1\|_{\mathcal{H}_\sigma} \|\Phi_\sigma(\boldsymbol{x}) - g_2\|_{\mathcal{H}_\sigma}, \quad \text{(B.2)}$$

for $g_1 \neq g_2$. On the contrary, suppose

$$\Big\langle \Phi_\sigma(\boldsymbol{x}) - g_1, \Phi_\sigma(\boldsymbol{x}) - g_2\Big\rangle_{\mathcal{H}_\sigma} = \|\Phi_\sigma(\boldsymbol{x}) - g_1\|_{\mathcal{H}_\sigma} \|\Phi_\sigma(\boldsymbol{x}) - g_2\|_{\mathcal{H}_\sigma}$$

holds. Then this implies that there is a function $a(\boldsymbol{x})$, depending only on $g_1$ and $g_2$, such that $a(\boldsymbol{x}) \neq 0$ for $\boldsymbol{x} \in \mathbb{R}^d$ and

$$\Phi_\sigma(\boldsymbol{x}) - g_1 = a(\boldsymbol{x})\left(\Phi_\sigma(\boldsymbol{x}) - g_2\right).$$

Rearranging the terms this implies that

$$\Phi_\sigma(\boldsymbol{x}) = \frac{g_1 - a(\boldsymbol{x})g_2}{1 - a(\boldsymbol{x})} = (1 + b(\boldsymbol{x}))g_1 + b(\boldsymbol{x})g_2,$$

where $b(\boldsymbol{x}) = -a(\boldsymbol{x})/(1 - a(\boldsymbol{x}))$ also does not vanish on $\boldsymbol{x} \in \mathbb{R}^d$. For $\boldsymbol{x}, \boldsymbol{y} \in \mathbb{R}^d$, from the reproducing property we have

$$
\begin{aligned}
K_\sigma(\boldsymbol{x}, \boldsymbol{y}) &= \left\langle \Phi_\sigma(\boldsymbol{x}), \Phi_\sigma(\boldsymbol{y}) \right\rangle_{\mathcal{H}_\sigma} \\
&= \left\langle g_1 + b(\boldsymbol{x})(g_1 + g_2), g_1 + b(\boldsymbol{y})(g_1 + g_2) \right\rangle_{\mathcal{H}_\sigma} \\
&= b(\boldsymbol{x})b(\boldsymbol{y}) \|g_1 + g_2\|_{\mathcal{H}_\sigma}^2 + (b(\boldsymbol{x}) + b(\boldsymbol{y})) \langle g_1, g_1 + g_2 \rangle_{\mathcal{H}_\sigma} + \|g_1\|_{\mathcal{H}_\sigma}^2.
\end{aligned}
$$

Note that because the kernel is translation invariant, i.e., $K_\sigma(\boldsymbol{x}, \boldsymbol{x}) = K_\sigma(\boldsymbol{y}, \boldsymbol{y}) = \sigma^{-d}\psi(0)$, this must imply that

$$
\begin{aligned}
0 &= \left(b(\boldsymbol{x})^2 - b(\boldsymbol{y})^2\right) \|g_1 + g_2\|_{\mathcal{H}_\sigma}^2 + 2(b(\boldsymbol{x}) - b(\boldsymbol{y}))\langle g_1, g_1 + g_2 \rangle_{\mathcal{H}_\sigma} \\
&= (b(\boldsymbol{x}) - b(\boldsymbol{y})) \left((b(\boldsymbol{x}) + b(\boldsymbol{y})) \|g_1 + g_2\|_{\mathcal{H}_\sigma}^2 + 2\langle g_1, g_1 + g_2 \rangle_{\mathcal{H}_\sigma} \right).
\end{aligned}
$$

Since $b(\boldsymbol{x})$ and $b(\boldsymbol{y})$ are nonvanishing, the above equation is satisfied only when $b(\boldsymbol{x}) = b(\boldsymbol{y})$. This implies that $K_\sigma(\boldsymbol{x}, \boldsymbol{y})$ is constant for all $\boldsymbol{y}$, giving us a contradiction. Thus, we have that Eq. (B.2) holds. Plugging this back in Eq. (B.1) we get that for $\lambda \in (0, 1)$ and $g = (1 - \lambda)g_1 + \lambda g_2$,

$$\|\Phi_\sigma(\boldsymbol{x}) - g\|_{\mathcal{H}_\sigma} < (1 - \lambda) \|\Phi_\sigma(\boldsymbol{x}) - g_1\|_{\mathcal{H}_\sigma} + \lambda \|\Phi_\sigma(\boldsymbol{x}) - g_2\|_{\mathcal{H}_\sigma}.$$

Since, $\rho$ is strictly increasing and convex, this implies that

$$\ell_g(\boldsymbol{x}) < (1 - \lambda)\ell_{g_1}(\boldsymbol{x}) + \lambda\ell_{g_2}(\boldsymbol{x}).$$

The map $\ell_g(\cdot) \mapsto \mathbb{P}\ell_g$ is a linear operator, and $\ell_g$ is strictly convex in $g$, this implies that $\mathcal{J}(g)$ is also strictly convex in $g$. The same holds for $\mathcal{J}_n(g)$.

**Strong Convexity around the minimizer.** We now turn our attention to the strong convexity property. For this, we first show that $\mathcal{J}(g)$ is twice Gâteaux differentiable. Let $g, h \in \mathcal{G}$, then the second Gâteaux derivative of the loss $\ell_g(\boldsymbol{x}) = \rho\left(\|\Phi_\sigma(\boldsymbol{x}) - g\|_{\mathcal{H}_\sigma}\right)$ at $g$ in the direction $h$ is given by,

$$
\begin{aligned}
\delta^2 \ell(\boldsymbol{x}, g; h) &= \frac{d^2}{d\alpha^2} \ell(\boldsymbol{x}, g + \alpha h) \Big|_{\alpha=0} \\
&= \frac{d^2}{d\alpha^2} \rho\left(\|\Phi_\sigma(\boldsymbol{x}) - g - \alpha h\|_{\mathcal{H}_\sigma}\right) \Big|_{\alpha=0} \\
&= \frac{d}{d\alpha} \left[ \varphi\left(\|\Phi_\sigma(\boldsymbol{x}) - g - \alpha h\|_{\mathcal{H}_\sigma}\right) \left(-\langle\Phi_\sigma(\boldsymbol{x}) - g, h\rangle_{\mathcal{H}_\sigma} + \alpha \|h\|_{\mathcal{H}_\sigma}^2\right) \right] \Big|_{\alpha=0} \\
&= \varphi\left(\|\Phi_\sigma(\boldsymbol{x}) - g\|_{\mathcal{H}_\sigma}\right) \|h\|_{\mathcal{H}_\sigma}^2 + \langle\Phi_\sigma(\boldsymbol{x}) - g, h\rangle_{\mathcal{H}_\sigma}^2 \frac{\varphi'\left(\|\Phi_\sigma(\boldsymbol{x}) - g\|_{\mathcal{H}_\sigma}\right)}{\|\Phi_\sigma(\boldsymbol{x}) - g\|_{\mathcal{H}_\sigma}} \\
&= \varphi\left(z(\boldsymbol{x}, g)\right) \|h\|_{\mathcal{H}_\sigma}^2 + \|h\|_{\mathcal{H}_\sigma}^2 \lambda(\boldsymbol{x}, g, h)z(\boldsymbol{x}, g)\varphi'\left(z(\boldsymbol{x}, g)\right), \quad\quad\quad (B.3)
\end{aligned}
$$

where for a fixed $g \in \mathcal{G}$, in the interest of brevity, we define $z(\boldsymbol{x}, g) = \|\Phi_\sigma(\boldsymbol{x}) - g\|_{\mathcal{H}_\sigma}$ and

$$\lambda(\boldsymbol{x}, g, h) = \left\langle \frac{\Phi_\sigma(\boldsymbol{x}) - g}{\|\Phi_\sigma(\boldsymbol{x}) - g\|_{\mathcal{H}_\sigma}}, \frac{h}{\|h\|_{\mathcal{H}_\sigma}} \right\rangle_{\mathcal{H}_\sigma}^2 \in [0, 1].$$

Observe that $z\varphi'(z) = \rho''(z) - \varphi(z)$, thus Eq. (B.3) becomes

$$\delta^2 \ell(\boldsymbol{x}, g; h) = \|h\|_{\mathcal{H}_\sigma}^2 \left((1 - \lambda(\boldsymbol{x}, g, h)) \varphi(z(\boldsymbol{x}, g)) + \lambda(\boldsymbol{x}, g, h)\rho''(z(\boldsymbol{x}, g))\right).$$

From assumption $(\mathcal{A}4)$ we have that $\rho''$ and $\varphi$ are nonincreasing, and

$$z(\boldsymbol{x}, g) = \|\Phi_\sigma(\boldsymbol{x}) - g\|_{\mathcal{H}_\sigma} \le 2 \|K_\sigma\|_\infty^{\frac{1}{2}}.$$

Thus, we have that

$$\delta^2 \ell(\boldsymbol{x}, g; h) \geq c \|h\|_{\mathcal{H}_\sigma}^2 , \tag{B.4}$$

where

$$c = \min \left\{ \varphi \left( 2 \|K_\sigma\|_\infty^{\frac{1}{2}} \right), \rho'' \left( 2 \|K_\sigma\|_\infty^{\frac{1}{2}} \right) \right\}.$$

We also note that $\delta^2 \ell(\boldsymbol{x}, g; h)$ is bounded above. To see this, note that from assumption $(\mathcal{A}4)$, $\rho''$ and $\varphi$ are bounded and nonincreasing. Consequently, for $\lambda(\boldsymbol{x}, g, h) \in (0, 1)$ and

$$C = \max \left\{ \rho''(0), \varphi(0) \right\} < \infty,$$

from Eq. (B.3) we have that

$$\delta^2 \ell(\boldsymbol{x}, g; h) \leq C \|h\|_{\mathcal{H}_\sigma}^2 < \infty.$$

The Gâteaux derivative of $\mathcal{J}(g)$ is, then, given by

$$\delta^2 \mathcal{J}(g; h) = \frac{d^2}{d\alpha^2} \mathcal{J}(g + \alpha h) \Big|_{\alpha=0} = \frac{d^2}{d\alpha^2} \int_{\mathbb{R}^d} \ell(\boldsymbol{x}, g + \alpha h) \, d\mathbb{P}(\boldsymbol{x}) \Big|_{\alpha=0}$$

$$= \int_{\mathbb{R}^d} \frac{d^2}{d\alpha^2} \ell(\boldsymbol{x}, g + \alpha h) \, d\mathbb{P}(\boldsymbol{x}) \Big|_{\alpha=0}$$

$$= \int_{\mathbb{R}^d} \delta^2 \ell(\boldsymbol{x}, g; h) \, d\mathbb{P}(\boldsymbol{x}).$$

The exchange of the derivative and integral in the second line follows from the dominated convergence theorem since $\left| \delta^2 \ell(\boldsymbol{x}, g; h) \right|$ is bounded. This confirms the Gâteaux differentiability of $\mathcal{J}(g)$. From Eq. (B.4) we have

$$\delta^2 \mathcal{J}(g; h) = \int_{\mathbb{R}^d} \delta^2 \ell(\boldsymbol{x}, g; h) \, d\mathbb{P}(\boldsymbol{x}) \geq c \|h\|_{\mathcal{H}_\sigma}^2 . \tag{B.5}$$

For $f_{\rho,\sigma} = \arg\inf_{g \in \mathcal{G}} \mathcal{J}(g)$ and $g \in \mathcal{G}$, we proceed to show the strong-convexity guarantee. Let $h = g - f_{\rho,\sigma}$. From the first-order Taylor approximation for $\mathcal{J}(g)$ we have,

$$\mathcal{J}(g) = \mathcal{J}(f_{\rho,\sigma}) + \delta\mathcal{J}(f_{\rho,\sigma}, h) + R_2(f_{\rho,\sigma}, h),$$

where the first Gâteaux derivative, $\delta\mathcal{J}(f_{\rho,\sigma}, h) = 0$ for all $h$ since $f_{\rho,\sigma}$ is the unique minimizer of $\mathcal{J}(g)$ and the remainder term $R_2(f_{\rho,\sigma}, h)$ is given by

$$R_2(f_{\rho,\sigma}, h) = \frac{1}{2} \int_0^1 (1-t) \delta^2 \mathcal{J}(f_{\rho,\sigma} + th; h) \, dt$$

$$\geq \frac{c}{2} \|h\|_{\mathcal{H}_\sigma}^2 \int_0^1 (1-t) dt = \frac{c}{4} \|h\|_{\mathcal{H}_\sigma}^2 ,$$

where the inequality follows from Eq. (B.5). As a result, for any $g \in \mathcal{G}$ and $\mu = \frac{c}{2}$ we have that

$$\mathcal{J}(g) - \mathcal{J}(f_{\rho,\sigma}) \geq \frac{\mu}{2} \|g - f_{\rho,\sigma}\|_{\mathcal{H}_\sigma}^2 ,$$

yielding the desired result. ∎

We now turn to examining the behaviour of the risk functional $\mathcal{J}(g)$ w.r.t. the underlying probability measure $\mathbb{P}$. For $0 \leq \epsilon \leq 1$ and $\boldsymbol{x} \in \mathbb{R}^d$, let $\mathbb{P}_{\boldsymbol{x}}^\epsilon = (1 - \epsilon)\mathbb{P} + \epsilon\delta_{\boldsymbol{x}}$ be a perturbation curve, as defined in Theorem 4.2. The risk functional associated with $\mathbb{P}_{\boldsymbol{x}}^\epsilon$ is given by

$$\mathcal{J}_{\epsilon,\boldsymbol{x}}(g) = \mathbb{P}_{\boldsymbol{x}}^\epsilon \ell_g = (1 - \epsilon)\mathcal{J}(g) + \epsilon\rho \left( \|\Phi_\sigma(\boldsymbol{x}) - g\|_{\mathcal{H}_\sigma} \right),$$

and $f_{\rho,\sigma}^{\epsilon,\boldsymbol{x}} = \inf_{g \in \mathcal{G}} \mathcal{J}_{\epsilon,\boldsymbol{x}}(g)$ is the minimizer. The convergence of $f_{\rho,\sigma}^{\epsilon,\boldsymbol{x}}$ to $f_{\rho,\sigma}$ can be studied by examining the convergence of $\mathcal{J}_{\epsilon,\boldsymbol{x}}$ to $\mathcal{J}$. Specifically, under conditions on $\mathcal{J}$ and $\mathcal{J}_{\epsilon,\boldsymbol{x}}$, it can be shown that $\left\| f_{\rho,\sigma}^{\epsilon,\boldsymbol{x}} - f_{\rho,\sigma} \right\|_{\mathcal{H}_\sigma} \to 0$ as $\epsilon \to 0$. The machinery we use here uses the notion of $\Gamma$–convergence, which is defined as follows.

**Definition B.2** (Γ convergence). *Given a functional $F : \mathcal{X} \to \mathbb{R} \cup \{\pm\infty\}$ and a sequence of functionals $\{F_n\}_{n \in \mathbb{N}}$, $F_n \xrightarrow{\Gamma} F$ as $n \to \infty$ when*

    *(i)* $F(\boldsymbol{x}) \leq \liminf\limits_{n \to \infty} F_n(\boldsymbol{x}_n)$ *for all $\boldsymbol{x} \in \mathcal{X}$ and every $\{\boldsymbol{x}_n\}_{n \in \mathbb{N}}$ such that $d(\boldsymbol{x}_n, \boldsymbol{x}) \to 0$;*

    *(ii) For every $\boldsymbol{x} \in \mathcal{X}$, there exists $\{\boldsymbol{x}_n\}_{n \in \mathbb{N}}$, $d(\boldsymbol{x}_n, \boldsymbol{x}) \to 0$ such that $F(\boldsymbol{x}) \geq \limsup\limits_{n \to \infty} F_n(\boldsymbol{x}_n)$.*

The following result shows that the sequence of functionals $\{\mathcal{J}_{\epsilon, \boldsymbol{x}}\}$ Γ–converges to $\mathcal{J}$.

**Proposition B.3** (Γ–convergence of $\mathcal{J}_{\epsilon, \boldsymbol{x}}$ to $\mathcal{J}$). *Under assumptions $(\mathcal{A}1)$–$(\mathcal{A}3)$,*

$$\mathcal{J}_{\epsilon, \boldsymbol{x}}(g) \xrightarrow{\Gamma} \mathcal{J}(g) \quad as \ \epsilon \to 0.$$

*Proof.* Let $g \in \mathcal{G}$ and $\{g_\epsilon\}_{\epsilon > 0}$ be a sequence in $\mathcal{G}$ such that $\|g_\epsilon - g\|_{\mathcal{H}_\sigma} \to 0$ as $\epsilon \to 0$. In order to verify Γ–convergence we first show that the following holds

$$\lim_{\epsilon \to 0} \left| \mathcal{J}_{\epsilon, \boldsymbol{x}}(g_\epsilon) - \mathcal{J}(g) \right| = 0.$$

For $\epsilon > 0$, using the triangle inequality we have that

$$
\begin{aligned}
\left| \mathcal{J}_{\epsilon, \boldsymbol{x}}(g_\epsilon) - \mathcal{J}(g) \right| &\leq \left| \mathcal{J}(g_\epsilon) - \mathcal{J}(g) \right| + \left| \mathcal{J}_{\epsilon, \boldsymbol{x}}(g_\epsilon) - \mathcal{J}(g_\epsilon) \right| \\
&\stackrel{(i)}{\leq} M \|g_\epsilon - g\|_{\mathcal{H}_\sigma} + \left| \mathcal{J}_{\epsilon, \boldsymbol{x}}(g_\epsilon) - \mathcal{J}(g_\epsilon) \right| \\
&\stackrel{(ii)}{\leq} M \|g_\epsilon - g\|_{\mathcal{H}_\sigma} + \epsilon \cdot \left| \mathcal{J}(g) - \rho\left( \|\Phi_\sigma(\boldsymbol{x}) - g\|_{\mathcal{H}_\sigma} \right) \right|,
\end{aligned}
$$

where (i) uses the fact that $\mathcal{J}(g)$ is $M$–Lipschitz from Proposition B.1, and (ii) uses the fact that

$$\mathcal{J}_{\epsilon, \boldsymbol{x}}(g) = (1 - \epsilon)\mathcal{J}(g) + \epsilon\rho\left( \|\Phi_\sigma(\boldsymbol{x}) - g\|_{\mathcal{H}_\sigma} \right).$$

Since $\|g_\epsilon - g\|_{\mathcal{H}_\sigma} \to 0$ as $\epsilon \to 0$ we have

$$\lim_{\epsilon \to 0} \left| \mathcal{J}_{\epsilon, \boldsymbol{x}}(g_\epsilon) - \mathcal{J}(g) \right| \leq M \lim_{\epsilon \to 0} \|g_\epsilon - g\|_{\mathcal{H}_\sigma} + \lim_{\epsilon \to 0} \epsilon \cdot \left| \mathcal{J}(g) - \rho\left( \|\Phi_\sigma(\boldsymbol{x}) - g\|_{\mathcal{H}_\sigma} \right) \right| = 0.$$

Since $\mathcal{J}_{\epsilon, \boldsymbol{x}}$ and $\mathcal{J}$ are continuous, using [17, Remark 4.8] it follows that $\mathcal{J}_{\epsilon, \boldsymbol{x}}(g) \xrightarrow{\Gamma} \mathcal{J}(g)$. ∎

Now, we examine the coercivity of the sequence $\{\mathcal{J}_{\epsilon, \boldsymbol{x}}\}$.

**Definition B.4** (Equi-coercivity). *A sequence of functionals $\{F_n\}_{n \in \mathbb{N}} : \mathcal{X} \to \mathbb{R} \cup \{\pm\infty\}$ is said to be equi-coercive if for every $t \in \mathbb{R}$, there exists a compact set $K_t \subseteq \mathcal{X}$ such that $\{\boldsymbol{x} \in \mathcal{X} : F_n \leq t\} \subseteq K_t$ for every $n \in \mathbb{N}$.*

The following result shows that the sequence $\{\mathcal{J}_{\epsilon, \boldsymbol{x}}\}$ is equi-coercive.

**Proposition B.5** (Equi-coercivity of $\mathcal{J}_{\epsilon, \boldsymbol{x}}$). *Under assumptions $(\mathcal{A}1)$–$(\mathcal{A}3)$, the sequence of functionals $\{\mathcal{J}_{\epsilon, \boldsymbol{x}}\}$ is equi-coercive.*

*Proof.* For $0 < \epsilon < 1$, $\boldsymbol{x} \in \mathbb{R}^d$ and $g \in \mathcal{G}$, we have that

$$\mathcal{J}_{\epsilon, \boldsymbol{x}}(g) = (1 - \epsilon)\mathcal{J}(g) + \epsilon\rho\left( \|\Phi_\sigma(\boldsymbol{x}) - g\|_{\mathcal{H}_\sigma} \right).$$

From [17, Proposition 7.7] in order to show that the sequence of functionals $\{\mathcal{J}_{\epsilon, \boldsymbol{x}}\}$ is equi-coercive, it suffices to show that there exists a lower semicontinuous, coercive functional $F : \mathcal{H}_\sigma \to \mathbb{R} \cup \{\pm\infty\}$ such that $F \leq \mathcal{J}_{\epsilon, \boldsymbol{x}}$ for every $\epsilon \geq 0$. To this end consider the functional

$$F(g) = \min\left\{ \mathcal{J}(g), \rho\left( \|\Phi_\sigma(\boldsymbol{x}) - g\|_{\mathcal{H}_\sigma} \right) \right\}.$$

As $\mathcal{J}_{\epsilon, \boldsymbol{x}}$ is a convex combination of $\mathcal{J}(\cdot)$ and $\rho\left( \|\Phi_\sigma(\boldsymbol{x}) - \cdot\|_{\mathcal{H}_\sigma} \right)$, it implies that $F \leq \mathcal{J}_{\epsilon, \boldsymbol{x}}$ for every $\epsilon \geq 0$. Additionally, because $\mathcal{J}(\cdot)$ and $\rho\left( \|\Phi_\sigma(\boldsymbol{x}) - \cdot\|_{\mathcal{H}_\sigma} \right)$ are both continuous, it follows that $F$ is also continuous, and, therefore, lower semicontinuous.

We now verify that $F$ is coercive. Since $\rho$ is strictly increasing we have that

$$\rho\left(\|\Phi_\sigma(\boldsymbol{x}) - g\|_{\mathcal{H}_\sigma}\right) \to \infty \quad \text{as} \quad \|g\|_{\mathcal{H}_\sigma} \to \infty,$$

verifying that $\rho\left(\|\Phi_\sigma(\boldsymbol{x}) - \cdot\|_{\mathcal{H}_\sigma}\right)$ is coercive. Next, from the reverse triangle inequality we have that

$$\|\Phi_\sigma(\boldsymbol{x}) - g\|_{\mathcal{H}_\sigma} \geq \left|\|\Phi_\sigma(\boldsymbol{x})\|_{\mathcal{H}_\sigma} - \|g\|_{\mathcal{H}_\sigma}\right| = \left|\sqrt{K_\sigma(\boldsymbol{x}, \boldsymbol{x})} - \|g\|_{\mathcal{H}_\sigma}\right|.$$

Observe that $K_\sigma(\boldsymbol{x}, \boldsymbol{x}) = \|K_\sigma\|_\infty$, and because $\rho$ is strictly increasing we have

$$\rho\left(\left|\|K_\sigma\|_\infty^{\frac{1}{2}} - \|g\|_{\mathcal{H}_\sigma}\right|\right) \leq \rho\left(\|\Phi_\sigma(\boldsymbol{x}) - g\|_{\mathcal{H}_\sigma}\right).$$

Taking expectations on both sides w.r.t. $\mathbb{P}$,

$$\rho\left(\left|\|K_\sigma\|_\infty^{\frac{1}{2}} - \|g\|_{\mathcal{H}_\sigma}\right|\right) \leq \int_{\mathbb{R}^d} \rho\left(\|\Phi_\sigma(\boldsymbol{x}) - g\|_{\mathcal{H}_\sigma}\right) d\mathbb{P}(\boldsymbol{x}) = \mathcal{J}(g).$$

Since

$$\rho\left(\left|\|K_\sigma\|_\infty^{\frac{1}{2}} - \|g\|_{\mathcal{H}_\sigma}\right|\right) \to \infty \quad \text{as} \quad \|g\|_{\mathcal{H}_\sigma} \to \infty,$$

it implies that $\mathcal{J}(g)$ is coercive as well. It follows from this that $F$ is coercive, and the sequence of functionals $\{\mathcal{J}_{\epsilon,\boldsymbol{x}}\}$ is equi-coercive. $\blacksquare$

Propositions B.3 and B.5 together imply, from the fundamental theorem of $\Gamma$-convergence [17, Theorem 7.8], that the sequence of minimizers associated with $\{\mathcal{J}_{\epsilon,\boldsymbol{x}}\}$ converge to the minimizer of $\mathcal{J}$, i.e.,

$$\left\|f_{\rho,\sigma}^{\epsilon,\boldsymbol{x}} - f_{\rho,\sigma}\right\|_{\mathcal{H}_\sigma} \to 0 \quad \text{as} \quad \epsilon \to 0.$$

## B.2  Some Additional Results

Next, we note an important property of the hypothesis class, $\mathcal{G} = \mathcal{H}_\sigma \cap \mathcal{D}_\sigma$. The elements of $\mathcal{G}$ can be shown to have their $\|\cdot\|_\infty$–norm related their $\|\cdot\|_{\mathcal{H}_\sigma}$–norm.

**Lemma B.6** ([39, Lemma 6] and [34, Proposition 5.1]). *For every $g \in \mathcal{H}_\sigma \cap \mathcal{D}_\sigma$,*

$$\|g\|_{\mathcal{H}_\sigma}^2 \leq \|g\|_\infty \leq \|K_\sigma\|_\infty^{\frac{1}{2}} \|g\|_{\mathcal{H}_\sigma}.$$

The following result, which is essentially the population analogue of [39, Lemma 7], guarantees that for small enough $\sigma > 0$, there exists $0 < \delta < 1$ such that $f_{\rho,\sigma}$ is contained in the RKHS ball $B_{\mathcal{H}_\sigma}(\boldsymbol{0}, \delta\nu_\sigma)$, where for brevity we denote $\nu_\sigma = \|K_\sigma\|_\infty^{1/2}$. We provide the proof for completeness, however, the proof uses exactly the same ideas from [39]. For notational convenience, we also define $\psi_\sigma(\|\boldsymbol{x} - \boldsymbol{y}\|_2) = K_\sigma(\boldsymbol{x}, \boldsymbol{y}) = \sigma^{-d}\psi(\|\boldsymbol{x} - \boldsymbol{y}\|_2 / \sigma)$.

**Lemma B.7.** *Let $\mathbb{P} \in \mathcal{M}(\mathbb{R}^d)$ and $f_{\rho,\sigma}$ be the robust KDE for $\sigma > 0$. For sufficiently small $\sigma > 0$, there exists $0 < \delta < 1$ such that $f_{\rho,\sigma} \in B(\boldsymbol{0}, \delta\nu_\sigma)$.*

*Proof.* For $\mathbb{P} \in \mathcal{M}(\mathbb{R}^d)$, and $\mathcal{G} = \mathcal{H}_\sigma \cap \mathcal{D}_\sigma$, consider the map $T_\sigma : \mathcal{G} \to \mathcal{G}$ given by

$$T_\sigma(g) = \int_{\mathbb{R}^d} \frac{\varphi\left(\|\Phi_\sigma(\boldsymbol{x}) - g\|_{\mathcal{H}_\sigma}\right)}{\int_{\mathbb{R}^d} \varphi\left(\|\Phi_\sigma(\boldsymbol{y}) - g\|_{\mathcal{H}_\sigma}\right) d\mathbb{P}(\boldsymbol{y})} K_\sigma(\cdot, \boldsymbol{x}) \, d\mathbb{P}(\boldsymbol{x}) = \int_{\mathbb{R}^d} K_\sigma(\cdot, \boldsymbol{x}) w_\sigma(\boldsymbol{x}) d\mathbb{P}(\boldsymbol{x}),$$

for each $g \in \mathcal{G}$. Observe that $w_\sigma \in L_1(\mathbb{P})$ is a non-negative function such that

$$\int_{\mathbb{R}^d} w_\sigma(\boldsymbol{x}) d\mathbb{P}(\boldsymbol{x}) = 1. \tag{B.6}$$

Let $S_\sigma = \text{Im}(T_\sigma) \subset \mathcal{G}$. It follows from [39, Page 11] that the robust KDE, $f_{\rho,\sigma} = \arg\inf_{g \in \mathcal{G}} \mathcal{J}(g)$, is the fixed point of the map $T_\sigma$ and therefore $f_{\rho,\sigma} \in S_\sigma$. For a small $\epsilon > 0$, from [39, Lemma 12; Corollary 13] there exist $r, s > 0$ such that $\mathbb{P}(B(\boldsymbol{x}, r)) \leq \epsilon$ and $\mathbb{P}(B(\boldsymbol{x}, r + s) \setminus B(\boldsymbol{x}, r)) \leq \epsilon$ for all

$\boldsymbol{x} \in \mathbb{R}^d$. This implies that $\mathbb{P}(B(\boldsymbol{x}, r+s)^c) > 1 - 2\epsilon$. We point out that the constant $\epsilon$ chosen here is related to $\sqrt{9/10}$ used by [39] as $\sqrt{1-\epsilon} = \sqrt{9/10}$, which, as remarked by the authors, was chosen simply for convenience. Define the sets $B_\sigma = B_{\mathcal{H}_\sigma}(\mathbf{0}, \nu_\sigma \sqrt{1-\epsilon})$, and let

$$R_\sigma \doteq S_\sigma \cap B_\sigma^c.$$

In what follows we will show that $f_{\rho,\sigma}$ does not lie in $R_\sigma$. To this end, let $g = \arg\inf_{h \in R_\sigma} \mathcal{J}(h)$. It suffices to show that $\mathcal{J}(g) > \mathcal{J}(\mathbf{0}) > \mathcal{J}(f_{\rho,\sigma})$. Since $g \in R_\sigma$, it must follow that

$$(1-\epsilon)\nu_\sigma^2 < \|g\|_{\mathcal{H}_\sigma}^2 \le \|g\|_\infty = g(\boldsymbol{z}), \tag{B.7}$$

for some $\boldsymbol{z} \in \mathbb{R}^d$, where the second inequality follows from Lemma B.6. Since $g \in S_\sigma$, there exists a non-negative function $w_\sigma$ satisfying Eq. (B.6), such that $g = \int_{\mathbb{R}^d} w_\sigma(\boldsymbol{x}) K_\sigma(\cdot, \boldsymbol{x}) d\mathbb{P}(\boldsymbol{x})$. Therefore,

$$
\begin{aligned}
(1-\epsilon)\nu_\sigma^2 \le g(\boldsymbol{z}) &= \int_{\mathbb{R}^d} K_\sigma(\boldsymbol{z}, \boldsymbol{x}) w_\sigma(\boldsymbol{x}) d\mathbb{P}(\boldsymbol{x}) \\
&= \int_{B(\boldsymbol{z},r)} K_\sigma(\boldsymbol{z}, \boldsymbol{x}) w_\sigma(\boldsymbol{x}) d\mathbb{P}(\boldsymbol{x}) + \int_{B(\boldsymbol{z},r)^c} K_\sigma(\boldsymbol{z}, \boldsymbol{x}) w_\sigma(\boldsymbol{x}) d\mathbb{P}(\boldsymbol{x}) \\
&\overset{(i)}{\le} \nu_\sigma^2 \int_{B(\boldsymbol{z},r)} w_\sigma(\boldsymbol{x}) d\mathbb{P}(\boldsymbol{x}) + \psi_\sigma(r) \underbrace{\int_{B(\boldsymbol{z},r)^c} w_\sigma(\boldsymbol{x}) d\mathbb{P}(\boldsymbol{x})}_{\le 1} \\
&\overset{(ii)}{\le} \nu_\sigma^2 \int_{B(\boldsymbol{z},r)} w_\sigma(\boldsymbol{x}) d\mathbb{P}(\boldsymbol{x}) + \psi_\sigma(r),
\end{aligned}
\tag{B.8}
$$

where (i) follows from the fact that $\sup_{B(\boldsymbol{z},r)^c} K_\sigma(\boldsymbol{z}, \boldsymbol{x}) = \psi_\sigma(r)$ and (ii) follows from Eq. (B.6). From [39, Lemma 7], there exists $\sigma$ small enough such that $\psi_\sigma(r) < \frac{\epsilon}{2}\nu_\sigma^2$. Plugging this back in Eq. (B.8) we get

$$\int_{B(\boldsymbol{z},r)} w_\sigma(\boldsymbol{x}) d\mathbb{P}(\boldsymbol{x}) \ge \left(1 - \frac{3\epsilon}{2}\right). \tag{B.9}$$

Additionally,

$$
\begin{aligned}
\sup_{\boldsymbol{y} \in B(\boldsymbol{z},r+s)^c} g(\boldsymbol{y}) &= \sup_{\boldsymbol{y} \in B(\boldsymbol{z},r+s)^c} \left( \int_{B(\boldsymbol{z},r)} K_\sigma(\boldsymbol{y}, \boldsymbol{x}) w_\sigma(\boldsymbol{x}) d\mathbb{P}(\boldsymbol{x}) + \int_{B(\boldsymbol{z},r)^c} K_\sigma(\boldsymbol{y}, \boldsymbol{x}) w_\sigma(\boldsymbol{x}) d\mathbb{P}(\boldsymbol{x}) \right) \\
&\le \sup_{\boldsymbol{y} \in B(\boldsymbol{z},r+s)^c} \sup_{\boldsymbol{x} \in B(\boldsymbol{z},r)} K_\sigma(\boldsymbol{y}, \boldsymbol{x}) \int_{B(\boldsymbol{z},r)} w_\sigma(\boldsymbol{x}) d\mathbb{P}(\boldsymbol{x}) \\
&\quad + \sup_{\boldsymbol{y} \in B(\boldsymbol{z},r+s)^c} \sup_{\boldsymbol{x} \in B(\boldsymbol{z},r)} K_\sigma(\boldsymbol{y}, \boldsymbol{x}) \int_{B(\boldsymbol{z},r)^c} w_\sigma(\boldsymbol{x}) d\mathbb{P}(\boldsymbol{x}) \\
&\le \psi_\sigma(s) + \nu_\sigma^2 \int_{B(\boldsymbol{z},r)^c} w_\sigma(\boldsymbol{x}) d\mathbb{P}(\boldsymbol{x}).
\end{aligned}
$$

For a choice of $\tau > 0$, there is $\sigma$ small enough satisfying $\psi_\sigma(s) \le \tau$ such that from Eq. (B.9)

$$\sup_{\boldsymbol{y} \in B(\boldsymbol{z},r+s)^c} g(\boldsymbol{y}) \le \tau + \frac{3\epsilon}{2}\nu_\sigma^2. \tag{B.10}$$

Then we have that

$$\mathcal{J}(g) = \int_{\mathbb{R}^d} \rho\left(\|\Phi_\sigma(\boldsymbol{x}) - g\|_{\mathcal{H}_\sigma}\right) d\mathbb{P}(\boldsymbol{x})$$

$$= \int_{B(\boldsymbol{z}, r+s)} \rho\left(\|\Phi_\sigma(\boldsymbol{x}) - g\|_{\mathcal{H}_\sigma}\right) d\mathbb{P}(\boldsymbol{x}) + \int_{B(\boldsymbol{z}, r+s)^c} \rho\left(\|\Phi_\sigma(\boldsymbol{x}) - g\|_{\mathcal{H}_\sigma}\right) d\mathbb{P}(\boldsymbol{x})$$

$$\geq \int_{B(\boldsymbol{z}, r+s)^c} \rho\left(\|\Phi_\sigma(\boldsymbol{x}) - g\|_{\mathcal{H}_\sigma}\right) d\mathbb{P}(\boldsymbol{x})$$

$$= \int_{B(\boldsymbol{z}, r+s)^c} \rho\left(\sqrt{\nu_\sigma^2 + \|g\|_{\mathcal{H}_\sigma}^2 - 2\langle g, \Phi_\sigma(\boldsymbol{x})\rangle_{\mathcal{H}_\sigma}}\right) d\mathbb{P}(\boldsymbol{x})$$

$$\geq \int_{B(\boldsymbol{z}, r+s)^c} \rho\left(\sqrt{\nu_\sigma^2 + \|g\|_{\mathcal{H}_\sigma}^2 - 2\sup_{\boldsymbol{y} \in B(\boldsymbol{z}, r+s)^c} g(\boldsymbol{y})}\right) d\mathbb{P}(\boldsymbol{x}).$$

Plugging in Equations (B.10) and (B.7) we get

$$\mathcal{J}(g) \geq (1 - 2\epsilon)\rho\left(\sqrt{(2 - 4\epsilon)\nu_\sigma^2 - 2\tau}\right).$$

Since $\rho$ is assumed to be strictly convex, this implies that $\rho'$ is strictly increasing. Additionally, from ($\mathcal{A}2$) we have that $\rho'$ is bounded. This implies that, for any $0 < \alpha < \|\rho'\|_\infty$, there is $\beta > 0$ such that $\rho'(z) > \|\rho'\|_\infty - \alpha$ for all $z > \beta$. Using [39, Eq. (11)], we have

$$\rho\left(\sqrt{(2 - 4\epsilon)\nu_\sigma^2 - 2\tau}\right) = \int_0^{(2-4\epsilon)\nu_\sigma^2 - 2\tau} \rho'(z)dz$$

$$\geq \int_\beta^{(2-4\epsilon)\nu_\sigma^2 - 2\tau} \rho'(z)dz$$

$$\geq \int_\beta^{(2-4\epsilon)\nu_\sigma^2 - 2\tau} \left(\|\rho'\|_\infty - \alpha\right) dz$$

$$\geq \left(\|\rho'\|_\infty - \alpha\right)\left(\sqrt{(2 - 4\epsilon)\nu_\sigma^2 - 2\tau} - \beta\right).$$

Without loss of generality, we can assume $\|\rho'\|_\infty = 1$. Choosing $\alpha$, $\tau$ and $\sigma$ small enough we obtain

$$\mathcal{J}(g) \geq \nu_\sigma.$$

Now we note that

$$\mathcal{J}(\boldsymbol{0}) = \int_{\mathbb{R}^d} \rho\left(\|\Phi_\sigma(\boldsymbol{x})\|_{\mathcal{H}_\sigma}\right) d\mathbb{P}(\boldsymbol{x})$$

$$= \rho(\nu_\sigma)$$

$$= \rho(0) + \int_0^{\nu_\sigma} \rho'(z)dz$$

$$\leq \rho(0) + \|\rho'\|_\infty \int_0^{\nu_\sigma} dz = \nu_\sigma.$$

Thus, we obtain that $\mathcal{J}(g) > \mathcal{J}(\boldsymbol{0})$. We have $g = \arg\inf_{h \in R_\sigma} \mathcal{J}(h)$ and $f_{\rho,\sigma} = \arg\inf_{h \in \mathcal{G}} \mathcal{J}(h)$, and, additionally we know that $f_{\rho,\sigma} \neq \boldsymbol{0}$. It follows that since $\mathcal{J}(f_{\rho,\sigma}) \leq \mathcal{J}(\boldsymbol{0}) < \mathcal{J}(g)$, then $f_{\rho,\sigma} \notin R_\sigma$ as $\sigma \to 0$. Taking $\delta = \sqrt{1 - \epsilon}$, we get the desired result. ∎

## C   Supplementary Results for the Persistence Influence

In this section, we collect the proofs for the results on persistence influence established in Section 4.1. The following result shows that when $\varphi$ is nonincreasing, the persistence influence in Eq. (3) can be written in a more succinct form.

**Proposition C.1.** *Under the conditions of Theorem 4.2, if $\varphi$ is nonincreasing, then the persistence influence of $\boldsymbol{x} \in \mathbb{R}^d$ on $\mathsf{Dgm}\left(f_{\rho,\sigma}\right)$ satisfies*

$$\Psi\left(f_{\rho,\sigma}; \boldsymbol{x}\right) \leq \|K_\sigma\|_\infty^{\frac{1}{2}} \, w_\sigma(\boldsymbol{x}) \, \|\Phi_\sigma(\boldsymbol{x}) - f_{\rho,\sigma}\|_{\mathcal{H}_\sigma} \,,$$

*where $w_\sigma$ is the measure of inlyingness from Eq. (2).*

*Proof.* From Theorem 4.2 we have that the persistence influence satisfies

$$\Psi\left(f_{\rho,\sigma}; \boldsymbol{x}\right) \leq \|K_\sigma\|_\infty^{\frac{1}{2}} \, \rho'\left(\|\Phi_\sigma(\boldsymbol{x}) - f_{\rho,\sigma}\|_{\mathcal{H}_\sigma}\right) \left(\int_{\mathbb{R}^d} \zeta\left(\|\Phi_\sigma(\boldsymbol{y}) - f_{\rho,\sigma}\|_{\mathcal{H}_\sigma}\right) d\mathbb{P}(\boldsymbol{y})\right)^{-1}, \quad \text{(C.1)}$$

where $\zeta(z) = \varphi(z) - z\varphi'(z)$. When $\varphi$ is nonincreasing, observe that $z\varphi'(z) \leq 0$ for all $0 \leq z < \infty$. Consequently, $\zeta$ can be bounded below by $\varphi$, and the r.h.s. in Eq. (C.1) can be bounded above by

$$\Psi\left(f_{\rho,\sigma}; \boldsymbol{x}\right) \overset{(i)}{\leq} \|K_\sigma\|_\infty^{\frac{1}{2}} \, \frac{\rho'\left(\|\Phi_\sigma(\boldsymbol{x}) - f_{\rho,\sigma}\|_{\mathcal{H}_\sigma}\right)}{\int_{\mathbb{R}^d} \varphi\left(\|\Phi_\sigma(\boldsymbol{y}) - f_{\rho,\sigma}\|_{\mathcal{H}_\sigma}\right) d\mathbb{P}(\boldsymbol{y})}$$

$$\overset{(ii)}{=} \|K_\sigma\|_\infty^{\frac{1}{2}} \, \frac{\varphi\left(\|\Phi_\sigma(\boldsymbol{x}) - f_{\rho,\sigma}\|_{\mathcal{H}_\sigma}\right)}{\int_{\mathbb{R}^d} \varphi\left(\|\Phi_\sigma(\boldsymbol{y}) - f_{\rho,\sigma}\|_{\mathcal{H}_\sigma}\right) d\mathbb{P}(\boldsymbol{y})} \, \|\Phi_\sigma(\boldsymbol{x}) - f_{\rho,\sigma}\|_{\mathcal{H}_\sigma}$$

$$\overset{(iii)}{=} \|K_\sigma\|_\infty^{\frac{1}{2}} \, w_\sigma(\boldsymbol{x}) \, \|\Phi_\sigma(\boldsymbol{x}) - f_{\rho,\sigma}\|_{\mathcal{H}_\sigma} \,,$$

where (i) follows from the fact that $\zeta(z) \geq \varphi(z)$, (ii) follows from the definition of $\varphi$, i.e., $\rho'(z) = z\varphi(z)$, and (iii) follows from the definition of $w_\sigma$ in Eq. (2), yielding the desired result. $\blacksquare$

The following result establishes the bound for the distance-to-measure described in Eq. (4).

**Proposition C.2.** *For $\mathbb{P} \in \mathcal{M}(\mathbb{R}^d)$, the persistence influence for the distance-to-measure function is given by*

$$\Psi\left(d_{\mathbb{P},m}; \boldsymbol{x}\right) \leq \frac{2}{\sqrt{m}} \sup\left\{\left|f(\boldsymbol{x}) - \int_{\mathbb{R}^d} f(\boldsymbol{y}) d\mathbb{P}(\boldsymbol{y})\right| : \|\nabla f\|_{L_2(\mathbb{P})} \leq 1\right\}$$

*where $\|\nabla f\|_{L_2(\mathbb{P})}$ is a modified, weighted Sobolev norm [31, 41].*

*Proof.* From [10, Theorem 3.5] the following stability result holds:

$$\left\|d_{\mathbb{P},m} - d_{\mathbb{P}_{\boldsymbol{x}}^\epsilon, m}\right\|_\infty \leq \frac{1}{\sqrt{m}} W_2\left(\mathbb{P}, \mathbb{P}_{\boldsymbol{x}}^\epsilon\right).$$

From [31, Theorem 1] we have that

$$W_2\left(\mathbb{P}, \mathbb{P}_{\boldsymbol{x}}^\epsilon\right) \leq 2\left\|\mathbb{P} - \mathbb{P}_{\boldsymbol{x}}^\epsilon\right\|_{\dot{H}^{-1}(\mathbb{P})},$$

where the weighted, homogeneous Sobolev norm $\|\cdot\|_{\dot{H}^{-1}(\mu)}$ for a signed measure $\nu$ w.r.t. a positive measure $\mu$ is given by

$$\|\nu\|_{\dot{H}^{-1}(\mu)} = \sup\left\{\left|\int_{\mathbb{R}^d} f(\boldsymbol{x}) d\nu(\boldsymbol{x})\right| : \|\nabla f\|_{L_2(\mu)} \leq 1\right\}.$$

Observe that $\mathbb{P} - \mathbb{P}_{\boldsymbol{x}}^\epsilon = \epsilon\left(\delta_{\boldsymbol{x}} - \mathbb{P}\right)$ and since $\|\cdot\|_{\dot{H}^{-1}(\mu)}$ defines a norm, we have that

$$\lim_{\epsilon \to 0} \frac{1}{\epsilon} \left\|d_{\mathbb{P},m} - d_{\mathbb{P}_{\boldsymbol{x}}^\epsilon, m}\right\|_\infty \leq \frac{1}{\sqrt{m}} \lim_{\epsilon \to 0} \frac{1}{\epsilon} W_2\left(\mathbb{P}, \mathbb{P}_{\boldsymbol{x}}^\epsilon\right)$$

$$\leq \frac{2}{\sqrt{m}} \lim_{\epsilon \to 0} \frac{1}{\epsilon} \left\|\epsilon\left(\delta_{\boldsymbol{x}} - \mathbb{P}\right)\right\|_{\dot{H}^{-1}(\mathbb{P})}$$

$$= \frac{2}{\sqrt{m}} \left\|\left(\delta_{\boldsymbol{x}} - \mathbb{P}\right)\right\|_{\dot{H}^{-1}(\mathbb{P})}$$

$$= \frac{2}{\sqrt{m}} \sup\left\{\left|f(\boldsymbol{x}) - \int_{\mathbb{R}^d} f(\boldsymbol{y}) d\mathbb{P}(\boldsymbol{y})\right| : \|\nabla f\|_{L_2(\mathbb{P})} \leq 1\right\}.$$

From the stability for persistence diagrams, we have that

$$\Psi\left(d_{\mathbb{P},m}; \boldsymbol{x}\right) \leq \lim_{\epsilon \to 0} \frac{1}{\epsilon} \left\|d_{\mathbb{P},m} - d_{\mathbb{P}_{\boldsymbol{x}}^\epsilon, m}\right\|_\infty$$

and the result follows. $\blacksquare$

**Persistence-Influence Experiment** Points $\mathbb{X}_n$ are sampled from an annular region inside $[-5, 5]^2$ along with some uniform noise in the ambient space, corresponding to the black points in Figure 6 (a). $\mathbb{X}_n$ has interesting $1^{st}$-order homological features. We compute the robust KDE $f^n_{\rho,\sigma}$ and the KDE $f^n_\sigma$ on the points $\mathbb{X}_n$ along with the corresponding persistence diagrams $\mathsf{Dgm}\left(f^n_{\rho,\sigma}\right)$ and $\mathsf{Dgm}\left(f^n_\sigma\right)$. Outliers $\mathbb{Y}_m$ are added to the original points at a distance $r$ from the origin, the number of points roughly equal to $r$. Figure 6 (a) depicts these outliers in orange when $r = 20$. The robust KDE $f^{n+m}_{\rho,\sigma}$ and $f^{n+m}_\sigma$ are now computed on the composite sample $\mathbb{X}_n \cup \mathbb{Y}_m$ along with the persistence diagrams $\mathsf{Dgm}\left(f^{n+m}_{\rho,\sigma}\right)$ and $\mathsf{Dgm}\left(f^{n+m}_\sigma\right)$. The bandwidth $\sigma(k)$ is chosen as the median distance to the $k^{th}$–nearest neighbour of each $\boldsymbol{x}_i \in \mathbb{X}_n$, for the Gaussian kernel with the Hampel loss and $k = 5$.

For the KDE and robust KDE, we compute the $L_\infty$ influence of $\mathbb{Y}_m$ i.e., $\left\|f^{n+m} - f^n\right\|_\infty$ as shown in Figure 6 (d). Additionally for each of the $0^{th}$-order and $1^{st}$-order persistence diagrams, we compute the persistence influence of $\mathbb{Y}_m$, i.e., $W_\infty\left(\mathsf{Dgm}\left(f^{n+m}\right), \mathsf{Dgm}\left(f^n\right)\right)$ as shown in Figures 6 (b, e), and the 1-Wasserstein influence, i.e., $W_1\left(\mathsf{Dgm}\left(f^{n+m}\right), \mathsf{Dgm}\left(f^n\right)\right)$ as shown in Figures 6 (c, f). We refer the reader to Eq. (E.1) in Appendix E for the definition of $W_1$ metric.

For each value of $r$, we generate 100 such samples and report the average in Figure 6. The results indicate that the robust persistence diagrams, $\mathsf{Dgm}\left(f^n_{\rho,\sigma}\right)$, are relatively unperturbed when the outliers are added. It exhibits stability even as $r$ become very large. The KDE persistence diagrams, $\mathsf{Dgm}\left(f^n_\sigma\right)$, on the other hand, are unstable as the outlying noise becomes more extreme.

As discussed in the Remark 4.3(iii), the persistence influence for DTM has a much weaker bound as the outliers become more extreme, and in general is not guaranteed to be bounded. In Figure 7 we illustrate the results from the same experiment when the persistence diagrams from DTM is contrasted with the persistence diagrams from the KDE. This analysis is for the same data as that used in Figure 3. We remark that even though DTM is highly sensitive to extreme outliers, DTM based filtrations have other remarkable properties, as described in [13]. They are very useful for analyzing persistent homology when one has access to just a single collection of points $\mathbb{X}_n$. For DTM the smoothing parameter is chosen as $m(k) = k/n$ with $k = 5$.

Figure 6: (a) An example of $\mathbb{X}_n$ in blue and the contamination $\mathbb{Y}_m$ when $r = 10$. (d) The $L_\infty$ influence of $\mathbb{Y}_m$ on the KDE and robust KDE. (b, e) The bottleneck influence of $\mathbb{Y}_m$. (c, f) The 1-Wasserstein influence of $\mathbb{Y}_m$ as the distance $r$ increases.

Figure 7: For the same data in Figure 6, (a, d) depicts the bottleneck influence for the DTM in contrast to the KDE – the red line is the same as the one from Figure 6 (b, e). Similarly, in (c, e) we see the $W_1$ persistence influence of $\mathbb{Y}_m$ for the DTM in contrast to the KDE. (b) shows the $L_\infty$ influence of $\mathbb{Y}_m$ on the DTM. The robust KDE lines were omitted from all plots as it appears to almost merge with the KDE at this scale.

# D   Additional Experiments with Robust Persistence Diagrams

In this section, we provide information on some additional experiments with the proposed robust persistence diagrams. The experimental setup is the same as in Section 5.

**Random Circles.** The objective of this simulation is to evaluate the performance of persistence diagrams in a supervised learning task. We select circles $\mathbb{S}_1, \mathbb{S}_2, \ldots, \mathbb{S}_{\mathbf{N}}$ randomly in $\mathbb{R}^2$ with centers inside $[0, 2]^2$, with the number of such circles, $\mathbf{N}$ uniformly sampled from $\{1, 2, \ldots, 5\}$. Conditional on $\mathbf{N} = N$, $\mathbb{X}_n$ is sampled uniformly from $\mathbb{S}_1, \ldots, \mathbb{S}_N$ with $50\%$ noise in the enclosing square. Two such point clouds are shown in Figure 8 (a, b). Persistence diagrams $\mathrm{Dgm}\left(f_\sigma^n\right)$ and $\mathrm{Dgm}\left(f_{\rho,\sigma}^n\right)$ are constructed for bandwidth $\sigma(k)$ selected from $k = 5, 7$, and vectorized in the form of persistence images $\mathrm{Img}\left(f_\sigma^n, h\right)$, and $\mathrm{Img}\left(f_{\rho,\sigma}^n, h\right)$ for varying bandwidths $h$ [1]. With $\mathbf{N}$ as the response and the persistence images as the input, results from a support vector regression, averaged over 50 random splits, is shown in Figure 8 (c, d). For a fixed $h$ the robust persistence diagram seems to always contain more predictive information, as observed in the envelope it forms in Figure 8 (c, d).

Figure 8: (a, b) A realization $\mathbb{X}_n$ when $\mathbf{N} = 2$ and $\mathbf{N} = 5$. (c, d) The predicted mean-squared error vs. the persistence image bandwidth for persistence diagrams in support vector regression.

# E    Background on Persistent Homology

Given a set of a points $\mathbb{X}_n = \{\boldsymbol{x}_1 \dots \boldsymbol{x}_n\}$ in a metric space $(\mathfrak{X}, d)$ their topology is encoded in a geometric object called a simplicial complex $\mathcal{K} \subseteq 2^{\mathbb{X}_n}$.

**Definition E.1.** [24]. *A simplicial complex $\mathcal{K}$ is a collection of simplices $\langle \sigma \rangle$ i.e. points, lines, triangles, tetrahedra and its higher dimensional analogues, such that*

1. *$\forall \tau \preccurlyeq \sigma, \sigma \in \mathcal{K}$ we have $\tau \in \mathcal{K}$;*

2. *$\forall \sigma, \tau \in \mathcal{K}$, we have that $\sigma \cap \tau \preccurlyeq \sigma, \tau$ or $\sigma \cap \tau = \phi$.*

For a given spatial resolution $r > 0$, the simplicial complex for $\mathbb{X}_n$, given by $\mathcal{K}(\mathbb{X}_n, r)$, can be constructed in multiple ways. For example, the Vietoris-Rips complex is the simplicial complex

$$\mathcal{K}_r = \{\sigma \subseteq \mathbb{X}_n : \bigcap_{\boldsymbol{x} \in \sigma} B(\boldsymbol{x}, r) \neq \varnothing\},$$

and the Čech complex is given by

$$\mathcal{K}_r = \{\sigma \subseteq \mathbb{X}_n : \max_{\boldsymbol{x}_i, \boldsymbol{x}_j \in \sigma} d(\boldsymbol{x}_i, \boldsymbol{x}_j) \leq r\}.$$

More generally, if $\mathcal{K}$ is a simplicial complex constructed using an approximation of the space $\mathfrak{X}$ (e.g., triangulation, surface mesh, grid, etc.), and $\phi : \mathfrak{X} \to \mathbb{R}$ a filter function, $\phi$ induces the map $\phi : \mathcal{K} \to \mathbb{R}$. Then, $\mathcal{K}_r = \phi^{-1}([0, r])$ encodes the information in the sublevel set of $\phi$ at resolution $r$. Similarly, $\mathcal{K}^r$ encodes the information in the superlevel sets at resolution $r$.

For $0 \leq k \leq d$, the $k^{th}$-*homology* [24] of a simplicial complex $\mathcal{K}$, given by $H_k(\mathcal{K})$ is an algebraic object encoding its topology as a vector-space (over a fixed field). Using the Nerve lemma, $H_k(\mathcal{K}(\mathbb{X}_n, r))$ is isomorphic to the homology of its union of $r$-balls, $H_k(\bigcup_{i=1}^n B_r(\boldsymbol{x}_i))$. The ordered sequence $\{\mathcal{K}(\mathbb{X}_n, r)\}_{r>0}$ forms a *filtration*, encoding the evolution of topological features over a spectrum of resolutions. For $0 < r < s$, the simplicial complex $\mathcal{K}(\mathbb{X}_n, r)$ is a *sub-simplicial complex* of $\mathcal{K}(\mathbb{X}_n, s)$. Their homology groups are associated with the inclusion maps

$$\iota_r^s : H_k(\mathcal{K}(\mathbb{X}_n, r)) \hookrightarrow H_k(\mathcal{K}(\mathbb{X}_n, s)),$$

which in turn carry information on the number of non-trivial $k$-cycles. As the resolution $r$ varies, the evolution of the topology is captured in the filtration. Roughly speaking, new cycles (e.g., connected components, loops, voids and higher order analogues) can appear or existing cycles can merge. Formally, a new $k$-cycle $\sigma_k$ with homology class $[\alpha_k]$ is *born* at $b \in \mathbb{R}$ if $[\alpha_k] \notin \mathrm{Im}(\iota_{b-\epsilon,b}^k)$ for all $\epsilon > 0$ and $[\alpha_k] \in \mathrm{Im}(\iota_{b,b+\delta}^k)$ for some $\delta > 0$. The same $k$-cycle born at $b$ dies at $d > b$ if $\iota_{b,d-\delta}^k([\alpha_k]) \notin \mathrm{Im}(\iota_{b-\epsilon,d-\delta}^k)$ and $\iota_{b,d}^k([\alpha_k]) \in \mathrm{Im}(\iota_{b-\epsilon,d}^k)$ for all $\epsilon > 0$ and $0 < \delta < d-b$. Persistent homology, $PH_*(\phi)$, is an algebraic module which tracks the persistence pairs $(b, d)$ of births $b$ and deaths $d$ across the entire filtration. By collecting all persistence pairs $(b, d)$, the persistent homology is represented as a persistence diagram

$$\mathsf{Dgm}(\mathcal{K}(\mathbb{X}_n)) \doteq \left\{ (b, d) \in \mathbb{R}^2 : 0 \leq b < d \leq \infty \right\}.$$

The persistence diagram is a multiset of points on the space $\Omega = \{(x, y) : 0 \leq x < y \leq \infty\}$, such that each point $(x, y)$ in the persistence diagram corresponds to a distinct topological feature which existed in $\mathcal{K}(\mathbb{X}_n, r)$ for $x \leq r < y$. Given a persistence diagram $\mathbf{D}$ and $1 \leq p \leq \infty$ the *degree-p total persistence* of $\mathbf{D}$ is given by

$$\mathrm{pers}_p(\mathbf{D}) = \left( \sum_{(b,d) \in \mathbf{D}} |d - b|^p \right)^{\frac{1}{p}}.$$

The space of persistence diagrams, given by $\mathcal{D}_p = \left\{ \mathbf{D} : \mathrm{pers}_p(\mathbf{D}) < \infty \right\}$, is endowed with the family of $p$-Wasserstein metrics $W_p$. Given two persistence diagrams $\mathbf{D}_1, \mathbf{D}_2 \in \mathcal{D}_p$, the $p$-Wasserstein distance is given by

$$W_p(\mathbf{D}_1, \mathbf{D}_2) \doteq \left( \inf_{\gamma \in \Gamma} \sum_{\boldsymbol{z} \in \mathbf{D}_1 \cup \Delta} \|\boldsymbol{z} - \gamma(\boldsymbol{z})\|_\infty^p \right)^{\frac{1}{p}}, \tag{E.1}$$

where $\Gamma = \{\gamma : \mathbf{D}_1 \cup \Delta \to \mathbf{D}_2 \cup \Delta\}$ is the set of all bijections from $\mathbf{D}_1$ to $\mathbf{D}_2$ including the diagonal $\Delta = \left\{ (x, y) \in \mathbb{R}^2 : 0 \leq x = y \leq \infty \right\}$ with infinite multiplicity.

Figure 9: Points are sampled from a circular region with adverse outlying noise in the enclosing region. The persistence diagrams from sublevel and superlevel filtration from $f_{\rho,\sigma}^n$ and $d_{n,m}$ respectively are compared with those from the DTM-filtration for $p \in \{1, 2, \infty\}$. The connected components are shown in ● and loops in ▲.

## E.1  Weighted Rips Filtrations

For $p \geq 1$ and a weight function $w : \mathbb{R}^d \to \mathbb{R}$, the $p^{\text{th}}$-power distance from $\boldsymbol{x} \in \mathbb{X}_n$ at resolution $t > 0$ is given by $r_{\boldsymbol{x},w,p}(t) \overset{\cdot}{=} (t^p - w(\boldsymbol{x})^p)^{\frac{1}{p}}$. Anai et al. [2] introduce the weighted-Rips filtration, where the weighted-Rips complex at resolution $t > 0$ is the simplicial complex

$$\mathcal{K}_{t,w,p} \overset{\cdot}{=} \left\{ \sigma \subseteq \mathbb{X}_n : \bigcap_{\boldsymbol{x} \in \sigma} B\left(\boldsymbol{x}, r_{\boldsymbol{x},w,p}(t)\right) \neq \varnothing \right\}. \tag{E.2}$$

The weighted-Rips filtration, $\{\mathcal{K}_{t,w,p}\}_{0 \leq t < \infty}$ is used to construct the persistence diagram $\mathrm{Dgm}\,(\mathbb{X}_n; w, p)$. On the computational front, the construction of $\mathrm{Dgm}\,(\mathbb{X}_n; w, p)$ does not depend on the dimension of the underlying space. As a result, weighted-Rips filtrations are very appealing for applications in high dimensions. In addition, the weighted-Rips filtrations obtained by using the distance-to-measure (DTM) as the weight function, i.e., $\mathrm{Dgm}\,(\mathbb{X}_n; d_{m,n}, p)$, have some very appealing approximation properties [2, Theorems 15 & 20].

We highlight some key differences between our approach and that in Anai et al. [2]. First, as remarked in [2, Section 5], many of the favourable properties of the DTM-filtrations follow from the stability of DTM w.r.t. the Wasserstein Distance. However, it should be noted that stability is inherently different from robustness, as we have described in our analysis using the *persistence influence* in Section 4.1, and particularly, in Proposition C.2. In this context, Figure 9 demonstrates the advantage of our proposed approach in the presence of adverse noise. Second, we note that that our implementations of the robust persistence diagrams use the superlevel filtrations of the robust KDE $f_{\rho,\sigma}^n$ (e.g., see Figure 10), in contrast to weighted-Rips filtrations. While latter is arguably better in higher dimensions, it becomes infeasible for large sample sizes. Notwithstanding, the contributions of [2] provide an interesting direction to pursue using the tools presented here to develop *efficient and robust* persistence diagrams.

(a) Birth at level $r \approx 15$

(b) Birth at level $r \approx 12$

(c) Birth at level $r \approx 8$

(d) Death at level $r \approx 7$

(e) Death at level $r \approx 5$

(f) Connected component continues to $r = 0$

Figure 10: An example for the superlevel filtration of $\phi : \mathbb{R} \to \mathbb{R}$. (a) As the superlevel set enters $r \approx 15$, the first connected component is born, corresponding to the blue dot on the highest peak of $\phi$. The superlevel set for $r = 15$ is depicted in pink below. This is recorded as a birth in the corresponding orange dot enclosed in the pink shaded region of the persistence diagram. (b) As the $r$ enters $r \approx 12$, another connected component is born. This is recorded as the second orange dot in the shaded region of the persistence diagram. (c) Again, at $r \approx 8$, a third connected component is born at the lowest peak of $\phi$. The three connected components in the superlevel set are shaded in pink below the function. The persistence diagram has three orange dots corresponding to these three connected components. (d) As $r$ enters the first valley of $\phi$, depicted by the red dot, two connected components merge (i.e., one of the existing connected components die). By convention, the most recent persistent feature is merged into the older one, i.e., the connected component from (c) merges into the one from (b), and thus, it dies at this resolution. In the persistence diagram, this is noted by the fact that the orange dot born in (c) dies at resolution $r \approx 7$. At this stage, there are only two orange dots in the pink shaded region of the persistence diagram, corresponding to the two pink connected components in the superlevel set of $\phi$. (e) When $r$ enters the second valley of $\phi$, the connected component from (b) merges into the connected component from (a), and form a single connected component. The orange dot in the persistence diagram records the death of this feature. (f) The single connected component persists from then on, and eventually dies at $r = 0$.