[Reviews · NeurIPS 2020]

Review 1

Summary and Contributions: This paper introduces a novel method for obtaining robust topological features. Specifically, it introduces a way to 'robustify' a kernel, such that the resulting topological descriptors are stable with respect to outliers. Next to explaining and illustrating the necessity of such robust kernels, the paper also analyses their theoretical properties, in particular the stability and convergence properties. This theoretical part is, in my opinion, the main contribution of the paper. In addition to a thorough theoretical discussion that even includes bounds, the paper also contains an empirical comparison of different filtrations and their behaviour under noise.

Strengths: The main strength of the paper is the thorough theoretical analysis of the problem at hand. Even though the actual procedure for making the kernel more robust is already known, the analysis of its approximation properties in the context of topological data analysis (TDA) is novel. Moreover, I appreciate the conceptual depth of this paper: stability properties are shown, but also bounds based on entropy. The paper is thus chock full of interesting and relevant theory. The subject discussed in this paper is highly relevant for TDA, as the existence of outliers indeed 'plagues' the calculations to some extent; shedding some light on this topic is thus highly relevant: often, the celebrated stability theorem is misunderstood in applications and large-scale outliers are not handled correctly. Having a more thorough discussion concerning this issue is very relevant and I am glad that the paper raises this point.

Weaknesses: Though I appreciate the thoroughness of the paper, it is also to some extent its main weakness: the main paper is rather dense and requires a thorough reading to be understandable. Given the length of the proofs, only the main results are stated in the text, but some terminology is missing (see my comments on clarity below) to fully understand all results. This goes slightly at the expense of the experimental section, which provides only a very cursory overview of the advantages of the proposed approach. More precisely, I would suggest expanding the experimental section to include experiments that serve to highlight the benefits of persistent homology *and* the new robust kernel. Currently, I fear that the section might be considered to be slightly underwhelming to non-expert readers, because it is not clear what the benefits of a topological view are. I realise that this might be tough to accomplish, but the paper could use examples from Adams et al. (Persistence Images: A Stable Vector Representation of Persistent Homology, https://arxiv.org/abs/1507.06217) to be better comparable to the existing literature. This could be achieved by moving Section 4.2 to the appendix, for example. While I like the results discussed here, they are not strictly required for the experimental section and serve more to illustrate the benefits of the proposed method in comparison to other filtrations. Moreover, I think the paper should delineate itself from the recent paper 'DTM-based Filtrations' by Anai et al. (arXiv:1811.04757), since the main feature of that filtration is also to be robust to noise and outliers. This is particularly relevant as the paper mentions DTM as one way to generate filtrations (notice that DTM-based filtrations are slightly more complex than calculating just DTM).

Correctness: The paper is technically correct; as far as I can tell, the derivation of the theoretical properties is also proper. There are some minor issues with certain claims and terminology: - The claim in l. 29--30 is somewhat debatable: the features generated by the latter approach (evaluating a function on the space) are not the same as the approach based on distances. See Carlsson 'Topological pattern recognition for point cloud data' for more details (in this paper, the approach is called 'Functional Persistence'). - I thought the curvature of the persistence diagram space was *unbounded* (as opposed to *bounded*, which the paper claims in l. 40); is *unbounded* curvature not more problematic? - Technically, $\phi$ in l. 69 is not required to be non-negative (at least not for persistent homology in general). Am I misunderstanding this? - The sequence in l. 71 should be $< \infty$, not $\leq \infty$. - $\phi$ is redefined in l. 121; previously, it referred to a filtration

Clarity: The paper is clearly written for the most part, but there are certain things that could be improved: - when defining filtrations, consider referring to this as a sequence of simplicial complexes, instead of a filtration of spaces (both are correct, but the link to simplicial complexes makes the paper more accessible) - I would suggest using different colours in Figure 1 to highlight the outliers; red is already used to highlight 'loops'. - Consider providing an intuition for 'tame functions' when mentioning them for the first time. - Consider using the term 'topological features' in l. 74 instead of 'cycles'; the latter terminology is slightly confusing to non-experts, who are not familiar with the terminology of cycles and boundaries. - Consider adding 'superlevel features' to Figure 2 instead of the abbreviations. - Consider stating 'persistent homology features' (or something similar) instead of merely 'the persistent homology'. - Consider providing a brief overview of kernel theory such that non-expert readers can also make use of terms such as 'feature maps' (I am familiar with RKHS etc., but since Section 3 is extremely dense in terms of text already, this would be helpful) - Would it be possible to improve the terminology at the beginning of Section 3 (l. 112--) in terms of KDE functions? - Would it be possible to give a more concrete example for $K_\sigma$? - What is $\phi_{\mathbf{P_{\mathbf{x}^{\epsilon}}}$ in l. 166? - Consider defining persistence images in Section 5; their usage here is slightly unexpected and requires more explanations.

Relation to Prior Work: Prior work is adequately referenced for the most part. However, I would suggest that the aforementioned paper 'DTM-based Filtrations' be cited, discussed, and potentially used as a comparison partner.

Reproducibility: Yes

Additional Feedback: All in all, I feel very favourably about this paper, despite my criticisms from above. The only thing preventing me from further endorsing this paper are the issues in the experimental section; I would suggest improving this section to make the paper really 'shine' the way it is supposed to shine. Here are some minor suggestions or comments: - 'have an unusual' --> 'has an unusual' - 'weightage' --> 'weight' - 'notion of influence function' --> 'notion of influence functions' - consider sorting citations by their numbers; this makes longer citation brackets easier to read. - I would suggest not to use citations as nouns. Instead of writing 'the stability results of [10, 14]', consider writing 'while existing stability results [10, 14]' or consider citing the authors of this work directly. - Please check the bibliography for consistency; some of the citations contain unnecessary or redundant parts (ISBNs, inconsistent names for publishers, etc.). --- Update after author response: I thank the authors very much for the rebuttal, which addressed my concerns. I am raising my score accordingly!


Review 2

Summary and Contributions: Despite appealing theoretical properties, the use of "vanilla" topological data analysis (TDA) ---e.g. Rips persistence diagrams---in applications is compromised due to (among other things) its sensitivity to outliers. In this paper, authors improve on this by proposing to compute the persistence diagrams on top of the super-level sets of *robust* kernel density estimation. Their approach is supported by theoretical results (quantitative evaluation of the sensitivity to outliers, convergence guaranties and convergence rates) and its benefits are illustrated in numerical experiments.

Strengths: - The theoretical content of the paper is well motivated and the results are interesting and informative. They are likely to be very useful when studying statistical properties of topological descriptors. - The introduction of metric-geometry notions such as the _persistence influence_ (metric derivative of the Bottleneck distance) to analyze statistical properties of topological descriptors is very appealing.

Weaknesses: - My major concern is that I could not find details about how the persistence diagrams are computed in practice, once the RKDE is evaluated. As far as I know, the practical computation of persistence diagrams on sub/super-level sets in R^d is essentially possible if the filtration has a distance-like structure (as in [29] or using the DTM) or using a discretization of the ground space and using cubical homology (or other variants of simplicial homology). If the former is true, this should be detailed. If the latter, this should be explicitly mentioned (and the impact of the grid-size in terms of quality of the output and running times should be discussed); in particular I assume it makes the practical computations intractable in large dimension. If you manage to compute the persistent homology of the sub/super-level sets of $f^n_rho, sigma$ using another technique, please explain it. - All experiments are done in 2D (perhaps linked with the above remark). The theoretical role played by the ambient dimension $d$ is discussed in Remark 4.2, but experiments in higher dimension (at least 3D) should be discussed/proposed. If this is intractable (e.g. due to requiring a too large grid), this should be mentioned and would consist in a major flaw of the method when targeting numerical applications.

Correctness: Proof of theorem 4.1 has been read carefully and seems correct up to few clarifications needed (see below). Proofs of Theorems 4.2, 4.3 and 4.4 have only been checked briefly; I cannot guarantee their correctness (although looking solid). [EDIT after rebuttal] If the paper got accepted, the proofs should be carefully proofread, as the main contribution of the paper consists of its theoretical content.

Clarity: The paper is very well written and pleasant to read.

Relation to Prior Work: The relation with [19, 11, 29] is well explained and detailed in the background. However, some important references and subsequent discussions are missing, such as: [A] DTM-Based Filtration (SoCG 2018) : https://arxiv.org/pdf/1811.04757.pdf [B] k-PDTM : https://arxiv.org/pdf/1801.10346.pdf Although these papers adopt a different approach (for instance, [A] builds diagrams on top of weighted Rips complexes using the DTM as a way to handle outliers), a discussion of the strengths and weaknesses of the Robust KDE compared to the Rips-DTM-filtration of [A] should be given. [EDIT after rebuttal] I appreciate the comparison between RKDE-PD and DTM-filtration made by the authors in their rebuttal. A detailed remark must be provided in the final version of the paper (in particular, the fact that RKDE-PD might be stable wrt the MMD is an interesting way to understand the difference between these two approaches). I did increase my grade consequently.

Reproducibility: No

Additional Feedback: - l.29-30: "they generate equivalent insight", what supports this claim? A reference should be given. - Given the content of the paper, it might be sufficient to present homology on super-level sets (in the main paper, l.69-77). - l.79: it should be mentioned that points can appear with a multiplicity. - l.104-106: it should be explained somehow that PDs built on KDE/KDist encode more than the topology of supp(P), the distribution of masses is (fortunately) taken into account. - l.113: What is $D_sigma$ here? - l.137-138: is \ell finite? If not, how is it picked in experiments? - Theorem 4.1: Shouldn't it be (A1)-(A4)? - Remark 4.1 (i) is weird. 1/2 z^2 clearly does not satisfy assumptions (A1)-(A4), thus applying Theorem 4.1 is improper. Does it mean "a similar bound holds when rho(z) = 1/2 z^2, reading..."? Please clarify this. - l.179, shouldn't alpha be < 2 ? When alpha=2, the generalized Charbonnier loss does not meet A1. - What would Theorem 4.1 and 4.2 look like when $p$ is finite? Figure 3 suggests that the improvement of using RKDE over simple KDE is even better in W1 (which is not a surprise since W1 is less robust to perturbation than the bottleneck distance). - Theorem 4.1 might have nice implications to assess robustness of (linear?) vectorization built on top of RKDE persistence diagrams (such as the persistence images used in the second experiment). - Some interesting experiments (Fig 6, 7, 8) are worth being included into the main paper. - [Proof of thm 4.1] l.402 : f should be argmin here. - l.406-407, "consequently" follows from Lemma B.1 right? If so, this should be mentioned (and perhaps using a common numbering for Prop and Lemma would serve clarity there, avoiding confusing Lemma B.1 and Proposition B.1). - l.408 : Why does the limit $hat(f)_rho, sigma$ exist? In [25, Theorem 8], this is part of the assumption of the theorem.


Review 3

Summary and Contributions: I thank the authors for the responses (especially the new experimental results). This helps address some of my concerns on empirical contributions of this paper. Nevertheless, my score remains the same (and I remain very positive about the paper). In recent years, persistent homology has been used in many applications to summarize / characterize different types of data. The resulting persistence diagram summary has certain stability guarantee w.r.t. certain perturbations, although in practice, the noise / perturbation often goes beyond the allowed model. Hence an important question is to develop robust persistence summaries (in terms of robust filtration functions to induce the persistence). This paper proposes such a summary for a specific, but arguably a very common setting, where input data is a set of points sampled from a distribution. (Note that persistent homology can be, and have been, applied to many other domains or data types, where the specific filtration introduced in this paper may not be applicable any more. Nevertheless, the point setting is one of the most important settings in data analysis.) In particular, they propose a filtration based on Robust KDE of underlying density function. While on the surface, this may look similar to several prior approaches such as those based on standard KDE or based on the DTM (distance to measures), the paper shows that using the robust estimator has better theoretical guarantees (in terms of sensitivity to noise) as well as better empirical performance. The theoretical results are established based on a generalization of the so-called influence function. While there are empirical results to demonstrate the new robust persistence diagrams are more noise-resilient, I view the development of this robust KDE based filtration, and many theoretical results/statistical analysis about the resulting robust persistence diagrams as the key contribution of this paper.

Strengths: + Having robust persistence based summaries that are robust against various forms of statistical noise is very important for the practical usage of persistence summaries. This paper tackles this problem of a specific but important setting. + While the robust KDE induced filtration seems similar to previous such filtrations, the paper provides a rather comprehensive range of theoretical / statistical analysis of the resulting robust persistence diagrams, and show that this robust persistence diagrams have better robustness/convergence properties. + The paper also presents some empirical results with synthetic data to show that the new diagrams indeed are more noise resilient.

Weaknesses: - The new robust KDE has better properties, however, it is also more expensive to compute the robust KDE (compared to KDE or DTM). The paper provided theoretical time complexity comparison. But it would be good to also show the empirical time needed in the experiments. - In all the empirical experiments, either the data is synthetic, or the noise is artificial. Furthermore, it appears usually the noise added are uniform noise. It would be good to experiment on more realistic datasets, and/or other types of noise.

Correctness: I read the main text carefully. However, I didn't check the technical details in Supplement -- although on the high level, the approaches / results look reasonable.

Clarity: The paper is in general well-written, and also written carefully. I appreciate the authors effort on the various Remarks / Discussions after main results, to help put the theorems (which can be quite hard to interpret) into more context, and provide implications. ** A minor question: In Definition 4.1, what if one uses a different norm for persistence diagrams, instead of W_\infty?

Relation to Prior Work: adequate

Reproducibility: Yes

Additional Feedback:


Review 4

Summary and Contributions: The authors propose to calculate the persistence diagram of a dataset using a robust KDE approach. A robust KDE [25] is a generalized version of kernel density estimation formulated as an M-estimation problem with a plug-in estimation loss (Eq. (1)). When the plug-in loss to be optimized in Eq. (1) is a robust loss, the solution may not be closed form but can be calculated using existing algorithms. Theoretically, the paper extends existing concept of influence function to the context of persistence diagrams, measuring how perturbations of the underlying probability density at a particular point will affect the persistence diagram through an intermediate estimated density function. It is shown that using a robust plug-in loss, the influence function (and its supreme norm, called gross-influence) on the persistence diagram can be proven to have a tighter bound than existing approaches like KDE as filter function, Kernel Distance (KDist) and Distance to the Measure (DTM). The paper proceeds and derive the convergence rates and confidence interval, assuming the plug-in loss is convex.

Strengths: This theoretical paper tackles a very important question in topological data analysis – how the density estimation affects the persistence diagram that characterizes the topology of the data. It uses a more generalized framework (robust KDE) and shows it achieves better robustness than existing results (KDE-based estimation, KDist and DTM). These results can be useful for statistical analysis of the topology of data in downstream applications. It can also be potentially used in downsteam tasks (e.g., [12,40]) relying on persistence homology to characterize the dataset.

Weaknesses: I am assuming the downside of the method is computational time. It would be useful to report the computational time in the experiments to provide a more comprehensive view of the paper.

Correctness: They seem correct. But I did not check all the proofs.

Clarity: Yes,

Relation to Prior Work: Yes.

Reproducibility: No

Additional Feedback: I am a bit worried the proposed method can be hard to reproduce. I hope the authors are willing to share the code in the future.

[Author Response · NeurIPS 2020]

We thank the reviewers for their time and effort in providing insightful and invaluable feedback. Below, we first
address the major common concerns of the reviewers, and then their individual comments. All minor comments will be
addressed in the camera-ready version. Additionally, all code related to experiments will be made publicly available.

**(I) Computational time.** In our experiments, the computational bottleneck is the persistent homology pipeline, and not
the KIRWLS for $f_{\rho,\sigma}$ (RKDE). A simple runtime analysis is presented in Table 1. For $n = 1000$, $\mathbb{X}_n$ is sampled from a
torus in $\mathbb{R}^3$, and the *total* time taken to compute the persistence diagrams (PDs) is reported for several grid resolutions.

**(II) Comparison to DTM-Filtrations.** We highlight some differences between our approach and those in [A, B].
First, as remarked in [A, §5], most properties of the DTM-Filtration follow
from the stability of DTM w.r.t. Wasserstein metric. Similar to DTM, it can be
shown that the robust KDEs (and KDEs) exhibit stability w.r.t. MMD metric
(Maximum Mean Discrepancy). However, it should be noted that stability
is inherently different from robustness, which we have expounded in our
analysis using the *persistence influence*. In this context, the figure indicates the
advantage of our proposed approach in the presence of adverse noise. Second,
we note that we use superlevel filtrations in the experiments, in contrast to
weighted Rips filtrations (which is computationally appealing, especially in
higher dimensions). Notwithstanding, extending our proposed approach (i) to
power-distances for constructing weighted Rips filtrations using the ideas in

[29, §3] and [C, Chap. 5], and (ii) using coresets, as in [B], will be interesting future directions. Finally, we would like
to emphasize that the objective of our work was to illustrate that outlier-robust persistence diagrams can be constructed
without compromising on statistical efficiency, with the hope that the theoretical tools presented here serve as a stepping-
stone for developing *efficient and robust* PDs. We will incorporate and expand on this discussion in the revised version.

**(III) Additional experiment.** We perform a variant of the six-class benchmark experiment in [1, §6.1] to address some
concerns shared by $\mathcal{R}\#1$–$3$. 25 point clouds are sampled from each of six 3D "objects" with additive Gaussian noise
($\sigma = 0.1$), *and* ambient Matérn cluster noise. Dgm $(f_{\rho,\sigma})$ is the PD constructed using $f_{\rho,\sigma}$; and Dgm $(d_{\mathbb{X}_n})$, from the
distance function $d_{\mathbb{X}_n}$, is transformed to the persistence image Img $(d_{\mathbb{X}_n})$. Note that the former is a robust PD while the
latter is a stable vectorization of a non-robust PD. Spectral clustering is performed on the resulting distance-matrices:
$W_p$ metric for Dgm $(f_{\rho,\sigma})$, and $L_p$ metric for Img $(d_{\mathbb{X}_n})$. The quality of the clustering is assessed using the rand-index.
The results are reported in Table 2. We will include a detailed version of this experiment in the revised version.

| Table 1. | 0.04 | 0.06 | 0.08 | 0.10 | Grid-size |
|---|---|---|---|---|---|
| RKDE Dgm | 76.7 | 17.1 | 6.7 | 3.5 | Runtime |
| KDE Dgm | 75.5 | 15.3 | 4.7 | 1.8 | (in Seconds) |

| Table 2. | $W_1/L_1$ | $W_2/L_2$ | $W_\infty/L_\infty$ |
|---|---|---|---|
| $H_0$, Dgm $(f_{\rho,\sigma})$ | 0.9093 | 0.9280 | 0.9032 |
| $H_0$, Img $(d_{\mathbb{X}_n})$ | 0.8612 | 0.8684 | 0.8723 |

**Response to $\mathcal{R}\#1$:** Please see **(II)** and **(III)** regarding the comparison with [A] and experiments. **1.** We agree that the
claim on L.29–30 is questionable, and it will be removed; our intention was to highlight that there are two similar (but
different) approaches. **2.** We agree that the space has curvature unbounded from above, *and* bounded from below [D,
Thm 2.5]. **3.** We use $\phi$(\phi) for filter functions and $\varphi$(\varphi) on L.69. **4.** $\phi_{\mathbb{P}_x^\epsilon}$ is the filter function induced by $\mathbb{P}_x^\epsilon$
on L.167. **5.** As the final version allows an additional page, we will have more space to alleviate the denseness. We will
carefully include all the necessary terminology as well as a concise introduction to kernel theory for enhancing clarity.

**Response to $\mathcal{R}\#2$:** Please see **(II)** and **(III)** regarding the comparison with [A, B] and experiments in 3D. **1.** The
numerical implementation is done using cubical homology; this will be clarified in the revised version. Despite being
infeasible in very high dimensions, this method is still widely used in applications (e.g., [40]). **2.** The runtime analysis
is discussed in **(I)**. The quality of the output can deteriorate if the grid is too fine [19, Lem 11]. **3.** The concern related
to L.29–30 is discussed in **1.** for $\mathcal{R}\#1$. **4.** $\mathcal{D}_\sigma$ is the space of mean embeddings, and is defined in L.65. **5.** $\ell$ is not
pre-defined, but the KIRWLS algorithm is run until the relative change of empirical risk is less than $10^{-6}$. In practice,
we have observed $\ell$ to be well below 100. **6.** Thm 4.1 only uses $(\mathcal{A}1)$–$(\mathcal{A}3)$, and a part of $(\mathcal{A}4)$ is used in Rmk 4.1.
**7.** Rmk 4.1 (i) is meant to say "*a similar bound holds when ...*", and $\alpha < 2$ in (ii). **8.** *Proof*: Lem B.1 will be added and
renumbered, as suggested. We also agree that the existence assumption must be included in the statement to Thm 4.1.
Thank you for pointing this out, and for the insightful comment on the implication of Thm 4.1 for linear vectorizations
of robust PDs. **9.** Replacing $W_\infty$ by $W_p$ and using the $W_p$ stability theorem in [E, §5] ensures that Thms 4.2 & 4.3 still
hold, while Thms 4.1 & 4.4 will require a careful analysis. We will include these clarifications in the revised version.

**Response to $\mathcal{R}\#3$:** Please see **(I)** regarding the empirical time; we will include this information in the revised version.
We also address the concern related to uniform noise in **(III)**, where we consider noise from a Matérn cluster process.
The implication of using $W_p$ instead of $W_\infty$ for examining persistence influence is discussed in **9.** for $\mathcal{R}\#2$.

**Response to $\mathcal{R}\#4$:** Please see **(I)** regarding the concern about computational time. In the final version, we will report
the runtime for all experiments. Regarding the concern of reproducibility, we will make the codes publicly available.

**References.** [A] Anai et al., 2018, SoCG; [B] Brécheteau et al., 2018; [C] Jisu Kim, 2018, PhD Thesis;
[D] Turner et al., 2014, DCG; [E] Cohen-Steiner et al., 2010, FoCM;


[Meta-Review · NeurIPS 2020]

This paper was reviewed and discussed by four expert reviewers. The reviewers were very excited about the theoretical contributions of the paper, but less impressed by the computational limitations and the experimental part. After the thorough rebuttal, several reviewers increased their scores and the paper was ultimately a clear accept. I strongly encourage the authors to follow the revewers' advice with more experiments in the final version to further increase the potential impact of the paper.